# Does dynamically modeled leaf area improve predictions of land surface water and carbon fluxes? - Insights into dynamic vegetation modules

Sven A. Westermann[1], Anke Hildebrandt[1,2], Souhail Bousetta[3], and Stephan Thober[1]

[1]Helmholtz Centre for Environmental Research GmbH - UFZ, Computational Hydrosystems, Leipzig, Germany
[2]Friedrich Schiller University Jena, Institute for Geosciences, Jena, Germany
[3]European Centre for Medium-Range Weather Forecasts, Reading, UK

**Correspondence:** Sven Westermann (sven.westermann@ufz.de)

**Abstract.** Land-surface models represent exchange processes between soil and atmosphere via the land surface by coupling water, energy and carbon fluxes. As a strong mediator between these cycles, vegetation is an important component of land surface models. Some land surface models include modules for vegetation dynamics, which allow the adjustment of vegetation biomass, especially leaf area index, to environmental conditions. Here, we conducted a model-data comparison to investigate whether and how vegetation dynamics in the models improve the representation of vegetation processes and related surface fluxes in two specific models, ECLand and Noah-MP, in contrast to using prescribed values from look-up tables or satellite-based products. We compared model results with observations across a range of climate and vegetation types from the FLUXNET2015 dataset and the MODIS leaf area product, and used on-site measured leaf area from an additional site. Yet, switching on the dynamic vegetation did not enhance representativeness of leaf area index and net ecosystem exchange in ECLand, while it improved performance in Noah-MP only for some sites. The representation of energy fluxes and soil moisture was almost unaffected for both models. Interestingly, the performance regarding variables of the carbon and water cycle was unrelated for both models, such that the weak performance of e.g. leaf area index did not deteriorate the performance of e.g. latent heat flux. We show that one potential reason for this could be that the implemented ecosystem processes diverge from the observations in their seasonal patterns and variability. Noah-MP includes a seasonal hysteresis in the relationship between leaf area index and gross primary production that is not found in observations. The same relationship is represented by a strong linear response in ECLand, which substantially underestimates the observed variability. For both water and carbon fluxes, the currently implemented dynamic vegetation modules in these two models did not result in better model performance compared to runs with static vegetation and prescribed leaf area climatology.

## 20  1  Introduction

Land-surface models (LSMs) represent the energy, water and biogeochemical cycles at the land surface. Traditionally, their main purpose has been to provide a surface component in coupled atmosphere-land models. LSMs are applied in meteorological models, reanalysis products or in the Coupled Model Intercomparison Project (CMIP). However, their scope is widening and new fields of application like historical land cover change simulations (Lawrence et al., 2018) or flood alert services (Har-
rigan et al., 2020) are arising. There is active development within the land surface modeling community, with more and more features being added to existing models to make them more realistic (Blyth et al., 2021).

Given the wide use of these models and the implications of their results, extensive model validation has been done already. Model validation covers a wide range of water, energy and carbon fluxes at global, regional and site scale (e.g. Niu et al., 2011; Haverd et al., 2018; Lawrence et al., 2019; Boussetta et al., 2021). Such works that introduce individual evaluation schemes are
often accompanied by studies that perform comparisons between models (e.g. Best et al., 2015; Krinner et al., 2018). Comparisons like those are conducted for different reasons. For example, one aim is to create a ranking between models that allows the assessment against alternative schemes. Using this method, Best et al. (2015) reported that simple statistical methods achieve a higher performance in energy partitioning at eddy-covariance sites than any single LSM tested. One limitation of that study is that they did not report metrics of individual model performance, but only normalized ones. This procedure does not allow
to judge whether the investigated methods have achieved a (dis-)satisfactory performance, since all methods might have a poor individual model performance. Other challenges in these activities are to maintain a standard protocol for model comparison, while not creating a superficial performance contest among them, and to minimize human errors (Menard et al., 2021).

Haughton et al. (2016) more closely explored the cause of poor model performance of LSMs shown in the PLUMBER study by Best et al. (2015), which they presented as the bias for the evaporative fraction (EF) derived from various tower sites exemplar-
ily. From all investigated aspects they concluded that mismatches between modeled and observed heat fluxes are most likely caused by calculations within the models and not related to errors in the observations. Yet, specific reasons for this mismatch, for example over-parameterization, missing processes, calibration issues etc., cannot be identified by benchmarking studies or model rankings alone, but requires further investigation of individual model performance. At the same time, the causes of poor model performance can be multifaceted, rendering their identification challenging (Haughton et al., 2018b). Nonetheless,
further LSM development needs understanding of how individual process implementation and parameterization affect model performances.

A wealth of studies evaluated different LSMs with respect to radiation, heat fluxes or surface temperature, and carbon fluxes. Carbon fluxes like gross primary production (GPP) are often validated by using global gridded fluxes like FLUXCOM (Ma et al., 2017; Jung et al., 2019; Lawrence et al., 2019). The correct implementation of ecosystem processes and related vari-
ables is crucial for using LSMs in assessing impacts due to climate change for example in drought evaluation (Ukkola et al., 2016; Dirmeyer et al., 2021) because plant transpiration directly links the terrestrial carbon and water cycle. For example, a substantial underestimation of evapotranspiration by eight LSMs during drought conditions was shown across different plant communities (Ukkola et al., 2016). De Kauwe et al. (2015) concluded from their simulations of drought responses for the

European FLUXNET sites with the Community Atmosphere Biosphere Land Exchange (CABLE) model that accounting for differing drought sensitivity of plant communities into LSMs may be required to correctly capture drought impacts. Currently, most LSMs are not able to represent direct vegetation control on surface exchange, in part because they under-represent biophysical responses to changing water availability and oversimplify vegetation dynamics, in particular leaf area index (LAI) (Forzieri et al., 2020). LSMs typically work with climatological LAI, e.g. seasonality read from look-up table files, or calculate LAI as a prognostic variable internally. At the same time, LAI has a large impact on both water and carbon fluxes (e.g. Fisher et al., 2014), and an understanding of how its parameterization impacts flux estimates by LSMs would help to shed light on the known discrepancies in representing vegetation.

Here, we investigate model performance for water and carbon fluxes with a focus on vegetation processes. We additionally check the reasons for model-data mismatch, by analysis of the underlying computer source code of the models (as stated by Dirmeyer et al. (2018)), which can only be done for a limited set of models due to the large effort that is needed. For this scope, we chose ECLand and Noah-MP as frequently used and continuously developing LSMs with available modules for vegetation dynamics. In this manuscript, we aim to answer the following research questions: (1) Does the representation of net ecosystem exchange (NEE) and LAI improve, if ECLand or Noah-MP represent vegetation dynamically? (2) How does dynamic vegetation in ECLand or Noah-MP impact other variables like heat fluxes and soil moisture? Do improvements in model performance for one variable compromise performance for other variables? (3) What are the mechanics behind modeled temporal patterns in vegetation dynamics and occurring misfits to the observations?

## 2 Methods

### 2.1 Data basis

#### Site selection

The FLUXNET2015 dataset (Pastorello et al., 2020) provides measurements from globally distributed eddy covariance sites. We selected a subset from all available FLUXNET sites, focusing on sites with long observation periods, covering different vegetation types and a gradient in aridity within each vegetation type. Vegetation types within FLUXNET rely on the IGBP Land Classification (National Center for Atmospheric Research, 2022). The aridity index of all sites was retrieved from the CGIAR-CSI Global-Aridity and Global-PET Database (Trabucco and Zomer, 2018) and inverted afterwards, bringing it back to the definition as the ratio of the long-term mean annual potential evapotranspiration to the long-term mean annual precipitation by Budyko (1974). We excluded sites with observation periods less than six years because they might not represent the local climate (Haughton et al., 2018a) and extreme years could create a systematic bias. Due to the small number of sites per vegetation type with long observation periods, the vegetation types savanna (SAV), woody savanna (WSA) and open shrubland (OSH) were merged into one savanna group before continuing with the selection procedure. For each vegetation type or group, first, we chose the site with the longest observation record. Next, other sites with similar aridity ($\pm0.1$ logarithmic aridity index) were dropped to avoid an overrepresentation of some vegetation type-aridity combinations due to heterogeneous site

distribution within FLUXNET. We used logarithmic values to create a linear scale of the aridity index so that a selection of too many dry sites was prevented. Afterwards, we repeated these steps for the remaining sites and continued until no more sites were available for selection in this vegetation type or group. For the selected sites, we double-checked data availability and quality and replaced with an alternative site if necessary. The most common reasons for discarding sites were missing or poor
quality soil moisture data or low-quality gap-filling, which reduced the length of the observation record below the threshold of six years. By doing so, only two sites with mixed forests (MF) were left which is critically few. Thus, we included all MF sites into the deciduous broadleaf forest (DBF) vegetation type and repeated the selection for this group. We were left with 24 sites, covering a wide range of site characteristics as recommended by Haughton et al. (2018a) including aridity, vegetation types and observation periods (Fig. 1). Additionally, we also used data of the eddy covariance site "Hohes Holz" (Rebmann
and Pohl, 2023) which is part of the TERENO Harz/Central German Lowland Observatory (Wollschläger et al., 2016) and is included in the ICOS network since 2019, because on-site measured LAI data was available for that DBF site.

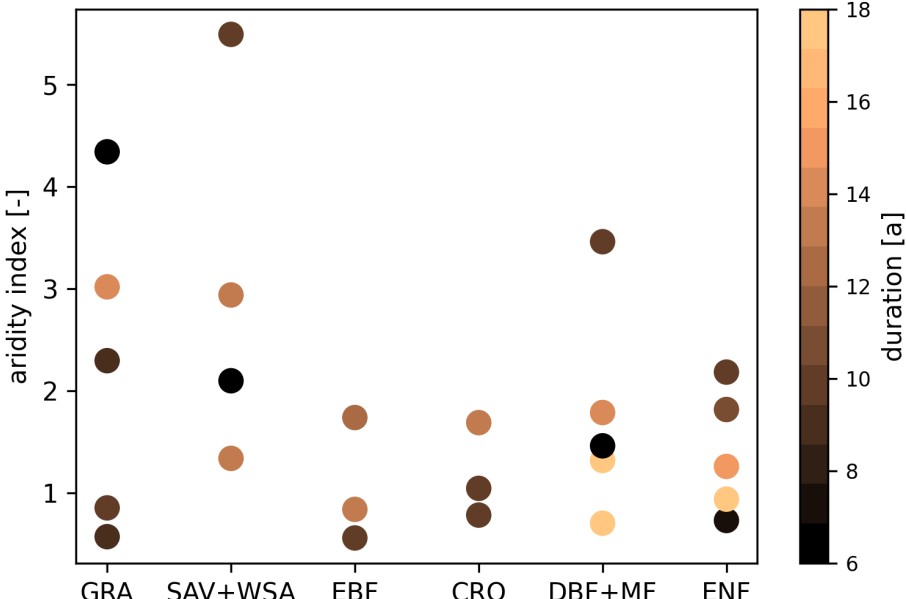

**Figure 1.** Selected FLUXNET sites grouped by their vegetation type. For each group, sites were chosen to cover a gradient in aridity (y-axis) if available. The vegetation types are: GRA - grassland, SAV - savanna, WSA - woody savanna, EBF - evergreen broadleaf forest, CRO - cropland, MF - mixed forest, DBF - deciduous broadleaf forest, ENF - evergreen needleleaf forest. The color scale represents the duration of the available time series in years.

**Variables used and data pre-processing**

From the FLUXNET (Pastorello et al., 2020) and Hohes Holz (Rebmann and Pohl, 2023) datasets, air temperature, downward short- and long-wave radiation, wind speed, relative humidity, air pressure and precipitation were used for model forcing.

Turbulent fluxes, i.e. latent heat flux (LE) and sensible heat flux (H), as well as net ecosystem exchange (NEE), gross primary production (GPP) and volumetric soil water content in $10\ cm$ depth were used for model evaluation. All data were provided and used at half-hourly resolution. FLUXNET data was retrieved from their website (fluxnet.org, 2020).

LE and H in FLUXNET2015 are available in two different variables: One is a product that corrects the turbulent fluxes for energy balance closure, while the other one provides a continuous time series filled by Marginal Distribution Sampling. We

decided to use the first one as long as they were available in the dataset since LSMs also consider for energy balance. Missing data in the "Hohes Holz" meteorological dataset was filled using a Kalman filter (Sayed, 2003) for short gaps up to $3\ h$, except for precipitation which was set to 0. For longer gaps, the Kalman procedure tends to overestimate the observations which resulted in offsets at the end of the filling periods. Thus, filling data for these gaps was retrieved from the ERA5 (Hersbach et al., 2020) data product (Copernicus, 2018) with $0.1°$ spatial and $1\ h$ temporal resolution.

For calculation of the evaporative fraction $\frac{LE}{LE+H}$, all time steps with $H \leq 0$ were excluded. The same time steps were left out for LE to focus the comparison of turbulent fluxes on periods with evaporative demand. For estimation of model performance, we excluded gap filled periods that were longer than one month.

## 2.2   Model description

We investigated how dynamic vegetation affects model outputs in two land-surface models capable of representing both static

and dynamic vegetation: ECLand (Balsamo et al., 2009; Dutra et al., 2010; Boussetta et al., 2021) and Noah-MP (Chen and Dudhia, 2001; Ek et al., 2003; Niu et al., 2007, 2011).

### ECLand

The European Centre for Medium-range Weather Forecasts (ECMWF) developed a Carbon-Hydrology Tiled Scheme for Surface Exchanges over Land (CHTESSEL) (Balsamo et al., 2009; Dutra et al., 2010; Boussetta et al., 2013) which represents the

land component of the Integrated Forecasting System (IFS). As part of the IFS, CHTESSEL has evolved into a more flexible system ECLand (Boussetta et al., 2021), which also allows for several modular extensions. Among these, a dynamic vegetation module simulates the temporal evolution of vegetation. Therein, LAI, vegetation biomass and vegetation coverage are calculated from the daily carbon budget, instead of taking them from the climatological LAI. However, LAI climatology can still be used for fully static or partly dynamic simulations.

In ECLand (IFS version "CY46R1"), each of the 19 vegetation types receives its own parameter values (e.g. for roughness lengths, stomata resistance, root distribution) from look-up tables (Boussetta et al., 2012, 2021). These vegetation types are categorized into high or low vegetation. Each grid-cell has one dominant high and one dominant low vegetation type, together forming the vegetation of a grid-cell (Balsamo et al., 2009). Surface fluxes are computed for the high and low vegetation tiles separately then merged for the whole grid-cell according to their fractional cover. The vegetation coverage is calculated from

a prescribed climatological vegetation fraction (part of input) and a vegetation type dependent density (from look-up table) and corrected by current LAI (Boussetta et al., 2021). Net assimilation results from carbon uptake of atmospheric $CO_2$ by the current leaf area (defines absorbed radiation) and is restricted by environmental factors such as soil moisture and nitrogen

availability (important equations can be found in section A.01). Together with the dark respiration and after scaling with a quantum use efficiency factor, potential gross assimilation is calculated. This value, then, is linearly linked to LAI and the humidity-corrected air density, resulting in gross primary productivity (GPP). With activated vegetation dynamics, a potential net assimilation, together with LAI, forms a damping factor for biomass senescence. Biomass senescence is determined from current biomass, linearly linked to current LAI, and the damping factor. The change in biomass results from this updated biomass and the net assimilation. Then, biomass is updated again and linearly transferred into updated LAI by using specific leaf area from a look-up table (Boussetta et al., 2021). For static ECLand, the prescribed climatological LAI is used. LAI in ECLand determines the canopy resistance for water vapour transport and thus, the evapotranspiration as well as the interception (Boussetta et al., 2012, 2013, 2021).

**Noah-MP**

Noah-MP is the widely used community Noah land-surface model (Chen and Dudhia, 2001; Ek et al., 2003) with multi-parameterization options (Niu et al., 2007, 2011). Predicted LAI in Noah-MP is calculated based on leaf carbon allocation and specific leaf-area per vegetation type (Ma et al., 2017). In contrast to ECLand, Noah-MP can either use prescribed LAI values per vegetation type or depend solely on dynamic LAI estimates, without the option to mix between the two.

In Noah-MP (version "HRLDAS 3.9"), parameter values (e.g. value range of stomatal resistance, number of rooted soil layers, specific leaf area) of the 27 vegetation types are taken from look-up tables. The vegetated sub-grid area of each grid cell is dominated by one vegetation type forming a one-layer canopy. Calculation of canopy interception and transpiration consider aerodynamic and stomatal resistances for the water vapour and carbon fluxes within the canopy and between the canopy and the atmosphere (Ma et al., 2017). Among others, stomatal resistance is predominantly controlled by photosynthesis (Niu et al., 2011) which depends on leaf area, and is limited by light and root zone soil moisture (important equations can be found in section A.02). Assimilation depends on LAI and is constrained by physiology and light availability. Assimilated carbon is allocated to different plant tissues (leaf, stem, wood, root), forming GPP, and reduced by respiration, dying and turnover processes such as drought stress and senescence representing leaf dynamics (Dickinson et al., 1998). Respiration rate is determined by LAI, GPP, temperature and soil moisture stress. Carbon that is allocated to leaves together with biomass losses forming an updated leaf biomass which converts into the LAI by using specific leaf area (Ma et al., 2017). Carbon assimilation and allocation and, thus, also GPP and NEE estimation are usually deactivated for the static Noah-MP since a prescribed LAI is given. We adapted the model code in a way that GPP and NEE for the static simulations are calculated anyways, but resetting all variables that would be dynamically predicted within the same function.

## 2.3 Model setup and simulations

Simulations with activated modules that predict LAI time series will be *activated vegetation dynamics* or *dynamic ECLand* and *dynamic Noah-MP* hereafter. For both models, the reference height (level of the forcing input) was set to the flux tower height of the sites which depends on the vegetation type. The models were set up as closely as possible to the available site information but there are some technical differences in the structure of the model input, i.e. in the initial files. Forcing and

model calculation were done in 30 minutes resolution if available, otherwise, hourly resolution was applied. We used four layered soil representation and used the uppermost layer for evaluation of soil moisture which is 7 $cm$ and 10 $cm$ deep for ECLand and Noah-MP, respectively. Every simulation started with a ten year spin-up phase by recalculating the first year.

**ECLand**

We used ERA5-based (Hersbach et al., 2020) global initial data for ECLand and selected the grid cells where the flux towers are located. These initial files contain information on albedo, orography, soil type, surface roughness and monthly LAI which is not available in the FLUXNET metadata. For the simulations that use alternative LAI forcing, monthly LAI in the initial files was replaced by the scenario specific alternative values (see section 2.3). We defined the vegetation on that grid-cell to be either high or low vegetation (and not a mixture) depending on the site information. Forests and savannas were treated as high

vegetation types while grasslands and croplands were allocated to low vegetation types. The vegetation type that fits most to the FLUXNET characterization was selected (see Tab. 1). The coverage of that vegetation type was set to 100 %. Meteorological forcing was taken from the FLUXNET/TERENO data sets mentioned above (section 2.1). The ECLand simulations were done with van Genuchten soil hydrologic parameters (van Genuchten, 1980), activated sub-grid surface runoff and activated snow parameterization.

**Table 1.** Assignment of vegetation types used in ECLand and in Noah-MP and referred initial LAI. The values in brackets for Noah-MP initial LAI refer to sites on the Southern Hemisphere due to shifted seasons.

| Fluxnet vegetation type | ECLand vegetation type | ECLand vegetation class | Noah-MP USGS | Noah-MP vegetation class | Noah-MP initial LAI |
|---|---|---|---|---|---|
| ENF | Evergreen Needleleaf Trees | 3 (high) | Evergreen Needleleaf Forest | 14 | 4.0 |
| MF | Mixed Forest/Woodland | 18 (high) | Mixed Forest | 15 | 2.0 (4.3) |
| DBF | Deciduous Broadlead Trees | 5 (high) | Deciduous Broadleaf Forest | 11 | 0.0 (4.5) |
| EBF | Evergreen Broadleaf Trees | 6 (high) | Evergreen Broadleaf Forest | 13 | 4.5 |
| SAV | Interrupted Forest | 19 (high) | Savanna | 10 | 0.3 (3.8) |
| WSA | Interrupted Forest | 19 (high) | Savanna | 10 | 0.3 (3.8) |
| CRO | Crops, Mixed Farming | 1 (low) | Mixed Dryland/Irrigated Cropland and Pasture | 4 | 0.0 (3.0) |
| GRA | Tall Grass | 7 (low) | Grassland | 7 | 0.4 (3.5) |

**Noah-MP**

Soil type for Noah-MP was taken from a global soil grid (Hengl et al., 2014) by selecting the grid cell including the flux tower location. Initial values for temperatures and soil moisture were taken as the FLUXNET/TERENO observations at January 1st 00:00 h in the first year of the simulation period. Vegetation types were chosen to match as closely as possible the USGS vegetation types (University Corporation for Atmospheric Research, 2023) and the initial LAI values were set according to

the defaults in the parameter file (see Tab. 1). Vegetation cover fraction was set to $100\%$ so that the entire grid-cell represents the vegetation type of the observation site. Minimum green vegetation fraction was set to $1\%$ to ensure that not the whole vegetation cover dies during winter which would hinder temperate short vegetation from growing in spring. For the simulations with alternative LAI forcing, the monthly LAI in the look-up table was replaced by the scenario specific alternative values (see section 2.3). The Noah-MP simulations were done with soil parameterization from look-up tables, Ball-Berry stomatal

resistance approach (Ball et al., 1987; Bonan, 1996) with using matric potential limitation. All other selected options can be found in Table 2.

Table 2. Options chosen for Noah-MP parameterization.

| Physical process | Noah-MP Option |
| --- | --- |
| Runoff and groundwater | TOPMODEL with groundwater (Niu et al., 2007) |
| Surface layer roughness | Monin-Obukhov |
| Supercooled water | no iteration (Yang and Niu, 2006) |
| Radiative transfer | two-stream (vegetated vs. vegetation-free) |
| Snow albedo | fresh snow with aging effects |
| Rain/snow partitioning | threshold temperature at $2.2\,^\circ C$ |
| Lower boundary for soil temperature | temperature at $8\,m$ depth (part of input) |
| Snow/soil temperature time scheme | fully implicit (original Noah) |
| Surface resistance for evaporation | Sakaguchi and Zeng (2009) |
| Glacier treatment | phase change of ice included |

**Leaf area index data and scenarios**

Monthly LAI values are part of the initial input of both models via look-up tables. These tables contain annual cycles of LAI for each vegetation type separately. This *default climatology* is already based on values from MODIS. For ECLand, the gridded

values of LAI were disaggregated to the high and low vegetation type of the grid cell for the time span 2000-2008 (Boussetta et al., 2013). For alternative LAI inputs, these values in the look-up tables were replaced manually.

LAI values were taken from the MOD15A2H data product from NASA's EarthData portal (Myneni et al., 2015). One grid cell of $500\,m$ x $500\,m$ was selected per eddy covariance tower according to the site coordinates and LAI values with temporal resolution of eight days were extracted for the years 2000 to 2014. To assure reliability of the values, the "MODIS15A2H"

data product comes with numeric quality flags. Although Fang et al. (2012) recommend using all values with quality flags less than 64, we excluded data with quality flag 8 because many of these LAI values were extremely low during the vegetation period which is unrealistic. Then again, due to lacking LAI values during winter or wet seasons, values with quality flags of 73 (empirically filled with clouds present), 81 (empirically filled with mixed cloudiness) and 97 (empirically filled for other reasons) were included as a trade-off between excluding as much bad-flagged data as possible and keeping roughly the same

amount of data values for each month (see MODIS documentation for more details). Afterwards, we smoothed the remaining

values by using a Savgol filter (window length: 11, polyorder: 2) (similarly done by e.g. Xiao et al., 2011; Huang et al., 2021) from the scipy-package (Savitzky and Golay, 1964; Luo et al., 2005) and prepared a mean annual LAI cycle for all available years with monthly resolution, further named *MODIS climatology*. For an additional experiment, the monthly LAI from MODIS of each year within the simulation period separately was used as input, called *MODIS single-year* from this point on. Missing LAI values for a month were filled by the average value of the adjacent months. If LAI values for at least two consecutive months were not available, the LAI values from the default look-up table were used for those months.

For the "Hohes Holz" site, on-site measured LAI data was available from Digital Cover Photography (DCP), which was shown to yield comparable results to established methods (Piayda et al., 2015). For each measurement date, we averaged the values from the whole plot area and, afterwards, calculated monthly means over time span 2014-2019. This alternative LAI forcing will be called *on-site LAI* hereafter. The nomenclature of all LAI scenarios can be found in Table 3.

**Table 3.** Nomenclature of all model scenarios using LAI data sources.

| Term | LAI source |
|---|---|
| default climatology | default monthly LAI for the dominant high and low vegetation type on respective grid cell (ECLand) or default monthly values per vegetation type from look-up table (Noah-MP) |
| MODIS climatology | mean annual cycle of monthly LAI values derived from MODIS dataset from 2000 to 2014 |
| MODIS single-year | same as before but without averaging, resulting in an annual cycle for each year separately within the observation period |
| on-site LAI | mean annual cycle of monthly LAI values based on on-site measured LAI |

The MODIS LAI was also applied for model evaluation but in high temporal resolution of eight days. Due to the usage of single day values, we solely used data with quality flags 0 (no issues) and 32 (saturated) to lower the uncertainty. Additionally, we refrained from smoothing to avoid an offset of the LAI values and left gaps as they were. For the static runs, comparison with MODIS LAI on daily basis provides the information how well a LAI climatology represents the local LAI evolution and whether an incorporation of more site-specific climatology can improve the representativeness. For the dynamic simulations, comparing modeled LAI with daily MODIS values is used to examine whether the models are able to capture inter- and intra-annual LAI dynamics.

## 2.4 Performance evaluation

Model outputs and observational data from the flux towers were averaged/summed to daily values for direct comparison. For LAI, we calculated the eight-day mean of the LAI model output to correspond to the temporal resolution of the MODIS LAI estimates. As performance criteria we used the Pearson's correlation coefficient, the normalized standard deviation and a modified relative bias for the model-observation relationship. Pearson's correlation coefficient $R$ describes the fit between model and observation values (Benesty et al., 2009) and is calculated from the numpy-package. The normalized standard deviation $s_n$ is the ratio of the standard deviation of the model predictions and the standard deviation of the observations. It is used to describe the models' ability to reproduce the variability of the observations. The relative bias $b$ applied here was

adapted to the domain of the variable to avoid division by zero or by values very close to zero (especially important for NEE). For this purpose, the distribution of the observed values was shifted by their minimum, resulting in only positive values with a minimum of zero:

$$b = \frac{\overline{y - x}}{\overline{x} - \check{x}} \tag{1}$$

whereby $y$ represents the model predictions, $x$ the observations, $\overline{x}$ the mean and $\check{x}$ the minimum of the observed values. To compare the model performance of simulations with static and dynamic vegetation, we determined the change in relative bias as follows:

$$\Delta b = |b_{static}| - |b_{dynamic}| \tag{2}$$

Negative values mean that the relative bias of the dynamic simulation was greater than that of the static simulation and, thus, that the performance was reduced by activating vegetation dynamics.

To investigate the sensitivity of dynamically modelled vegetation on the model performance, we checked how strongly the quality of the model simulation of one target variable (e.g. LE) depends on the model quality of another one (e.g LAI). For this, we used the *elasticity* as a metric. Elasticity is calculated as ratio of the change in one statistical measure (analogous to equation 2) for two different target variables:

$$E = \frac{\Delta m_i}{\Delta m_j} \tag{3}$$

where $m$ is one of the statistical measures mentioned above, i.e. $R$, $s_n$ or $b$, while $i$ and $j$ denote different target variables, e.g. GPP or LE. For variables that are strongly related, like LAI and GPP, we expect elasticity to be positive. Two variables are considered independent if $-0.1 \leq E \leq 0.1$ because the change in $m_j$ then would need to be larger than one order of magnitude to cause a change in $m_i$. Changes in model performances of the target variables were plotted in Taylor diagrams (Copin, 2021).

## 3 Results

### 3.1 Effect of dynamic or prescribed leaf area index on leaf area and carbon uptake prediction

Figure 2 shows the quality metrics for the model performance regarding LAI in a Taylor diagram. The location an optimal model simulation would occupy is indicated with a star. The model performance of the dynamic run is shown with the symbols, while the static runs can be read from the start of each arrow. The direction and length of each arrow highlights the difference in the performance metrics between static and dynamic runs. Shown are simulations started (dynamic) or run (static) with *default* vs. *MODIS climatology*.

While in the Noah-MP simulations with static vegetation the model performance depended on the LAI forcing applied, the simulation results were unaffected by the type of LAI forcing with vegetation dynamics switched on since the symbols in Figure 2 c and d have the same positions. For ECLand, this was also the case for many sites but not all, e.g. *AT-Neu* and *AU-How* (Fig. 2 a+b). Initializing ECLand with *default climatology* (Fig. 2 a) and activating vegetation dynamics generally increased

the variance of simulated LAI compared to static simulations but it also decreased model performance, e.g., mean Pearson correlation decreased from 0.72 to 0.62. At the same time, whether the predicted LAI fit better to MODIS observations than *default climatology* was ambiguous, as can be seen by the shift in relative bias which ranged between −0.5 and 1.3. On the contrary, the results for Noah-MP showed a different pattern (Fig. 2 c) because there was no clear shift to higher variances or worse correlation when activating vegetation dynamics. Especially short (GRA+CRO) or sparse (SAV+WSA) vegetation types had the highest changes towards decreased but also enhanced model performance for LAI. For other sites (mostly forests), modelled dynamic LAI correlated well with the observations.

For both models, using *MODIS climatology* instead of *default climatology* in static simulations resulted in the best performances with regard to LAI of all simulations (start of the arrows in Fig. 2 b+d), e.g., the mean correlation coefficient increased to 0.83 and 0.84 and mean relative bias (Tab. A1) improved to −16% and −2% for ECLand and Noah-MP, respectively. This can be expected because MODIS was also used as reference dataset for LAI evaluation. With activated vegetation dynamics, the performance of both models decreased, as all quality metrics shift away from the point indicating best performance in the Taylor diagram (Fig. 2 b+d). The same applied to the relative biases of LAI since their shift was predominantly negative. In other words, switching on vegetation dynamics did not result in improved LAI representation compared to just using *MODIS climatology*.

Forest ecosystems, in general, were better represented by model predictions with vegetation dynamics than short or sparse vegetation. Figure 3 shows the results of the forest site "Hohes Holz" in more detail. Although the representation of LAI variability detoriated when simulating dynamic vegetation with Noah-MP, those runs resulted in LAI predictions that closely match MODIS observations (Fig. 3 d-f), represented by a relative bias of −18% and a correlation coefficient of 0.78. ECLand more generally suffered from larger relative biases in LAI, especially when simulating with vegetation dynamics (−30% on average, Fig. 3 c). The only scenario where model performance generally increased for ECLand, was through switching on vegetation dynamics compared to static runs with *default climatology*.

In contrast to LAI, the model performance of ecosystem exchange variables in ECLand was less affected by activating vegetation dynamics. A common feature is that the variance predominantly increased when using dynamic vegetation (Fig. 4 a+b). Mostly, sites with short or sparse vegetation reacted more sensitively to dynamic vegetation modeling in their NEE and GPP representation especially when forcing with *MODIS climatology*, which is indicated by the longer arrows in Figure 4 a and b (for GPP see Fig. A1 in Appendix). For forest ecosystems in general, the changes in the model performance of NEE and GPP were small, as also shown for the site "Hohes Holz" (Fig. 3 a-c). Nevertheless, the performance of NEE (and GPP) decreased when activating vegetation dynamics, mainly driven by lowered correlation coefficients, on average from 0.41 to 0.37 (0.72 to 0.68). Only three sites showed improvements in NEE representation when predicting with dynamic ECLand and just one did so for GPP. Relative bias changed in both directions, towards lower and higher model performance. Dynamic ECLand mainly overestimated NEE by 11% on average, indicating that ecosystems were predicted to be a smaller carbon sink than observed (Tab. A2). Instead, dynamic Noah-MP estimated on average 10% lower NEE compared to the observations for the most sites (Fig. 4 c+d, Fig. 3 c+f).

Activating the dynamic vegetation affected the model performance of NEE and GPP for Noah-MP heterogeneously. Some sites

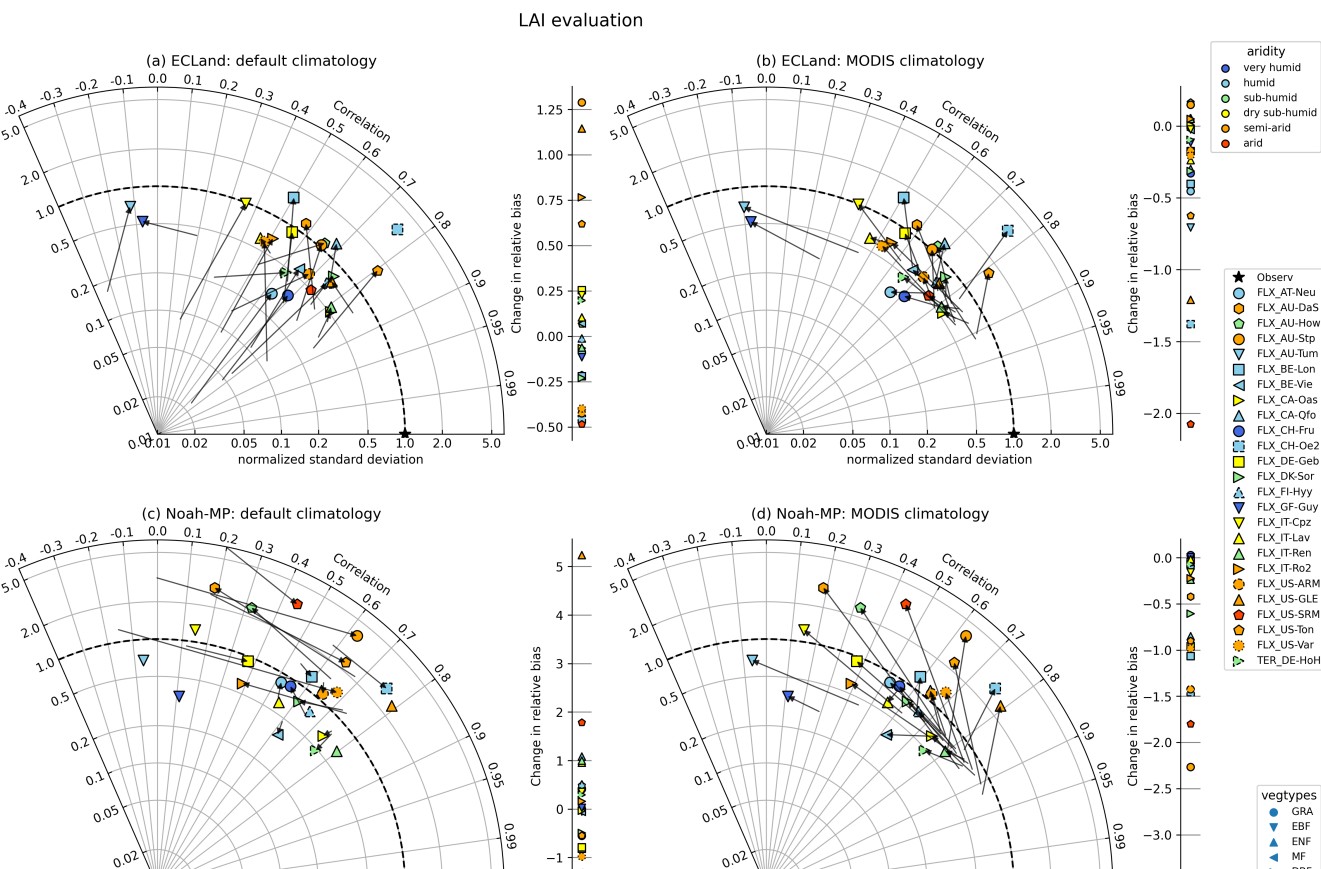

**Figure 2.** Change of model quality metrics for LAI when switching on vegetation dynamics for all included sites and by using *default climatology* (left) or *MODIS climatology* (right). The star ("Observ") marks the location of the perfect correlation between observation and model and perfect agreement between observed and modelled variance. The model performance of the static runs can be read from the start of each arrow. When no arrow appears, either no correlation could be calculated (e.g. for evergreen forests where default climatological LAI is constant) or values could not be placed on the logarithmic axis. The symbol colors indicate the site aridity (top right legend) as following: very humid - aridity index $(AI) < 0.6$, humid - $AI < 1.25$, sub-humid - $AI < 1.54$, dry sub-humid - $AI < 2$, semi-arid - $AI < 5$, arid - $AI \geq 5$ (Ashaolu and Iroye, 2018). Vegetation types are symbolized by different marker types (bottom right legend).

showed very small changes (e.g. *IT-Lav* and *IT-Ren*, Fig 4 c+d) while model performance of NEE was largely impacted by vegetation dynamics for other sites (e.g *US-Var* and *AU-DaS*). In contrast to ECLand, no evidence could be found that certain vegetation types or aridity classes were more sensitive to activated vegetation dynamics in Noah-MP and even forests showed larger changes in model performance (Fig. 3 e). While, for GPP, the variance predominantly enlarged by activating vegetation dynamics in Noah-MP (Fig. A1 c+d), the normalized standard deviation of NEE changed in opposing directions which is another difference compared to ECLand. Despite the higher sensitivity of NEE model performance to Noah-MP vegetation

dynamics, the overall model performance was barely affected since relative bias shifted from $-12\,\%$ to $-11\,\%$ and correlation coefficient from $0.50$ to $0.53$ on average. Changes of statistical measures can be in opposing directions as can be seen for normalized standard deviation and relative bias of the forest site "Hohes Holz" (Fig. 3 e+f) which eliminated trends towards improved or reduced model performance. Only the site *AU-Stp* clearly improved regarding NEE representation by activating vegetation dynamics in Noah-MP by initializing with either *default* or *MODIS climatology*. The GPP performance showed small improvements by activating vegetation dynamics in Noah-MP as the mean correlation coefficient shifted from $0.68$ to $0.74$ and the range of the relative bias was lowered from $-32\,\% - +69\,\%$ to $-28\,\% - +42\,\%$.

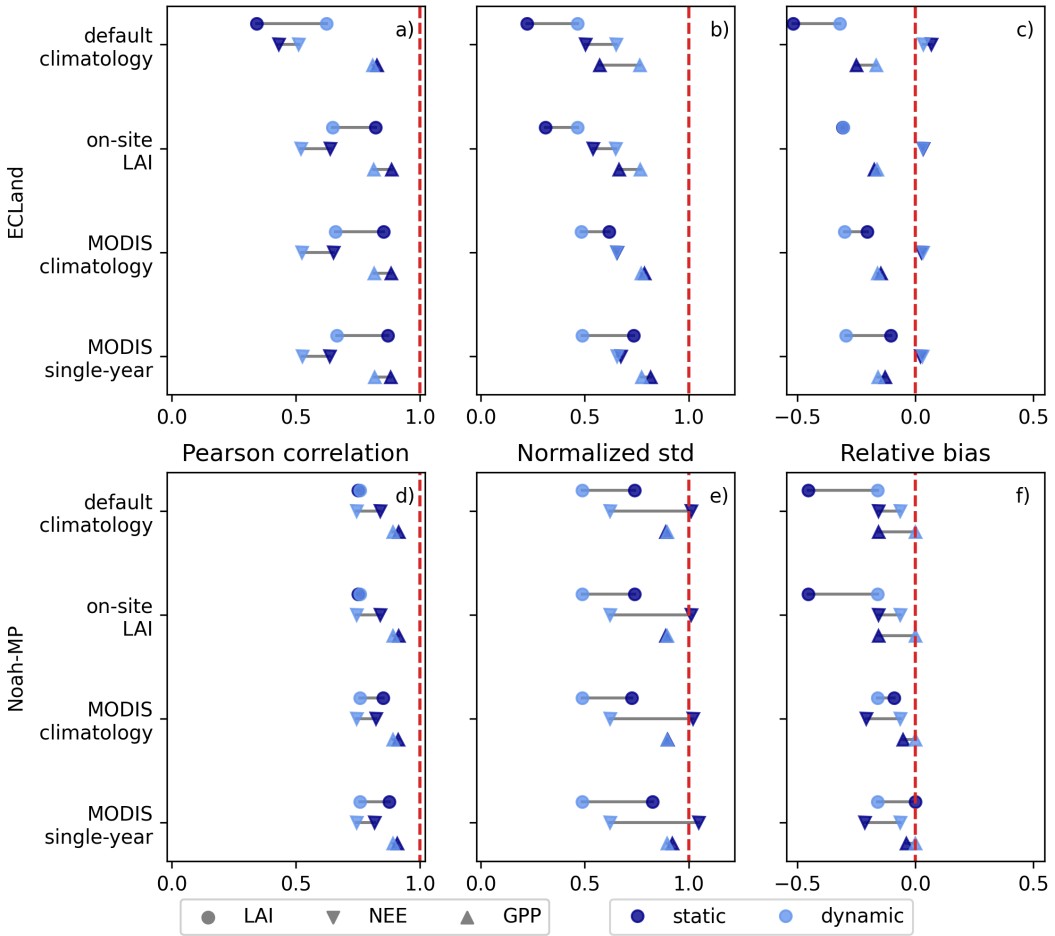

**Figure 3.** Statistical measures for the variables LAI, NEE and GPP of the model runs for the site "Hohes Holz". The categories on the y-axis mark the different LAI forcings. Statistical measure of the static and dynamic simulations of the same variable are connected by a horizontal line. The red dotted vertical line marks the optimum of each measure.

In general, Noah-MP seemed to capture NEE representations better as the mean deviance from a normalized standard deviation of 1 was $0.33$ (ECLand: $0.39$) and showed with $0.51$ a higher correlation coefficient on average than ECLand (Fig. 4 c).

Remarkably, the four and the nine best sites regarding NEE correlation and variance were forests for ECLand and Noah-MP, respectively. At the same time, all evergreen broadleaf forests suffered from low performance in both models. GPP representation in both models was better than for NEE (Fig. A1, Tab. A3). Overall, static and dynamic Noah-MP performed well in representing NEE and GPP for most forest sites apart the evergreen broadleaf forests.

In line with the finding that model performances of dynamic Noah-MP were independent of the prescribed LAI forcing, the availability of *on-site LAI* data for the site "Hohes Holz" yielded no improvement in the representation of NEE or GPP compared to other LAI climatology (Fig. 3). The same appeared for dynamic ECLand. Forcing static ECLand with *on-site LAI* data resulted in NEE and GPP correlation and relative bias comparable to the forcing with *MODIS climatology*, only variability was lower.

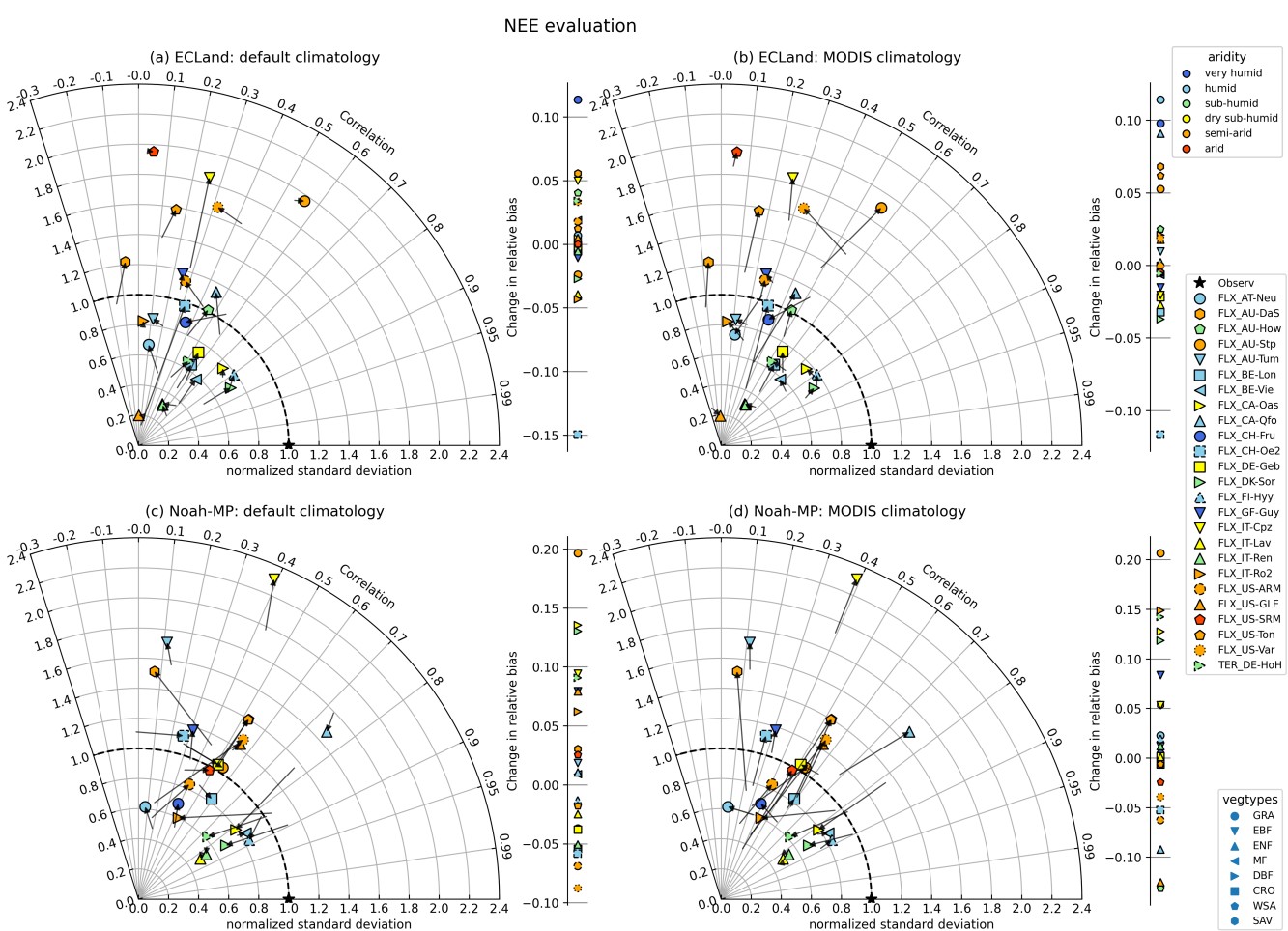

**Figure 4.** Same as Taylor diagram before but with NEE evaluation.

## 3.2 On the sensitivity of heat fluxes and soil moisture in LSMs to vegetation dynamics

For both models, activating vegetation dynamics had a small impact on the representation of turbulent fluxes and soil moisture. The strongest changes occurred for short or sparse vegetation types or for drier climates which had the largest arrows in the Taylor diagrams (Fig. 5, Fig. 6). In ECLand, activating vegetation dynamics enhanced the variance of latent heat flux for the most sites (from $0.80$ and $0.84$ to $0.94$ on average for *default* and *MODIS climatology*, respectively), but correlation between simulated and observed values remained unaffected or even diminished (mean change smaller than $-0.03$). For several sites, LE estimates from dynamic ECLand better represented the observations as shown by the positive shift in relative bias (reduction from $-32\%$ to $-21\%$) (Fig. 5 a, Tab. A4), but no relationship regarding vegetation type or site aridity can be seen and changes are small in general. The big exception appeared for *CH-Oe2* which was caused by its default LAI climatology that did not fit the vegetation type.

Activating vegetation dynamics in Noah-MP hardly affected model performance of LE (mean change in correlation was $0.02$, in standard deviation $0.00$ and in relative bias $0.02$). Sites that showed some sensitivity predominantly have drier climate (e.g *AU-Stp*, *US-Var*, see Fig. 5 c). Several sites showed less bias in LE predictions when using dynamics vegetation predictions in Noah-MP. When using *MODIS climatology* as LAI forcing, activating vegetation dynamics could be advantageous for some sites regarding LE representation (*AU-Stp*, *CH-Fru*, *US-GLE*), but mostly it would not lead to higher model performance.

Model performance regarding the evaporative fraction (EF) was lower compared to LE as points are further away from the point of optimal model performance (Fig. 6). Running ECLand with activated vegetation dynamics lowered the representation of the evaporative fraction which is demonstrated by many points in the Taylor diagram drifting away from the star indicating best performance. Thereby, the mean standard deviation changed from $0.95$ to $1.08$ and correlation coefficient was reduced slightly from $0.48$ to $0.46$ on average (Fig. 6 a+b). Exceptions were *BE-Lon*, *US-SRM* and *US-Ton* where model performance slightly improved regarding correlation and variability. Again, relative bias of EF changed in both directions without any trend regarding vegetation type or aridity for both models (see also Tab. A5). For Noah-MP, eight sites showed an improved representation of the evaporative fraction when running the model with vegetation dynamics. This amount was reduced to six when the model was initialized with *MODIS climatology*. But changes were very small on average.

Regarding soil moisture, the model performance was almost insensitive to the used vegetation dynamics option or the type of LAI forcing for both models (Fig. A2). Some sites showed improvement of soil moisture prediction by activating vegetation dynamics for both models although the improvement was very weak. Interestingly, no humid site was among them. However, the simulation of soil moisture resulted in a broad range of model performances starting with very well-fitting predictions ($R > 0.9$, $b \approx 0\%$) up to very poor-fitting predictions ($R < 0.2$, $b < -40\%$ or $b > 100\%$, see Tab. A6).

To investigate the sensitivity of dynamically modeled vegetation on the model performance, we checked how strongly the quality metrics of NEE, GPP, LE and soil moisture change with the quality metrics of LAI. For this, we used the elasticity (defined in equation 3) as a metric which is summarized for all sites in Figure 7. Surprisingly, the quality metrics of those closely related variables were independent from each other, i.e. the elasticity was very low (within grey band) or randomly distributed around zero. The strongest connection of all pairs tested could be found for the correlation coefficient between LAI

and GPP in ECLand when using *MODIS climatology* but without affecting normalized standard deviation or relative bias. Here,
the mean elasticity of correlation and normalized standard deviation is positive, meaning that, as expected, an increased model
performance in LAI co-occurs with enhanced performance for GPP in the same order of magnitude. In a similar manner, NEE
and LAI performance were positively related regarding correlation coefficient in Noah-MP. Other elasticity values that include
LAI were small predominantly. In other words, changes in the model quality for LAI, for most of the sites, do not affect the
model performance of LE or soil moisture and even not that of carbon fluxes.

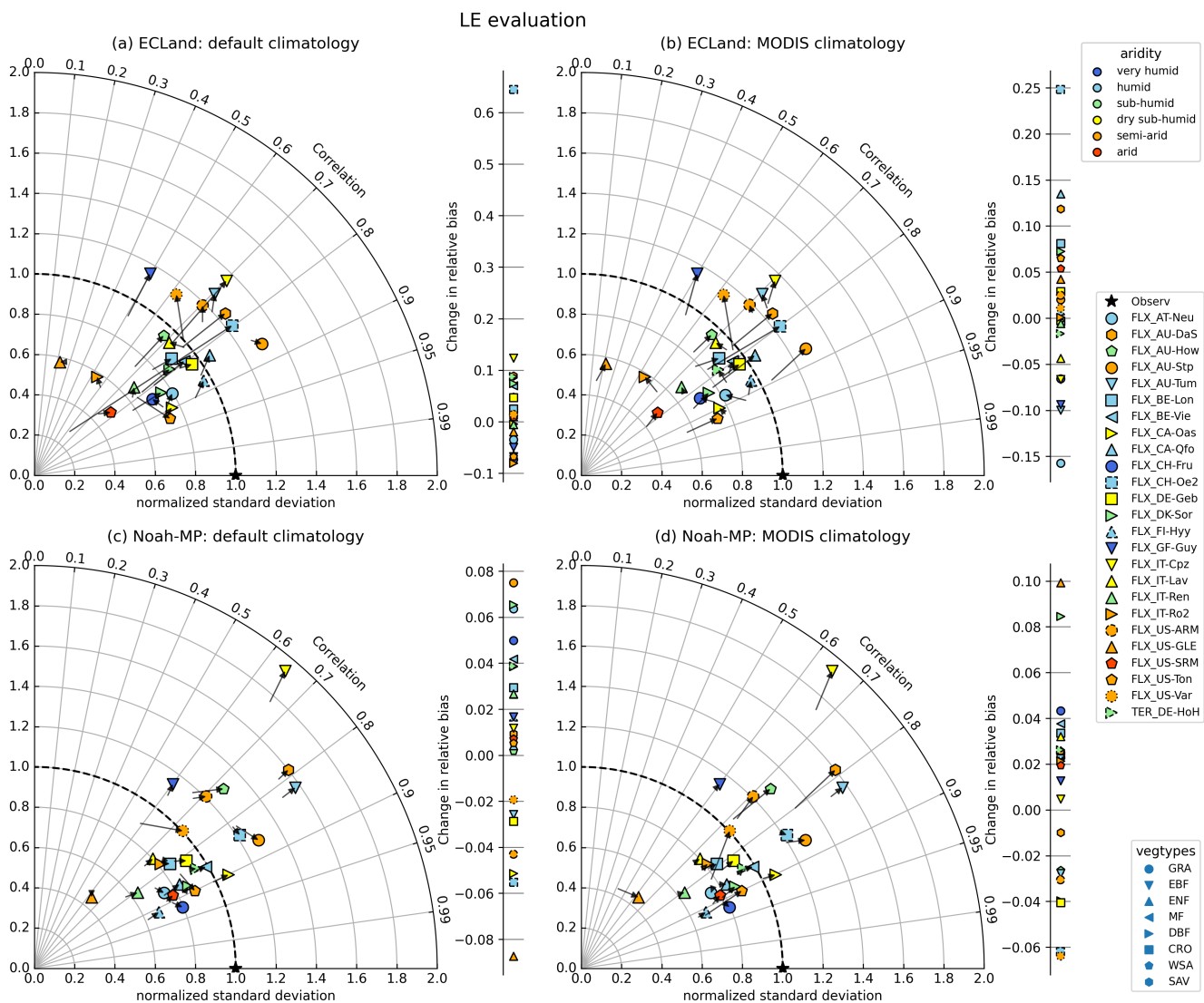

**Figure 5.** Change of statistical measures for LE modeling when switching on dynamic vegetation for all included sites and by using *default climatology* (left) or *MODIS climatology* (right) as forcing.

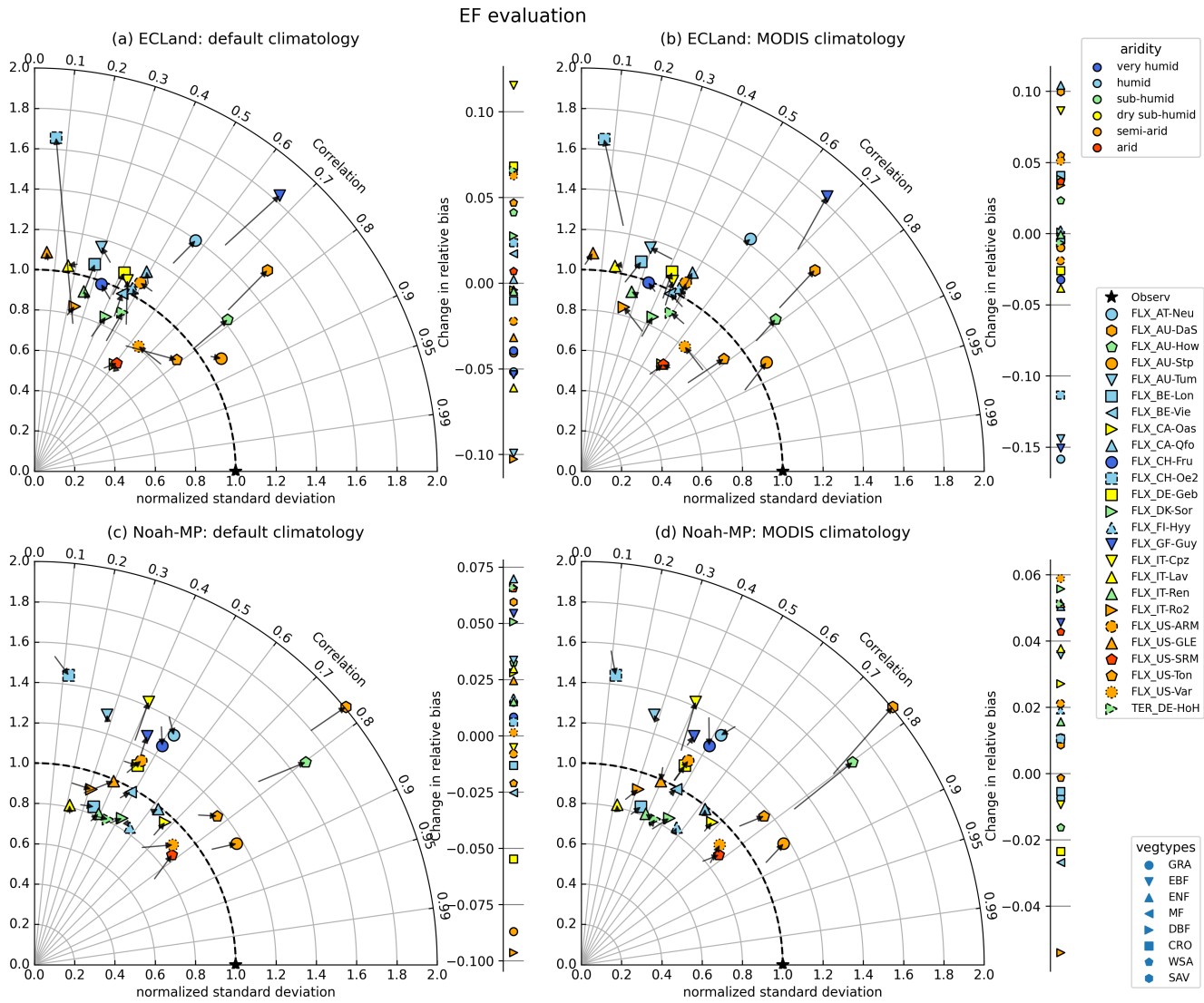

**Figure 6.** Same as before but for evaporative fraction which represents the turbulent flux partitioning.

## 3.3  Observed and simulated relationships between ecohydrological variables

One possible explanation for the small contribution of model quality of LAI to that of the turbulent fluxes could be a weak relation between LAI and carbon exchange in the model. However, this is not the case as illustrated in Figure 8. The figure shows the relation between GPP and LAI for four exemplary sites: *DE-HoH* is a deciduous broadleaf forest in a humid climate, *IT-Ren* is an evergreen needleleaf forest in a semi-arid climate, *GF-Guy* is an evergreen broadleaf forest in a tropical climate, and *US-Var* is a grassland in a semi-arid climate. In general, the relationships between GPP and LAI is much more scattered in

the observations (top row) compared to the models (other rows), and this is true for both models, across biomes and vegetation types.

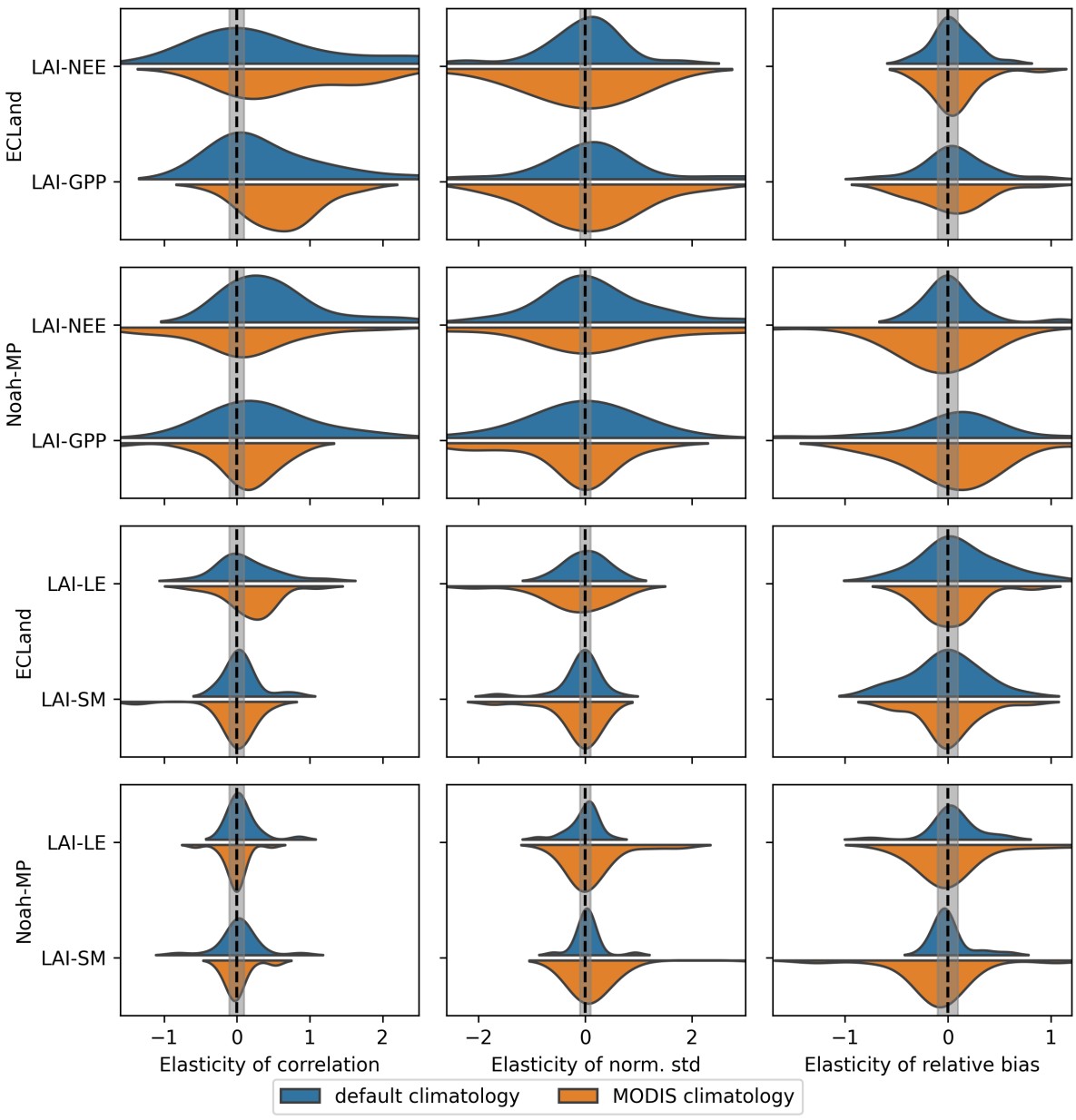

**Figure 7.** Density plots showing the elasticity of correlation (left column), normalized standard deviation (middle column) and relative bias (right column) for different variable relationships in both models when activating dynamic vegetation and using *default climatology* (blue) or *MODIS climatology* (orange) as forcing. For reasons of practicability, elasticity is used reciprocal. Accordingly, the explanatory variable is the first one of each relationship showed on the y-axis. The grey shaded area marks the range between the thresholds of independence.

The two European sites (left columns, *De-HoH* and *IT-Ren*) reach maximum LAI and GPP in JJA and minimum values in DJF, leading to a correlation that is mainly governed by the seasonal cycle. Similarly, at the U.S. site, with an overall tighter relation, vegetation productivity and LAI peak together in spring (i.e., MAM). For these three sites, correlation coefficients range

between 0.80 to 0.86 indicating a clear but not perfect relation between LAI and GPP. However, the scatter of the observed relation is considerable with the standard deviation of the residuals ($\sigma_r$) being between 58 and 102 $\cdot 10^{-6}$ $gCO_2m^{-2}s^{-1}$. The variance is highest for the peak of the growing season, when GPP quickly responds to environmental conditions (e.g., cloudiness, precipitation, and soil moisture stress) that LAI responds much slower to. The tropical site in French Guiana (*GF-Guy*) shows, as expected, no seasonal cycle, leading to an extremely weak relation between LAI and GPP. The latter is

comparatively high all year round (GPP between 250 and 600 $\cdot 10^{-6}$ $gCO_2m^{-2}s^{-1}$) although LAI values from the MODIS dataset surprisingly varied between 1 and 7 $m^2m^{-2}$. For this tropical site, GPP and LAI dynamics seem decoupled (Fig. 8 c). Noah-MP shows a non-linear relationship with a pronounced hysteresis effect at all sites except the tropical one (Fig. 8 m-p). Thereby, GPP increases linearly with LAI during biomass built-up up to a point where allocation to leaves becomes minimal (vegetation type specific), and drops considerably without any substantial reductions in LAI towards the end of the growing

season (e.g., Fig. 8 m). When GPP values reduce below approximately 100 $\cdot 10^{-6}$ $gCO_2m^{-2}s^{-1}$, then LAI reduces from values about three towards zero. This hysteresis is shifted in seasons due to local climate as for the site *US-Var* (Fig. 8 p). At the tropical site, Noah-MP shows some variability in GPP, but almost no change in LAI which is around a value of five.

ECLand, in general, shows a linear relationship with considerable less variability compared to the observations. The slope and intercept of the linear regression is dependent on the choice of static or dynamic vegetation. Dynamic ECLand shows a very

tight linear relation between LAI and GPP with much lower scatter compared to the observations (Fig. 8 third row) as $R$ is larger than 0.99 and $\sigma_r$ is between 10 and 14 $\cdot 10^{-6}$ $gCO_2m^{-2}s^{-1}$ for all non-tropical sites. With slope values of 104 to 254 $\cdot 10^{-6}$ $gCO_2m^{-2}s^{-1}$, that relationship is much steeper than in the observations. Even for the tropical site, the relationship between LAI and GPP is clearly and tightly linear (Fig. 8 k).

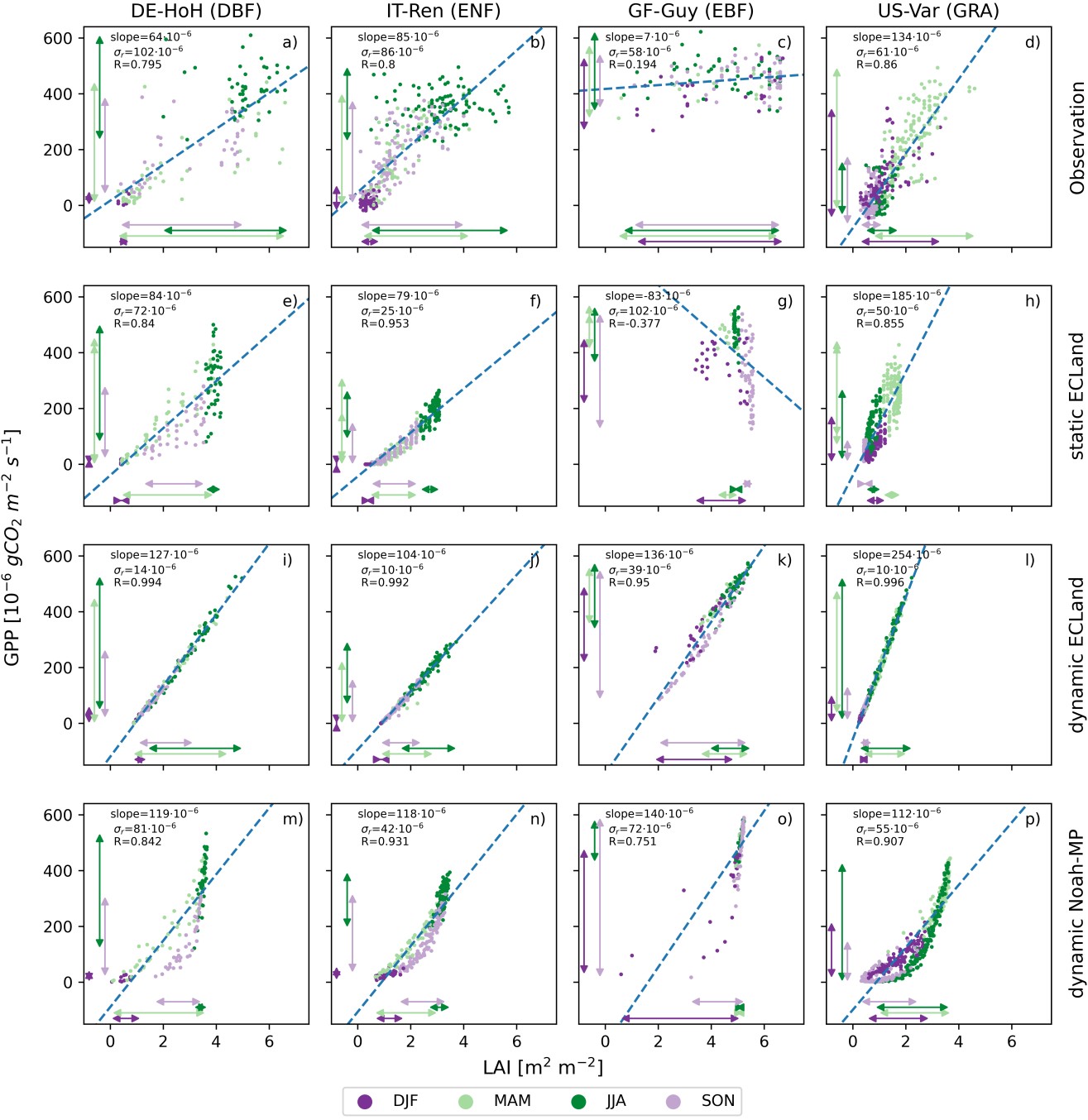

**Figure 8.** Scatter plots of the relationship between LAI on the x-axes and GPP on the y-axes as 8-day averages for four selected sites (columns). The rows from top to down show observations, static ECLand model output, dynamic ECLand model output, and dynamic Noah-MP model output. Seasons are represented by different dot colors. The arrows represent the range of GPP and LAI values for the individual seasons. A simple linear regression model was applied (blue dashed line) and its correlation coefficient (R), slope and standard deviation of the residuals ($\sigma_r$) are given for each relationship.

## 4 Discussion

 ### 4.1 Using LAI climatology for ECLand and Noah-MP runs is the best way to reproduce leaf area and carbon uptake

Comparison between model output and observational data of LAI, NEE or GPP on a daily basis is rarely done. The ability of the two models to reproduce these observed ecosystem variables was in line with previous results. For Noah-MP, model quality metrics were in the range of other studies (Brunsell et al., 2020; Li et al., 2022; Xu et al., 2021; Liang et al., 2020). However, dynamic LAI modelled by Noah-MP in our assessment with a mean of $+70\,\%$ was more biased compared to a mean of $+20\,\%$ for annual LAI values reported by Li et al. (2022). Ma et al. (2017) reported a relative bias in GPP of $40\,\%$ on average which is higher than the relative bias found here. For ECLand, we could not find any comparable study reporting the performance of daily LAI or NEE/GPP specifically, neither for dynamic nor static simulations. However, for static ECLand, correlation coefficients between modeled and observed NEE and GPP were in line with those obtained by Boussetta et al. (2013) for 10-day averages at several FLUXNET sites. Also, for the mean annual cycles of NEE and GPP, Stevens et al. (2020) found a lower prediction error (RMSD) when using MODIS LAI forcing compared to default prescribed LAI, and, like in our study, a substantial bias in LAI.

For both models, using *MODIS climatology* in static simulations resulted in the best performances concerning LAI. This agreed with expectations. Since all our simulations were validated with MODIS data, a better performance using static runs using *MODIS climatology* itself would likely yield better results than the default values in either model. For Noah-MP, static simulations with *MODIS climatology* indeed yielded the best performance regarding LAI in some sites, but, interestingly, using the *default climatology* performed also well for others. LAI deviations with the *default climatology* occurred specifically in short vegetation, which was also true for the dynamic runs (see below). For ECLand, where the *default climatology* is already based on MODIS data (Boussetta et al., 2012), the performance of the static run was generally improved compared to the validation dataset, as the higher spatial resolution allowed for a better geographical mapping. Also, ECLand *default climatology* was created by disaggregating the total LAI in the MODIS data to the low and high vegetation type on the grid cell. Both points together can explain the better performance for LAI of static ECLand simulations with *MODIS climatology* compared to *default climatology*.

Dynamic vegetation yielded no better LAI results compared to using static runs with *MODIS climatology* for either model. Evergreen broadleaf forests showed the lowest correlation coefficients for dynamic LAI predictions which was also shown by Yang et al. (2011) for tropical regions simulated by Noah-MP. Additionally, Brunsell et al. (2020) reported overestimation of LAI with dynamic Noah-MP for the Eastern Amazonian Forest which we could not find here. ECland suffered from overall strong relative biases regarding LAI in dynamic simulations. The underestimation of prognostic LAI was already shown by Boussetta et al. (2021). Substantial biases also occurred in Noah-MP (also shown by Huang et al., 2022), but especially so for short or sparse vegetation types. The latter could be due to LAI overestimation in the early growing season as reported by Cai et al. (2014). Also, Liu et al. (2016) found that neither look-up table LAI nor predicted dynamic LAI annual cycles seemed to reproduce LAI observations for short vegetation. On the other hand, Pilotto et al. (2015) achieved satisfactory model predictions for crop sites without vegetation dynamics. Thus, for short vegetation such as grasslands, the Noah-MP Crop module maybe

better represents LAI dynamics (Liu et al., 2016), which should be tested in the future.

The performance of NEE and GPP in ECLand was not very sensitive to different vegetation dynamics. Generally, using static *MODIS climatology* yielded the best predictions of GPP and NEE, although the correlation between modeled and observed NEE was generally low (mean Pearson correlation coefficient was $0.44$). In many sites, even static simulations with *default climatology* resulted in comparable performances. Interestingly, adding more detailed information by using *MODIS single-year* LAI forcing did not further improve model performance (not shown), as we would have expected if LAI dynamics contributed substantially to enhancing model performance for the carbon fluxes. However, other authors found improved model performance of turbulent fluxes, GPP and soil moisture for roughly $50\%$ of their set of sites by updating the LAI forcing using near real-time data assimilation (Boussetta et al., 2015). In other words, a more frequent reset of LAI to the correct value can improve the ECLand performance in general, but did not have an effect for the annual resolution applied here.

Assimilation of LAI during model runs and instead of fixed forcing (as in a static case) also improved LAI and GPP model quality in a study by Xu et al. (2021) using dynamic Noah-MP. We, therefore, expected that LAI dynamics potentially improve model quality regarding carbon fluxes which was predominantly not the case. However, dynamic Noah-MP is already known to overestimate GPP (Ma et al., 2017; Liang et al., 2020; Brunsell et al., 2020). Especially short and sparse vegetation types suffered from low predictive power mainly in NEE correlation (Yang et al., 2021) and in GPP relative bias (Li et al., 2022) which could be also observed here. None of the parameter sets Yang et al. (2021) tested for simulations with dynamic Noah-MP resulted in well-fitting predictions of daily changes in NEE for three of the four sites with short vegetation within ChinaFLUX. Note, however, that LAI, NEE and GPP of short and sparse vegetation were also not well-represented in static runs either. Also, Kumar et al. (2019) could only achieve marginal improvements in GPP representation by dynamic Noah-MP due to LAI assimilation for crops and grasslands which suggests that LAI dynamics had only a limited effect on simulated NEE there. One possible reason for lower predictive power of the models regarding the carbon fluxes of short or sparse vegetation could be that these vegetation types more quickly and dynamically respond to fluctuations in the environment (e.g. soil moisture limitations). Forest ecosystems might be able to compensate restrictions by a larger intrinsic carbon storage or deeper roots resulting in less variability of their productivity within and between the years. It seems that both models investigated here cannot mimic this differentiation. Nonetheless, it was shown here that correlation coefficients for GPP simulated with dynamic Noah-MP were high (also found by Liang et al., 2020; Li et al., 2022) and, at the same time, relative bias was small for all forests except the evergreen broadleaf forests (see section 3.1 and Fig. A1). Thus, although some previous studies found substantial uncertainties in modeled GPP for different vegetation types (Ma et al., 2017; Liang et al., 2020; Li et al., 2022), predicting ecosystem variables using dynamic Noah-MP could be useful at least for forests in studies when LAI climatology cannot be used such as climate change impact studies.

Considering the opposing biases in NEE (and GPP) indicates that the models differ in their estimates of ecosystem respiration. One important difference is the sequence of the calculation of GPP, NEE and respiration. ECLand estimates net assimilation and respiration first separately whereby respiration is set to be $11\%$ of net assimilation and, then, both are used to calculate GPP. In Noah-MP, the first estimate is for GPP which is reduced by respiration to gain a values for NEE and, additionally, respiration is scaled by GPP and available biomass inclusive of LAI. Including our findings, for dynamic ECLand, the underestimated LAI

directly transfers into lower NEE values and, thus, also to GPP since respiration is a fixed fraction of NEE. Apart from the fraction of GPP that is directly needed for metabolism, the estimation of respiration in dynamic Noah-MP also considers leaf maintenance which is another difference compared to ECLand. As a result, respiration is slightly overestimated in ECLand, and slightly underestimated in Noah-MP.

## 4.2 Model performance of turbulent fluxes and soil moisture were almost unaffected by vegetation dynamics in both LSMs

The model performance of ECLand and Noah-MP regarding heat fluxes and soil moisture seems almost insensitive to vegetation dynamics. Correlation, variability and bias of turbulent fluxes in this study were comparable to other studies. While evaluating static ECLand with FLUXNET data, Stevens et al. (2020) found correlation coefficients of $0.79$ and $0.77$ for the annual cycle of latent and sensible heat, respectively, and Boussetta et al. (2013) showed a mean correlation coefficient of $0.81$ for 10-day averages of latent heat. For Noah-MP, statistical measures for turbulent fluxes and soil moisture were mostly in line with other studies (Niu et al., 2011; Ma et al., 2017; Yang et al., 2018; Xu et al., 2021) although Pilotto et al. (2015) presented lower correlation coefficients between $0.20$ and $0.43$. Interestingly, Ma et al. (2017) showed opposing relative bias for evapotranspiration on annual time scale over the continental U.S. of $4\%$ and $22\%$ for static and dynamic simulations, respectively. For ECLand, it had little impact on turbulent fluxes whether vegetation was simulated dynamically instead of statically. Model performance for LE and EF changed only for some sites and towards lower performance (see section 3.2). The predominant underestimation of LE agrees with the findings of Stevens et al. (2020). For dynamic ECLand, the underestimation of GPP and LAI (also in Boussetta et al., 2021) could also be the reason for the poor correlation of EF between modeled and observed values because the energy fraction that is used for transpiration is underestimated. Boussetta et al. (2021) found that dynamic vegetation in ECLand improved numerical weather predictions. There, the main improvements in model performance were achieved through updating land cover maps and the LAI in the look-up table or by including LAI seasonality which both is comparable with our experiment using *MODIS climatology* in static ECLand simulations. Here, we could not confirm theses findings being related to improved performance in heat fluxes since model performance of LE and EF were almost unaffected by the used LAI forcing, which was already experienced by others (Stevens et al., 2020; Nogueira et al., 2021). The reason might be that parameters are adapted to the prior vegetation information (Ruiz-Vásquez et al., 2023) and, thus, the model needs a re-calibration.

Also for Noah-MP, activating vegetation dynamics had mostly little impact on LE and EF predictions. A slight improvement in model performance was found for some sites with short vegetation types or semi-arid climates. Ma et al. (2017) found that using LAI climatology resulted in better model performances for LE than simulations with activated vegetation dynamics for Noah-MP using the monthly FLUXNET Multi-Tree Ensemble data over the U.S.. However, here, we did not find enhanced biases in LE predictions with dynamic Noah-MP compared to the static simulations as they did which could be due to the differing timescales for model evaluation. Both, overestimation and underestimation of LE predicted by dynamic Noah-MP is reported in the literature (Brunsell et al., 2020; Ma et al., 2017; Cai et al., 2014). Brunsell et al. (2020) showed a positive bias of monthly evapotranspiration in the Eastern Amazonian Forest simulated with dynamic Noah-MP while we found a negative bias

of LE for the FLUXNET site *GF-Guy*. For short vegetation types, using the Noah-MP Crop module with activated vegetation dynamics might be sufficient in predicting surface fluxes (Liu et al., 2016). Achieved improvement for LE might be not as large as for sensible heat flux (Liu et al., 2016) which could be a reason for poor performances in EF presented here.

Although vegetation and soil moisture state variables are directly coupled within land-surface models, we found almost no impact of different vegetation modeling on soil moisture predictions for both models. Activating vegetation dynamics or changing LAI forcing did not improve soil moisture representation on average. The reason might be due to the implemented interaction of carbon and water processes. First, the potential photosynthetic activity in dependence of leaf area and radiative conditions is calculated. Then, the limitation factor of extractable water is estimated according to available soil water and roots. Lastly, the
photosynthetic activity is adapted to that restriction and transpiration rate adapted to conductivity and atmospheric conditions. As a result, the only included path is that soil moisture impacts photosynthetic activity and biomass build-up. But there is no feedback that more biomass needs/loses more water that will be taken from the soil because photosynthetic activity in the models relates only to the carbon fluxes but not to the water fluxes.

Additionally, modeled soil moisture suffers from substantial biases in both directions which was also found by Liang et al.
(2020) for Noah-MP and by Garrigues et al. (2021) for ECLand although correlation between observed and modeled soil moisture can be satisfactory (Beck et al., 2021; Xu et al., 2021; Pilotto et al., 2015; Liang et al., 2019). The reason might be underlying default values for soil characteristics such as field capacity and permanent wilting point that possibly deviate from on-site soil conditions and optimal values for soil parameters are still uncertain (Li et al., 2020). Alternatively, it could be an effect of differing scales since the observation from FLUXNET refers to point measurements. The Multiscale parameter
regionalization (MPR) might provide an improved way to estimate soil parameters by applying pedo-transfer function on local soil characteristics and, recently, has been applied to Noah-MP in a proof-of-concept (Schweppe et al., 2022).

Overall, the model performance of soil moisture and heat fluxes was barely affected by vegetation dynamics or applied LAI forcing. However, the sensitivity to LAI might be given since van den Hurk et al. (2003) found some effect of changed LAI values given into TESSEL, a predecessor of ECLand, as well as Ma et al. (2017) and Zhang et al. (2016) did for Noah-MP.
Xu et al. (2021) showed improved LE and soil moisture simulations with more realistic LAI although, also there, the effect was not only site-dependent but also differed with season and year. But those authors also highlighted that transpiration is only partly determined by LAI and other factors controlling the canopy conductance to water vapor might play a larger role. Therefore, other compensating mechanisms may explain low elasticity between LAI and LE or soil moisture (see section 3.2). Yang et al. (2011) demonstrated that the applied runoff scheme more strongly determined model performance of soil moisture
and evapotranspiration than the schemes for dynamic vegetation, stomatal resistance and soil moisture stress. Still, optimizing parameters can be effective in improving model predictions which could be shown by several studies (Bohm et al., 2020; Li et al., 2021, 2020). Even more, the sensitivity of soil moisture to vegetation parameters was shown to enlarge with dynamic vegetation representation (Arsenault et al., 2018). Yet, uncertainty about the optimal values for especially soil and vegetation parameters remains (Li et al., 2020).

Overall, the impact of vegetation dynamics and LAI on turbulent heat fluxes and soil moisture in this investigation was slim across sites and seasons for both models. Thus, modelers who are mainly interested in the performance of carbon processes

should be careful using performance metrics for hydrological variables as a proxy (e.g. LE) because the model formulation for the latter might have controlling processes other than LAI or NEE which dominate the result. Whether applying vegetation dynamics in model simulations is advantageous might depend on the target variables. While, for heat flux predictions, using
*MODIS climatology* might be sufficient, activating vegetation dynamics could play a role for improve carbon flux predictions at seasonal or annual timescale (Jarlan et al., 2008).

### 4.3 Discrepancy between observed and simulated GPP-LAI relationship is caused by model structure

The substantial scatter in the observed relation between GPP and LAI is in close agreement with previous work, showing that GPP also depends on the short-term availability of resources (e.g., light, soil water) (Hu et al., 2022). Additionally, Zhang et al. (2021) found that in LSMs the relation between LAI and GPP was too tight. We therefore checked the underlying relations in the models causing this.

GPP-LAI relationship in Noah-MP showed a clear exponential hysteresis (see section 3.3). This hysteresis is related to the partitioning of GPP to the carbon pools in the plants and to LAI reducing processes as leaf turnover and leaf die-back. Noah-MP uses a non-linear function for allocation of GPP to the leaves that limits the maximum LAI the model can grow resulting in the LAI saturation in summer that can be seen in Figure 8 m-p. On the other hand, leaf turnover due to leaf aging is implemented as a linear function of leaf mass while leaf die-back due to environmental limitations follows exponential functions. Taken together, leaf die-back dominates in the later growing season which results in the hysteresis. The reduction of LAI (i.e., leaf die-back) is implemented to be dependent both on water and temperature stress, but temperature stress is the main driver. In the specific implementation used here, water stress only occurs at a very low soil saturation of $0.1\ vol\%$ for silt loam exemplarily which is even below the permanent wilting point of this soil texture type according to the look-up table value. These values are rarely reached and, thus, water stress is negligible most of the time. In contrast, temperature stress is implemented as an exponential function causing the late growing season non-linear decline of GPP observed throughout the non-tropical sites. Temperature stress is at maximum at $5\ ^\circ C$ for forest ecosystems resulting in no active biomass below this threshold. For this reason, LAI values are almost constant at the tropical forest site because temperature is never limiting there.

ECLand with static vegetation shows a similar pattern of seasonal dynamics as Noah-MP with vegetation dynamics but with less pronounced exponential relationship. In contrast, dynamic ECLand simulates LAI that is strongly coupled to daily meteorological conditions, leading to higher daily fluctuations of LAI than expected, including strong drops of LAI in summer. Three processes govern these daily LAI dynamics: GPP, respiration and senescence. GPP is linearly related to LAI and varies with environmental and meteorological conditions causing the variability in static runs. In dynamic runs, losses in biomass due to high or low daily GPP linearly affect LAI. In other words, unfavourable GPP can reduce LAI almost immediately. The second process affecting LAI is senescence. ECLand distinguishes growing and senescence phases by comparing active biomass due to assimilation with the biomass from the previous time step. If active, then senescence is a linear function of active biomass and a folding-factor. The folding-factor reduces part of the senescent biomass, depending on photosynthesis (reduced in the case of high assimilation) and LAI. Overall, the folding-factor changes only slightly with LAI. Additionally, a reduction of LAI and, thus, active biomass due to reduced GPP (as explained before) causes the model to trigger senescence because the

active biomass of the previous time step was higher. The third process is respiration. About $11\%$ of physiologically possible assimilation is used for dark respiration without considering actual light conditions. This might cause high values of dark respiration compared with possible assimilation based on meteorological conditions and, thus, reduce net primary production, even producing negative values. Notably, no aboveground biomass storage is built up and there is no turnover. Most locations show a linear relationship comparable to ECLand but with a higher variability (Fig. 8 first and third row). This might be due to the fact that leaf growth and leaf fall, in particular for trees, happen on longer timescales than the daily one as implemented in ECLand which inhibits immediate effects of GPP on LAI.

Overall, the current implementations of leaf dynamics in both models use very different approaches to represent LAI dynamics. In Noah-MP it is mainly temperature-driven, and GPP depends little on LAI once the canopy is fully developed. In contrast, in ECLand, LAI and GPP are coupled very tightly and, thus, the LAI dynamics follow almost the same sensitivities to water limitation and radiation as turbulent fluxes, which is unrealistic. Realistic LAI is less dynamic and less sensitive to environmental conditions, as also indicated by the observations. Hence for very different reasons, in both models the performance regarding LAI and turbulent and carbon fluxes is disconnected.

### 4.4 Implications and limitations

For modeling LAI and carbon fluxes, using dynamic vegetation modules in their current implementation in either model is not yet efficient because they increase model complexity encompassing more dynamic processes and parameters without improving the predictive skill. As the dynamic vegetation components in ECLand are still under development, findings from this study can help better understand and represent the processes involved to improve its performance in modeling carbon and energy fluxes. But also for Noah-MP, we showed that the dynamic vegetation module has potential for improvement especially related to the relationship between GPP (and thus also NEE) and LAI. Underlying processes such as carbon allocation, root dynamics, plant hydraulics, feedbacks on photosynthesis and their parameterization can still be worked on (Ma et al., 2017; Li et al., 2021). Overall, we recommend using *MODIS climatology* forcing for static simulations which yielded the best model performances for carbon and water fluxes. This might be valid for other remote sensing LAI products as well but would need to be tested beforehand.

The value of a model evaluation like in this study depends on the reliability of the included datasets. Uncertainty in the forcing data might have a larger impact on the model runs than processes within the models (Zhang et al., 2016), but Haughton et al. (2016) demonstrated that observational errors, in general, are unlikely to cause poor model performance. Nonetheless, model evaluations are also restricted by uncertainty in the reference data (Li et al., 2022) especially when considering flux measurements (Li et al., 2019). We tried to address by carefully inspecting the time series data from FLUXNET2015 before their usage. However, as in all measurements, there are still uncertainties, e.g. from instrumental errors or incomplete energy balance closure.

Also, the MODIS dataset harbors uncertainty originating from cloud coverage, especially in the tropics. We tried to minimize this uncertainty by excluding all days from the dataset that were flagged with significant cloudiness. But saturation also limits the representativeness of the LAI measurements. Even when using only data with the highest possible quality flag, we found

suspiciously low LAI values in summer for temperate forests and grasslands, and especially for tropical forests throughout the year (Fig. 8 c). Noisy and uncertain LAI data from MODIS for tropical forests was already reported among the literature (Weiss et al., 2007; Garrigues et al., 2008; Xiao et al., 2016; Zhang et al., 2024). As a result, reference data remains a source for uncertainty and a deviation in model outputs from it is expected. In any case, reference data is essential for model verification, calibration, and validation but should be treated carefully concerning its reliability and uncertainty.

## 5 Conclusions

Land-surface models often include modules for dynamic vegetation processes. Yet, an evaluation of the representativeness of key variables such as leaf area index or net ecosystem exchange is rarely done on high temporal resolution. The impact of different parameterization of vegetation processes on water and carbon flux estimates by land-surface models is still poorly understood. Therefore, we evaluated the change in model performance of ecohydrological target variables when dynamic veg-
605 etation processes are included for two land-surface models and further gained insight into critical process implementations that lead to the observed patterns.

Surprisingly, neither for ECLand nor for Noah-MP, including modules for dynamic vegetation in their implementation improved the model predictions of ecohydrological variables. We expected vegetation dynamics in these land-surface models to better capture the higher variability in ecosystem exchange, especially that of highly dynamic short or sparse vegetation types,
but this was predominantly not the case. Using alternative input for leaf area index other than default climatology also had a negligible effect on the model performance but this needs to be evaluated in more detail since we were limited in data sources. Moreover, model performances of carbon and hydrological fluxes appeared to be weakly coupled. Therefore, the question arose whether exchange fluxes themselves in these land surface models are sensitive to changes in leaf area index estimates and not only to changing parameter sets. Indeed, different leaf area index estimates lead to different predictions in exchange fluxes
but without affecting the overall model performance of these variables. This might be caused by the mismatch in the seasonal patterns between observations and models for the relationship of gross primary productivity and leaf area index. While this relationship in dynamic Noah-MP showed a logarithmic hysteresis, mainly driven by temperature, both variables are tightly linearly coupled in dynamic ECLand without allowing for the leaf area index to remain unchanged in suboptimal conditions for photosynthesis.
This deeper analysis of the model performance for ecohydrological fluxes that pinpoints to the reasons for model behavior was only possible with a reduced number of models. We used specific setups for the two land surface models evaluated here. Adapting or changing parameters and investigating the effect of other processes within the models were beyond the scope of this study. At this point, it remains unclear how representative our model selection is for the performance and process evaluation of other land surface models since they have processes implemented differently. Nonetheless, we highlighted some crucial
relationships in the implementation of vegetation processes that have the potential for further improvement. Additionally, they might be a good starting point for a similar intensive investigation with other land surface models or other alternative LAI climatology.

*Code and data availability.* Observational data from the FLUXNET2015 dataset were accessed via FLUXNET data portal (fluxnet.org, 2020). Observational data for TERENO observatory "Hohes Holz" can be found at PANGAEA (Rebmann and Pohl, 2023). IGBP Land
Classification is published by National Center for Atmospheric Research (2022). Aridity index was taken from Trabucco and Zomer (2018). Gap-filling of meteorological data was done by using ERA5 re-analysis product (Hersbach et al., 2020), accessed by Climate Data Store API (Copernicus, 2018). USGS vegetation types can be found at University Corporation for Atmospheric Research (2023). Global gridded soil information (Hengl et al., 2014) is available at https://soilgrids.org. MODIS Leaf area index was retrieved via Earth Data Portal from NASA (Myneni et al., 2015).

 **Appendix**

## A.01 Dynamic ECLand processes

For more details, see the published model descriptions (Boussetta et al., 2012, 2013, 2021). Photosynthesis model is based on Calvet et al. (1998). Therein, potential net assimilation $A_n$ is estimated from physiological constrains as

$$A_n = A_{max} \cdot (1 - e^{-\frac{g_{meso} \cdot (c_i - c_{comp})}{A_{max}}}) \tag{A.1}$$

where $A_{max}$ is the leaf photosynthetic capacity, $g_{meso}$ is the mesophyll conductance, $c_i$ is the leaf-internal $CO_2$ concentration and $c_{comp}$ is the $CO_2$ compensation point. Potential gross assimilation $A_g$, then, is calculated as

$$A_g = (A_n + R_d) \cdot \epsilon \tag{A.2}$$

where $R_d$ is the dark respiration from

$$R_d = A_n \cdot f_R \tag{A.3}$$

with $f_R = \frac{1}{9}$ as dark respiration factor, and where $\epsilon$ is a quantum use efficiency factor, estimated as

$$\epsilon = 1 - e^{-\frac{\epsilon_0 \cdot E_{PAR}}{A_n + R_d}} \tag{A.4}$$

where $\epsilon_0$ is the maximum quantum use efficiency and $E_{PAR}$ is the absorbed photosynthetic active radiation. Actual gross assimilation $GPP$ results from

$$GPP = A_g \cdot LAI \cdot \rho_a \tag{A.5}$$

where $LAI$ is the leaf area index of the prior time step and $\rho_a$ is the air density corrected for humidity.

$A_n$ is used as the maximum leaf assimilation for the senescence model (Calvet and Soussana, 2001). To avoid immediate leaf die-back, a damping factor for senescence $f_s$ is introduced as

$$f_s = \max(\frac{\tau_{lim} \cdot t_s}{100 \cdot N_{day}}, \max(10^{-8}, \frac{t_s}{N_{day}} \cdot \min(1, \frac{A_n}{A_{max}}) \cdot \frac{\max((r_{meso} \cdot 1000)^{0.321} \cdot LAI}{f_{LAI}}, 1))) \tag{A.6}$$

where $\tau_{lim}$ is a limiting factor for immediate biomass loss, $t_s$ is the damping time for senescence, which basically is the amount of seconds per year, $N_{day}$ is the amount of seconds per day, $A_{max}$ is the maximum photosynthesis rate with optimal conditions and $f_{LAI}$ is a LAI correction parameter that reduces mortality at high LAI values which would occur due to shadowing. The amount of biomass loss $B_{loss}$ then is

$$B_{loss} = \min(B - LAI_{min} \cdot f_{LAI-B}, B \cdot (1 - e^{-\frac{1}{f_s}})) \tag{A.7}$$

where $B$ is the biomass of the prior time step and $f_{LAI-B}$ is a conversion factor between $LAI$ and $B$. Then, biomass $B$ is updated by subtracting $B_{loss}$. The change in biomass due to assimilation $B_{gain}$ results from

$$B_{gain} = \max(LAI_{min} \cdot f_{LAI-B} - B, A_n \cdot f_{Cbiom}) \tag{A.8}$$

where $f_{Cbiom} \approx 0.68$ is a factor converting the amount of $CO_2$ uptake from assimilation to carbon in dry biomass. Biomass $B$ is updated again by adding $B_{gain}$. In the end, this updated biomass is transferred into an updated LAI value by

$$LAI = \frac{B}{f_{LAI-B}} \tag{A.9}$$

LAI determines the interception reservoir $W$ by

$$W = W_{max} \cdot (c_B + c_H \cdot LAI_H + c_L \cdot LAI_L) \tag{A.10}$$

where $W_{max}$ is the maximum thickness of the water layer on leafs or bare ground, $c_B$, $c_H$ and $c_L$ are the fractions for bare soil, high vegetation and low vegetation on a grid cell and $LAI_H$ and $LAI_L$ are the LAI values for high and low vegetation, respectively (Boussetta et al., 2012). Additionally, canopy resistance $r_c$ depends on LAI via

$$r_c = f_1 f_2 f_3 \cdot \frac{r_{s,min}}{LAI} \tag{A.11}$$

where $r_{s,min}$ is the minimum stomatal resistance and $f_n$ are the restriction factors for low input in shortwave radiation, soil moisture stress and saturated atmospheric conditions (Boussetta et al., 2012).

### A.02 Dynamic Noah-MP processes

For more details, see the published model descriptions (Niu et al., 2011; Ma et al., 2017; Oleson et al., 2012). The model for leaf dynamics within Noah-MP is based on Dickinson et al. (1998). Leaf biomass $C_{leaf}$ is balanced over time with

$$\frac{\delta C_{leaf}}{\delta t} = f_{leaf} \cdot A_{tot} - (d_{stress} + d_{turnover} + R_{leaf}) \cdot C_{leaf} \tag{A.12}$$

where $A_{tot}$ is the total carbon assimilation rate, $f_{leaf}$ is the fraction of allocation to the leaves, $d_{stress}$ is the dying rate caused by cold and drought stress, $d_{turnover}$ is the turnover rate due to senescence, herbivory or mechanical loss as a vegetation-type dependent parameter and $R_{leaf}$ is the respiration rate of the leaf biomass. $f_{leaf}$ is determined by LAI via

$$f_{leaf} = e^{0.01 \cdot LAI(1 - e^{\chi \cdot LAI})} \tag{A.13}$$

where $\chi = 0.75$ is a parameter defining the partitioning of carbon allocation between leaves and stem. $A_{tot}$ is split up to photosynthesis rates from sunlit and shaded leaves, respectively:

$$A_{tot} = 12 \cdot 10^{-6} \cdot (A_{sunlit} \cdot LAI_{sunlit} + A_{shaded} \cdot LAI_{shaded}) \tag{A.14}$$

where the first factor is for unit conversion. The partitioning of sunlit and shaded LAI results from a two-stream radiation transfer scheme (Niu et al., 2011). Assimilation rate for sunlit and shaded leaves, respectively, is estimated with a bottle-neck

principle as

$$A = I_g \min(A_L, A_C, A_S) \tag{A.15}$$

where $I_g$ is a growing season index according to leaf temperature and $A_L$, $A_C$, $A_S$ are the photosynthesis rates limited by light, Rubisco and export, respectively (Bonan, 1996). $A_L$ results from

$$A_L = \frac{4.6 \cdot \epsilon \cdot E_{PAR}(c_i - c_{comp})}{c_i + 2c_{comp}} \tag{A.16}$$

with $c_i$ being the leaf-internal $CO_2$ concentration, $c_{comp}$ being the $CO_2$ compensation point, $\epsilon$ being the quantum use efficiency and $E_{PAR}$ being the absorbed photosynthetic active radiation. Additionally, $A_S = 0.5 \cdot V_{max}$ and

$$A_C = \frac{V_{max}(c_i - c_{comp})}{c_i + K_c(1 + \frac{c_o}{K_o})} \tag{A.17}$$

where $c_o$ is the atmospheric $O_2$ concentration, $K_c$ and $K_o$ are the Michaelis-Menton constants for $CO_2$ and $O_2$ (Collatz et al., 1991), respectively, and $V_{max}$ is the maximum carboxylation rate, defined by

$$V_{max} = V_{max,25} \cdot \alpha_{max}^{\frac{T_v - 25}{10}} \cdot f_N f_{T_v} \beta \tag{A.18}$$

where $V_{max,25}$ is the maximum carboxylation rate at 25 °C, $\alpha_{max}$ is a temperature conversion factor, $T_v$ is the vegetation temperature, $f_N$ is a factor for nitrogen limitation of the leaves, $f_{T_v}$ is a factor for temperature limitation (Collatz et al., 1991) and $\beta$ represents the limitation by available soil moisture.

$d_{stress}$ for the leaf mass balance is estimated from

$$d_{stress} = d_{cold} \cdot e^{-0.3 \cdot \max(0, T_v - T_{min})} \frac{C_{leaf}}{120} + d_{dry} \cdot e^{-100\beta} \tag{A.19}$$

where $T_{min}$ is a vegetation type dependent threshold temperature for leaf survival, $\beta$ is the soil moisture limitation factor and $d_{cold}$ and $d_{dry}$ are vegetation type dependent dying rates (prescribed parameter) for temperature and dryness stress, respectively. Leaf respiration $R_{leaf}$ is calculated with

$$R_{leaf} = f_{res}((f_{leaf} - \frac{LAI}{\chi \cdot f_{leaf}}) \cdot A_{tot} - R_l \tag{A.20}$$

where $f_{res}$ is a factor defining the fraction of assimilation that is used for respiration and $R_l$ is the respiration for leaf maintenance from

$$R_l = \min(\frac{C_{leaf} - C_{leaf,min}}{\Delta t}, 0.5 \cdot 12 \cdot 10^{-6} \cdot r_l(T_v) \cdot LAI \cdot \beta \cdot \frac{c_N}{c_{N,max}}) \tag{A.21}$$

where $C_{leaf,min}$ is the minimum leaf biomass, $\Delta t$ is the time step duration, 0.5 is a reduction factor for respiration during non-growing season, $r_l(T_v)$ is the vegetation type dependent respiration rate for leaf maintenance at $T_v$ and $\frac{c_N}{c_{N,max}}$ is the nitrogen saturation within the leaves. Afterwards, net primary production $NPP$ is estimated as

$$NPP = (f_{leaf} - \frac{LAI}{\chi \cdot f_{leaf}}) \cdot A_{tot} - R_{leaf} - R_l \tag{A.22}$$

$GPP$ is set to $A_{tot}$ and $LAI$ is updated with

$$LAI = C_{leaf} \cdot f_{LAI-B} \tag{A.23}$$

where $f_{LAI-B}$ is the leaf area per biomass.

Assimilation rate $A$ determines the stomatal resistance $r_s$ by

$$\frac{1}{r_s} = g_{min} + \frac{m \cdot p_{air} \cdot A}{c_{air}} \frac{e_{air}}{e_{sat}(T_v)} \tag{A.24}$$

where $g_{min}$ is the minimum stomatal conductance, $m$ is an empirical parameter for the relationship between transpiration and $CO_2$ flux, $p_{air}$ is the surface air pressure, $c_{air}$ is the $CO_2$ concentration at leaf surface, $e_{air}$ is the vapor pressure at leaf surface

and $e_{sat}(T_v)$ is the saturation vapor pressure inside the leaves (Ball et al., 1987; Bonan, 1996). $r_s$ then is used to estimate latent heat flux and, thus, evapotranspiration.

# A.03  Additional figures

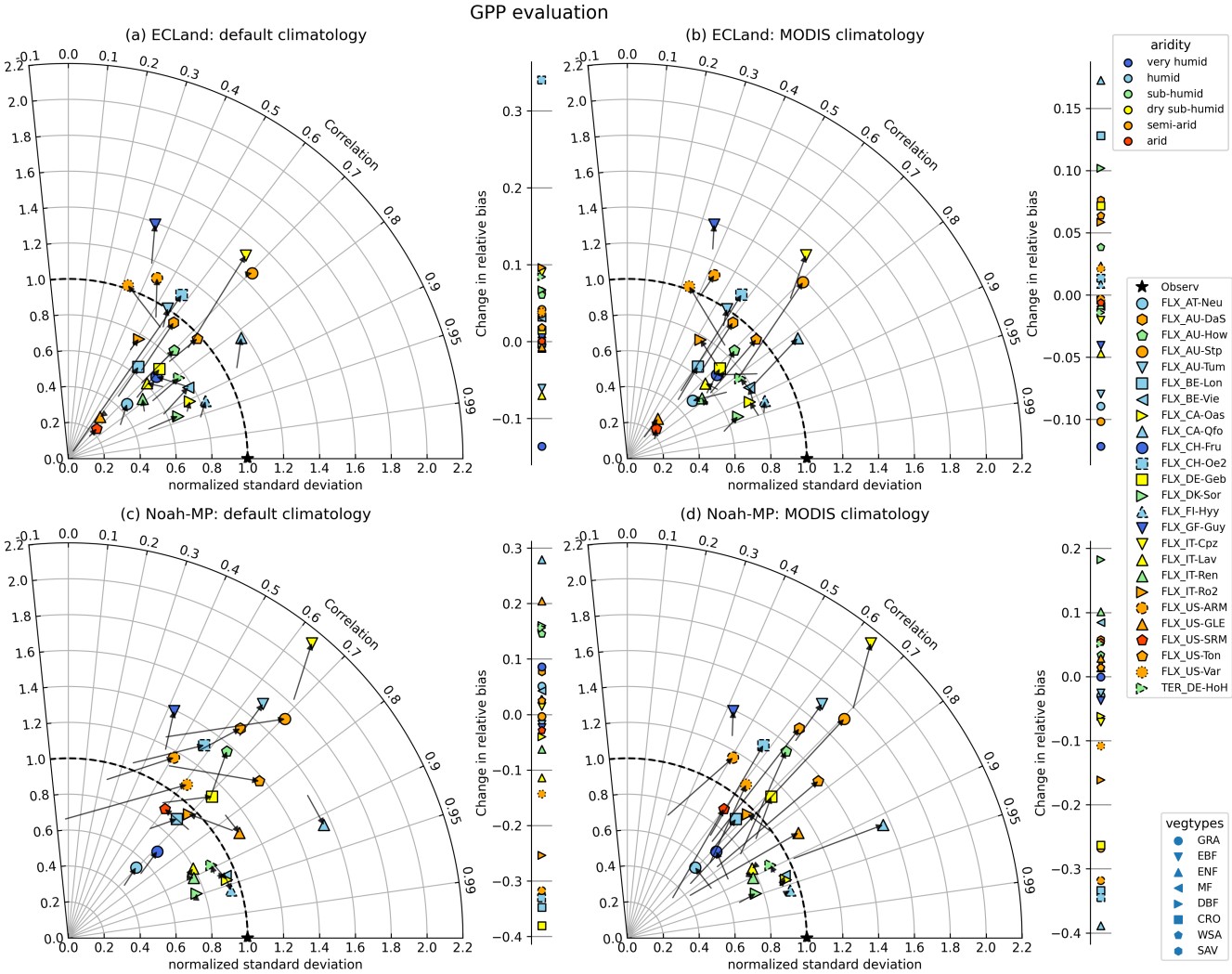

**Figure A1.** Change of model quality metrics for GPP prediction when switching on dynamic vegetation for all included sites and by using *default climatology* (left) or MODIS climatology (right). The star ("Observ") marks the location of the perfect correlation between observation and model and perfect agreement between observed and modelled variance. The model performance of the static runs can be read from the start of each arrow. The point colors indicate the site aridity (top right legend). Vegetation types are symbolized by different marker types (bottom right legend).

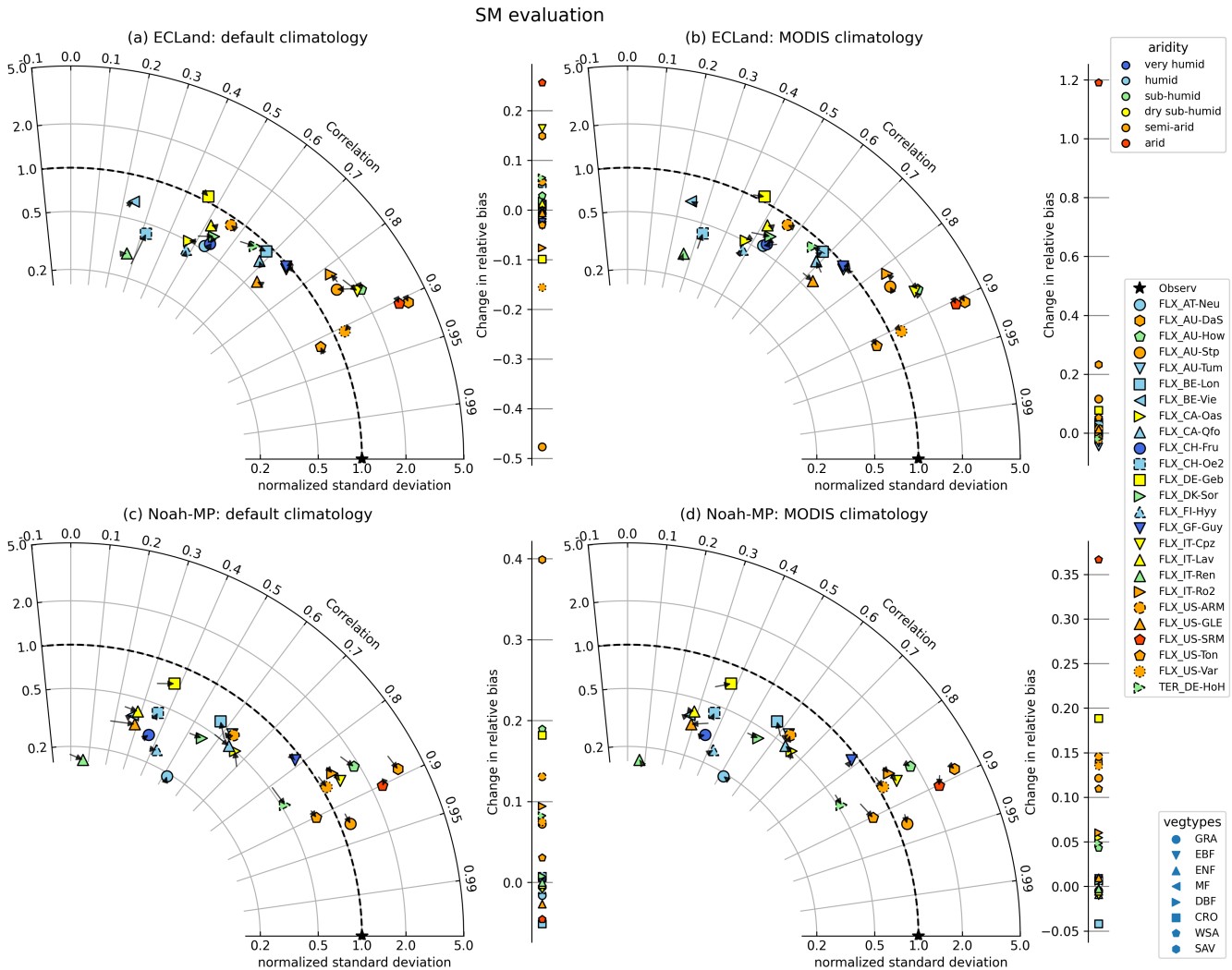

**Figure A2.** Same as before but for soil moisture.

## A.04 Performance metrics tables

**Table A1.** Relative bias for LAI. The word climatology is shortened with clim.. MODIS single refers to the MODIS single-year setup.

| ECLand | | | | | | Location | Noah-MP | | | | | |
| default clim. | | MODIS clim. | | MODIS single | | | default clim. | | MODIS clim. | | MODIS single | |
| static | dynamic | static | dynamic | static | dynamic | | static | dynamic | static | dynamic | static | dynamic |
| --- | --- | --- | --- | --- | --- | --- | --- | --- | --- | --- | --- | --- |
| -68% | -77% | -28% | -73% | -31% | -73% | AT-Neu | -54% | -14% | 5% | -14% | 0% | -14% |
| 0% | 22% | -24% | 20% | -42% | 19% | AU-DaS | -54% | 51% | -9% | 51% | -3% | 51% |
| -7% | 0% | -17% | -1% | -17% | -1% | AU-How | -75% | 27% | -3% | 27% | -1% | 27% |
| 153% | -24% | -51% | -36% | -50% | -36% | AU-Stp | 176% | 231% | -4% | 231% | -5% | 231% |
| -31% | -74% | -1% | -72% | 0% | -71% | AU-Tum | 0% | -5% | 0% | -5% | 1% | -5% |
| 29% | 51% | -7% | 48% | -10% | 47% | BE-Lon | -28% | 110% | 3% | 110% | 0% | 110% |
| -26% | -19% | -16% | -18% | -15% | -18% | BE-Vie | 3% | 9% | -6% | 9% | -6% | 9% |
| -4% | -10% | -12% | -11% | -14% | -11% | CA-Oas | -3% | -7% | -3% | -7% | -5% | -10% |
| 51% | 52% | -17% | 47% | -17% | 47% | CA-Qfo | 200% | 92% | -7% | 92% | -7% | 92% |
| -19% | -67% | -35% | -67% | -34% | -67% | CH-Fru | -44% | -3% | -5% | -3% | -4% | -3% |
| -95% | 141% | -10% | 148% | -10% | 148% | CH-Oe2 | 6% | 146% | 0% | 146% | 0% | 146% |
| 56% | 30% | -8% | 26% | -11% | 26% | DE-Geb | -20% | 99% | 3% | 99% | -1% | 99% |
| -52% | -32% | -20% | -30% | -10% | -29% | DE-HoH | -45% | -16% | -9% | -16% | 0% | -16% |
| 21% | 44% | -11% | 42% | -11% | 42% | DK-Sor | 13% | 62% | -1% | 62% | -1% | 61% |
| -8% | -9% | -16% | -10% | -15% | -10% | FI-Hyy | 62% | -11% | -6% | -11% | -5% | -10% |
| -5% | -16% | 4% | -16% | 1% | -16% | GF-Guy | -7% | 4% | 5% | 4% | 3% | 4% |
| -32% | -9% | 5% | -7% | -1% | -7% | IT-Cpz | 57% | 21% | 5% | 21% | 0% | 20% |
| 32% | -22% | 0% | -24% | -3% | -24% | IT-Lav | 109% | 14% | 12% | 14% | 8% | 14% |
| 2% | -9% | -15% | -9% | -14% | -9% | IT-Ren | 129% | 29% | -5% | 29% | -4% | 29% |
| 78% | -1% | -13% | -7% | -14% | -8% | IT-Ro2 | -38% | 22% | 0% | 22% | -2% | 22% |
| -9% | -51% | -36% | -53% | -36% | -53% | US-ARM | 12% | 144% | 1% | 144% | -3% | 144% |
| 266% | 152% | -14% | 135% | -16% | 135% | US-GLE | 872% | 348% | -3% | 348% | -5% | 348% |
| 182% | 231% | -13% | 221% | -14% | 220% | US-SRM | 365% | 186% | -6% | 186% | -1% | 186% |
| 149% | 87% | -15% | 77% | -14% | 77% | US-Ton | 38% | 93% | -3% | 93% | 0% | 93% |
| 18% | -57% | -40% | -61% | -40% | -61% | US-Var | -2% | 100% | -2% | 100% | -1% | 100% |

**Table A2.** Relative bias for NEE. Note that a positive bias in NEE means an underestimation of carbon uptake by the ecosystem. Abbreviations in the headings are as before.

| ECLand | | | | | | Location | Noah-MP | | | | | |
|---|---|---|---|---|---|---|---|---|---|---|---|---|
| default clim. | | MODIS clim. | | MODIS single | | | default clim. | | MODIS clim. | | MODIS single | |
| static | dynamic | static | dynamic | static | dynamic | | static | dynamic | static | dynamic | static | dynamic |
| -21% | -20% | -34% | -22% | -33% | -22% | AT-Neu | -19% | -23% | -25% | -23% | -24% | -23% |
| 67% | 62% | 69% | 62% | 72% | 62% | AU-DaS | -9% | -6% | -11% | -6% | -11% | -6% |
| 70% | 65% | 68% | 65% | 68% | 65% | AU-How | 8% | 14% | 0% | 14% | 1% | 14% |
| 57% | 59% | 65% | 60% | 65% | 60% | AU-Stp | 22% | 2% | 22% | 2% | 22% | 2% |
| -2% | 2% | -3% | 2% | -3% | 2% | AU-Tum | -34% | -32% | -34% | -32% | -33% | -32% |
| -6% | -6% | -3% | -6% | -3% | -6% | BE-Lon | -12% | -17% | -16% | -17% | -16% | -17% |
| 18% | 16% | 15% | 16% | 15% | 16% | BE-Vie | -6% | -6% | -5% | -6% | -5% | -7% |
| -6% | -6% | -5% | -5% | -5% | -5% | CA-Oas | -23% | -9% | -22% | -9% | -21% | -10% |
| -1% | -2% | 10% | -1% | 11% | -1% | CA-Qfo | -32% | -31% | -22% | -31% | -22% | -31% |
| -17% | -6% | -15% | -6% | -15% | -6% | CH-Fru | -2% | -6% | -6% | -6% | -6% | -6% |
| 5% | -20% | -9% | -20% | -9% | -20% | CH-Oe2 | -20% | -26% | -21% | -26% | -21% | -26% |
| -7% | -7% | -4% | -7% | -4% | -7% | DE-Geb | -15% | -18% | -19% | -18% | -18% | -18% |
| 7% | 3% | 3% | 3% | 2% | 3% | DE-HoH | -16% | -6% | -21% | -6% | -21% | -6% |
| -1% | -4% | 0% | -4% | 0% | -4% | DK-Sor | -20% | -7% | -19% | -7% | -19% | -9% |
| 5% | 6% | 6% | 6% | 6% | 6% | FI-Hyy | -3% | -4% | -6% | -4% | -6% | -6% |
| 13% | 14% | 13% | 14% | 14% | 14% | GF-Guy | -62% | -54% | -62% | -54% | -61% | -54% |
| 15% | 10% | 7% | 9% | 8% | 9% | IT-Cpz | -40% | -30% | -36% | -30% | -34% | -33% |
| 45% | 49% | 46% | 49% | 47% | 49% | IT-Lav | 30% | 32% | 32% | 32% | 33% | 32% |
| 20% | 20% | 20% | 20% | 20% | 20% | IT-Ren | 0% | 5% | 7% | 5% | 7% | 5% |
| 14% | 19% | 22% | 19% | 22% | 20% | IT-Ro2 | -9% | 3% | -18% | 3% | -17% | 3% |
| -3% | -1% | -1% | -1% | -2% | -1% | US-ARM | -2% | -9% | -2% | -9% | -3% | -9% |
| -1% | 0% | 2% | 0% | 2% | 0% | US-GLE | 27% | -19% | -7% | -19% | -7% | -19% |
| 60% | 59% | 59% | 59% | 59% | 59% | US-SRM | -8% | -6% | -3% | -6% | -4% | -6% |
| 10% | 8% | 15% | 9% | 15% | 9% | US-Ton | -7% | -9% | -8% | -9% | -9% | -9% |
| -11% | -7% | -9% | -7% | -9% | -7% | US-Var | -4% | -13% | -9% | -13% | -9% | -13% |

**Table A3.** Relative bias for GPP. Abbreviations in the headings are as before.

| ECLand | | | | | | Location | Noah-MP | | | | | |
| --- | --- | --- | --- | --- | --- | --- | --- | --- | --- | --- | --- | --- |
| default clim. | | MODIS clim. | | MODIS single | | | default clim. | | MODIS clim. | | MODIS single | |
| static | dynamic | static | dynamic | static | dynamic | | static | dynamic | static | dynamic | static | dynamic |
| -25% | -25% | -15% | -23% | -15% | -24% | AT-Neu | -25% | -20% | -17% | -20% | -17% | -20% |
| -38% | -31% | -39% | -31% | -42% | -31% | AU-DaS | -24% | -16% | -22% | -16% | -21% | -16% |
| -44% | -38% | -41% | -38% | -42% | -38% | AU-How | -33% | -18% | -22% | -18% | -22% | -18% |
| 16% | 12% | 0% | 10% | 0% | 10% | AU-Stp | 30% | 30% | -3% | 30% | -4% | 30% |
| -8% | -14% | -6% | -14% | -6% | -14% | AU-Tum | 21% | 23% | 21% | 23% | 20% | 23% |
| -8% | -5% | -18% | -6% | -19% | -6% | BE-Lon | -7% | 42% | 9% | 42% | 7% | 42% |
| -27% | -24% | -23% | -24% | -24% | -24% | BE-Vie | -7% | -3% | -11% | -3% | -11% | -3% |
| -13% | -14% | -13% | -14% | -15% | -14% | CA-Oas | 4% | 8% | 2% | 8% | 0% | 5% |
| -4% | -3% | -21% | -4% | -22% | -4% | CA-Qfo | 69% | 41% | 2% | 41% | 1% | 41% |
| -15% | -29% | -17% | -29% | -17% | -29% | CH-Fru | -32% | -23% | -23% | -23% | -23% | -23% |
| -49% | 15% | -17% | 16% | -17% | 16% | CH-Oe2 | -2% | 35% | -1% | 35% | -1% | 35% |
| -6% | -5% | -12% | -5% | -13% | -5% | DE-Geb | 4% | 42% | 16% | 42% | 13% | 42% |
| -25% | -17% | -15% | -16% | -13% | -16% | DE-HoH | -16% | 0% | -5% | 0% | -4% | 0% |
| -34% | -27% | -38% | -27% | -38% | -27% | DK-Sor | -27% | -11% | -29% | -11% | -29% | -11% |
| -25% | -25% | -26% | -25% | -27% | -25% | FI-Hyy | 10% | -8% | -10% | -8% | -10% | -8% |
| -12% | -12% | -8% | -12% | -10% | -12% | GF-Guy | -2% | 4% | 0% | 4% | -1% | 4% |
| -22% | -13% | -10% | -12% | -12% | -12% | IT-Cpz | 16% | 14% | 7% | 14% | 5% | 13% |
| -51% | -58% | -53% | -58% | -54% | -58% | IT-Lav | -17% | -29% | -30% | -29% | -32% | -29% |
| -39% | -40% | -39% | -40% | -39% | -40% | IT-Ren | 4% | -11% | -21% | -11% | -21% | -11% |
| 11% | -1% | -9% | -3% | -9% | -3% | IT-Ro2 | -5% | 30% | 14% | 30% | 14% | 30% |
| 12% | 9% | 8% | 8% | 10% | 8% | US-ARM | 7% | 38% | 7% | 38% | 7% | 38% |
| -22% | -23% | -25% | -23% | -25% | -23% | US-GLE | 33% | 13% | -15% | 13% | -16% | 13% |
| -23% | -23% | -22% | -23% | -22% | -23% | US-SRM | 2% | 5% | -11% | 5% | -10% | 5% |
| -7% | -5% | -11% | -5% | -11% | -5% | US-Ton | -6% | 3% | -5% | 3% | -5% | 3% |
| 12% | 8% | 9% | 7% | 9% | 7% | US-Var | -1% | 16% | 5% | 16% | 5% | 16% |

**Table A4.** Relative bias for latent heat flux. Abbreviations in the headings are as before.

| ECLand | | | | | | Location | Noah-MP | | | | | |
|---|---|---|---|---|---|---|---|---|---|---|---|---|
| default clim. | | MODIS clim. | | MODIS single | | | default clim. | | MODIS clim. | | MODIS single | |
| static | dynamic | static | dynamic | static | dynamic | | static | dynamic | static | dynamic | static | dynamic |
| -27% | -31% | -13% | -28% | -13% | -28% | AT-Neu | -25% | -19% | -21% | -19% | -22% | -19% |
| -27% | -18% | -30% | -18% | -37% | -19% | AU-DaS | -14% | -13% | -12% | -13% | -12% | -13% |
| -53% | -49% | -51% | -48% | -51% | -48% | AU-How | -43% | -43% | -40% | -43% | -40% | -43% |
| 2% | -1% | -3% | -1% | -3% | -1% | AU-Stp | 8% | -1% | -3% | -1% | -4% | -1% |
| -11% | -17% | -7% | -17% | -7% | -17% | AU-Tum | 5% | 7% | 4% | 7% | 4% | 7% |
| -16% | -14% | -22% | -14% | -24% | -14% | BE-Lon | -3% | 0% | -3% | 0% | -4% | 0% |
| -27% | -20% | -19% | -20% | -20% | -20% | BE-Vie | -5% | 1% | -4% | 1% | -4% | 1% |
| -9% | -9% | -9% | -9% | -10% | -10% | CA-Oas | 3% | 8% | 4% | 8% | 3% | 7% |
| -22% | -21% | -35% | -22% | -36% | -22% | CA-Qfo | -36% | -36% | -38% | -36% | -38% | -36% |
| -21% | -28% | -22% | -29% | -22% | -29% | CH-Fru | -30% | -25% | -29% | -25% | -29% | -25% |
| -65% | -1% | -25% | 0% | -25% | 0% | CH-Oe2 | 3% | 9% | 2% | 9% | 2% | 9% |
| 8% | 3% | -5% | 2% | -8% | 2% | DE-Geb | -2% | 5% | -1% | 5% | -1% | 5% |
| -41% | -33% | -30% | -32% | -29% | -32% | DE-HoH | -12% | -8% | -11% | -8% | -10% | -8% |
| -37% | -29% | -36% | -29% | -37% | -29% | DK-Sor | -19% | -12% | -21% | -12% | -21% | -12% |
| -22% | -21% | -22% | -21% | -22% | -21% | FI-Hyy | -34% | -32% | -35% | -32% | -35% | -32% |
| -31% | -36% | -26% | -35% | -30% | -36% | GF-Guy | -17% | -15% | -17% | -15% | -16% | -15% |
| -21% | -8% | -1% | -7% | -3% | -7% | IT-Cpz | 17% | 16% | 16% | 16% | 14% | 14% |
| -40% | -46% | -42% | -47% | -43% | -47% | IT-Lav | -50% | -47% | -50% | -47% | -50% | -47% |
| -41% | -41% | -41% | -41% | -41% | -41% | IT-Ren | -43% | -41% | -43% | -41% | -43% | -41% |
| -30% | -38% | -39% | -39% | -39% | -39% | IT-Ro2 | -9% | -6% | -8% | -6% | -8% | -6% |
| 6% | 4% | 7% | 4% | 6% | 4% | US-ARM | 6% | 11% | 7% | 11% | 6% | 11% |
| -62% | -64% | -68% | -64% | -68% | -64% | US-GLE | -54% | -63% | -73% | -63% | -73% | -63% |
| -22% | -21% | -27% | -21% | -26% | -22% | US-SRM | -20% | -19% | -21% | -19% | -21% | -19% |
| -5% | -12% | -19% | -13% | -19% | -13% | US-Ton | -8% | -7% | -10% | -7% | -10% | -7% |
| 29% | 28% | 29% | 27% | 28% | 27% | US-Var | 21% | 23% | 17% | 23% | 17% | 22% |

**Table A5.** Relative bias for evaporative fraction. Abbreviations in the headings are as before.

| ECLand | | | | | | Location | Noah-MP | | | | | |
| --- | --- | --- | --- | --- | --- | --- | --- | --- | --- | --- | --- | --- |
| default clim. | | MODIS clim. | | MODIS single | | | default clim. | | MODIS clim. | | MODIS single | |
| static | dynamic | static | dynamic | static | dynamic | | static | dynamic | static | dynamic | static | dynamic |
| -34% | -39% | -21% | -37% | -22% | -37% | AT-Neu | -12% | -10% | -9% | -10% | -9% | -10% |
| -43% | -36% | -47% | -37% | -53% | -37% | AU-DaS | -40% | -34% | -35% | -34% | -35% | -34% |
| -49% | -45% | -47% | -45% | -47% | -45% | AU-How | -43% | -40% | -38% | -40% | -38% | -40% |
| -14% | -18% | -16% | -17% | -17% | -17% | AU-Stp | -1% | -10% | -10% | -10% | -11% | -10% |
| -50% | -60% | -45% | -60% | -45% | -60% | AU-Tum | -26% | -23% | -27% | -23% | -27% | -23% |
| -4% | -5% | -10% | -6% | -11% | -6% | BE-Lon | 8% | 10% | 9% | 10% | 9% | 10% |
| -12% | -10% | -10% | -10% | -10% | -10% | BE-Vie | 0% | 3% | 0% | 3% | 0% | 3% |
| -4% | -5% | -5% | -5% | -5% | -5% | CA-Oas | -4% | -1% | -4% | -1% | -4% | -1% |
| -21% | -21% | -32% | -22% | -33% | -22% | CA-Qfo | -33% | -26% | -31% | -26% | -31% | -26% |
| -8% | -12% | -9% | -12% | -9% | -12% | CH-Fru | -10% | -9% | -10% | -9% | -10% | -9% |
| -32% | -30% | -18% | -29% | -18% | -29% | CH-Oe2 | -4% | -4% | -5% | -4% | -5% | -4% |
| 10% | 4% | 0% | 3% | -2% | 3% | DE-Geb | 8% | 13% | 11% | 13% | 10% | 13% |
| -52% | -46% | -45% | -45% | -43% | -45% | DE-HoH | -27% | -21% | -26% | -21% | -25% | -21% |
| -14% | -11% | -15% | -11% | -15% | -11% | DK-Sor | -9% | -4% | -10% | -4% | -10% | -4% |
| -5% | -6% | -6% | -6% | -6% | -6% | FI-Hyy | -11% | -10% | -12% | -10% | -12% | -10% |
| -73% | -78% | -63% | -78% | -67% | -78% | GF-Guy | -48% | -42% | -47% | -42% | -47% | -42% |
| -12% | 0% | 9% | 1% | 7% | 1% | IT-Cpz | 34% | 34% | 33% | 34% | 32% | 33% |
| -25% | -31% | -28% | -32% | -29% | -32% | IT-Lav | -33% | -30% | -34% | -30% | -34% | -30% |
| -7% | -8% | -8% | -8% | -8% | -8% | IT-Ren | -13% | -12% | -13% | -12% | -13% | -12% |
| -7% | -17% | -22% | -18% | -22% | -18% | IT-Ro2 | 6% | 16% | 11% | 16% | 11% | 16% |
| -7% | -9% | -7% | -9% | -8% | -9% | US-ARM | 1% | 2% | 4% | 2% | 4% | 2% |
| -41% | -44% | -50% | -45% | -50% | -45% | US-GLE | -29% | -26% | -31% | -26% | -31% | -26% |
| -8% | -7% | -11% | -7% | -11% | -7% | US-SRM | -15% | -9% | -13% | -9% | -13% | -9% |
| 5% | 0% | -6% | 0% | -6% | 0% | US-Ton | 1% | 3% | 3% | 3% | 3% | 3% |
| 8% | 2% | 7% | 2% | 6% | 2% | US-Var | 9% | 9% | 15% | 9% | 15% | 9% |

**Table A6.** Relative bias of soil moisture. Abbreviations in the headings are as before.

| ECLand | | | | | | Location | Noah-MP | | | | | |
|---|---|---|---|---|---|---|---|---|---|---|---|---|
| default clim. | | MODIS clim. | | MODIS single | | | default clim. | | MODIS clim. | | MODIS single | |
| static | dynamic | static | dynamic | static | dynamic | | static | dynamic | static | dynamic | static | dynamic |
| 1% | 2% | -5% | 1% | -4% | 1% | AT-Neu | -10% | -11% | -11% | -11% | -10% | -11% |
| 695% | 680% | 704% | 681% | 711% | 681% | AU-DaS | 481% | 441% | 455% | 441% | 456% | 441% |
| 196% | 193% | 198% | 193% | 198% | 193% | AU-How | 129% | 110% | 114% | 110% | 114% | 110% |
| 116% | 163% | 180% | 169% | 179% | 169% | AU-Stp | 62% | 54% | 66% | 54% | 66% | 54% |
| 54% | 57% | 52% | 57% | 52% | 57% | AU-Tum | 29% | 28% | 29% | 28% | 29% | 28% |
| 10% | 9% | 13% | 9% | 14% | 9% | BE-Lon | -8% | -13% | -8% | -13% | -8% | -13% |
| 37% | 36% | 36% | 36% | 36% | 36% | BE-Vie | 7% | 7% | 7% | 7% | 7% | 6% |
| 131% | 133% | 133% | 133% | 133% | 134% | CA-Oas | 73% | 65% | 71% | 65% | 71% | 66% |
| 84% | 83% | 86% | 83% | 86% | 83% | CA-Qfo | 97% | 96% | 95% | 96% | 95% | 96% |
| -33% | -32% | -33% | -32% | -33% | -32% | CH-Fru | -44% | -44% | -44% | -44% | -44% | -44% |
| 53% | 47% | 50% | 47% | 50% | 47% | CH-Oe2 | 29% | 28% | 29% | 28% | 29% | 28% |
| 82% | 92% | 101% | 93% | 102% | 93% | DE-Geb | 45% | 27% | 45% | 27% | 46% | 27% |
| 161% | 154% | 152% | 154% | 150% | 153% | DE-HoH | 109% | 101% | 106% | 101% | 105% | 101% |
| 80% | 78% | 80% | 78% | 80% | 78% | DK-Sor | 12% | 11% | 11% | 11% | 11% | 11% |
| 51% | 50% | 51% | 50% | 50% | 50% | FI-Hyy | -3% | -4% | -3% | -4% | -4% | -5% |
| 251% | 253% | 250% | 253% | 251% | 253% | GF-Guy | 159% | 159% | 160% | 159% | 160% | 159% |
| 246% | 230% | 228% | 229% | 230% | 239% | IT-Cpz | 124% | 125% | 124% | 125% | 123% | 124% |
| -87% | -85% | -86% | -85% | -86% | -85% | IT-Lav | -25% | -25% | -25% | -25% | -25% | -25% |
| -42% | -42% | -42% | -42% | -42% | -42% | IT-Ren | -11% | -11% | -11% | -11% | -11% | -12% |
| 75% | 83% | 84% | 83% | 83% | 83% | IT-Ro2 | 25% | 16% | 22% | 16% | 22% | 16% |
| -7% | -1% | -2% | 0% | -2% | 0% | US-ARM | 14% | 1% | 16% | 1% | 16% | 1% |
| 48% | 48% | 50% | 49% | 50% | 49% | US-GLE | 36% | 38% | 39% | 38% | 39% | 38% |
| 455% | 429% | 552% | 433% | 552% | 433% | US-SRM | 298% | 303% | 339% | 303% | 338% | 303% |
| 36% | 39% | 45% | 40% | 45% | 40% | US-Ton | 6% | 3% | 14% | 3% | 14% | 3% |
| 58% | 74% | 72% | 74% | 72% | 74% | US-Var | 60% | 53% | 66% | 53% | 66% | 53% |

*Author contributions.* SW prepared model setups for the selected sites and prepared input data from available datasets in consultation with ST and AH. Model source code was provided by ST. SB supported setup of ECLand model runs. Simulations, analysis and plotting were done by SW with the involvement of AH and ST. SW took the lead in writing the manuscript with contributions from all authors.

*Competing interests.* The authors declare that they have no conflict of interest.

*Acknowledgements.* This work used eddy covariance data acquired and shared by the FLUXNET community and by the TERENO network. On-site LAI data from site "Hohes Holz" was acquired, analyzed, and shared by the working group Model Driven Monitoring led by site PI Corinna Rebmann at Helmholtz-Centre for Environmental Research - UFZ. We thank all station PIs, scientists, and technicians for their efforts in collecting, processing, and sharing their data.

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
