# Peer review of "Does dynamically modeled leaf area improve predictions of land surface water and carbon fluxes? - Insights into dynamic vegetation modules"

_EGUsphere, 2023_

## Referee Comment (RC1)

**Review of "Does dynamically modelled leaf area improve predictions of land surface water and carbon fluxes? – Insights into dynamic vegetation modules" by Westermann et al.**

**General Comments**

In this manuscript, Westermann et al. test a range of implementations of vegetation dynamics, namely dynamic vs static LAI from different data sources, within two land surface models, ECLand and Noah-MP. By contrasting the model performance across three different metrics for these different model setups, the authors identify not only which implementations produce better predictions of various fluxes but also potential reasons for these differences in model performance. In particular, the authors find that implementing dynamic vegetation in the two models actually decreased model performance with respect to the analysed fluxes. This is an interesting finding that is of importance to those both performing modelling studies and utilising the outputs of these models in model intercomparison studies.

Unfortunately, due to the number of comments I have, I must recommend major revisions to this manuscript before it is suitable for publication in Biogeosciences. In general, additional work is required to ensure the manuscript is well structured with clear and concise results and discussion. I am sympathetic to the fact that the length of this review and the number of technical corrections may be disheartening, but the majority of these are no more than the result of a further proofread and should be simple to address for the main author. I hope that these comments will provide the necessary guidance for bringing this submission up to the standards required by Biogeosciences.

**Specific Comments**

1. A key finding in this paper is that activation of vegetation dynamics in Noah-MP and ECLand does not improve model performance across several variables and metrics. Such a result appears at odds with the necessary testing that accompanies model development – new features are rarely implemented if performance is decreased. As such, I would like to see further exploration of the discrepancies between the findings from the model development team and this paper. For instance, the development of ECLand (and CHTESSEL) has papers by Miguel Nogueira, in addition to those from Souhail Boussetta, that explore model performance. It is important to synthesise such publications in the manuscript and investigate reasons for any divergence of results. As an example, if model iterations are tested against variables such as land surface temperature over water and carbon fluxes during development then perhaps the authors' results indicate a need for a broader testing process.

2. The methodology uses LAI taken from MODIS as both an input and the observation against which the model performance is analysed. Such analysis is circular and, although it could still prove useful in determining how model outputs change based on inputs, should be discussed further within the paper. It could also be avoided by using independent datasets – were efforts extended to identify alternative sources of LAI data, beyond the on-site data

used for DE-HoH? There are other remote sensing products and certainly other flux sites with on-site LAI measurements.

3. The authors investigate the behaviour of two models, ECLand and Noah-MP. However, the analysis is limited in places due to the static implementation of Noah-MP not producing output for carbon fluxes. With a large array of LSMs to choose from, further justification for selecting a model that can only partially contribute to the manuscript is required.

4. The site selection for the study was performed such that sites with the same IGBP PFT class that fell within the same aridity bracket as already selected sites were dropped. This takes place to "avoid including more than one representative site for each combination of aridity and vegetation type". This data selection criteria, in addition to the sensible choice of excluding sites with less than 5 years of data, results in 22 sites being used out of a potential of 212 FLUXNET2015 sites available (of which 120 have 5 years or more of data). More information on this is required. What were the possible consequences of having two or more sites in the same PFT and aridity classes? One would assume that including more sites could provide additional insight into the reasons for model performance – namely helping provide strength to any statements made around the role of PFT and aridity interaction. In addition, the study is then such that 5 grassland sites are included but only one savannah site and one mixed forest site. How might this unbalanced dataset affect the interpretation of results?

5. Unfortunately, the manuscript would benefit from an additional thorough proofread. I have included many small issues in the Technical Corrections below, but there are sure to be some I have missed. Some errors are particularly important to address for a scientific publication – for instance, there is a reference to a dataset having an author of "The PLOS ONE Staff" as the bibliography reference is to a correction of the original dataset article. Aside from this, the Results and Discussion section is hard to follow in places and frequently jumps from discussing one topic to another before returning to the original topic. It is not always clear which figure or model run is being discussed. Figure 8 is introduced multiple times, and some statements are duplicated. I would suggest the authors restructure the manuscript to have separate sections for the results and the discussion as this should provide additional structure and clarity. This should be combined with the addition of more quantitative results, providing numbers to support statements, and further explanation of how results are reached. For example, on line 223, "simulation results were unaffected by the type of LAI forcing with vegetation dynamics switched on" could be explained further by clarifying that this is seen from the symbols being in similar locations in Figures 2c and 2d. Supporting this statement with the mean difference in model performance across the sites would also help satisfy the need for more quantitative results.

**Technical Corrections**

1. Line 2: Superfluous "and".

2. Line 5: "improves" should be "improve".

3. Line 7: "range in" should be "range of".

4. Line 8: "and use more … "Hohes Holz"." is overly detailed for an abstract.

5. Line 9: "current implementation" – the current implementation of what exactly?

6. Line 9 and elsewhere: the use of "e.g." is not ideal as it is not clear what other processes or variables are to be inferred. I would amend this to detail the results more explicitly.

7. Line 10: "while Noah-MP improved it only for some sites" should be more along the lines of "while performance improved in Noah-MP only for some sites".

8. Line 13: "One reason, we showed here, might …" would read better as "We show that one potential reason for this might …".

9. Line 17: "For both, water and carbon fluxes" should be "For both water and carbon fluxes".

10. Line 28: What does "a.o." mean? I do not believe this to be a standard English abbreviation.

11. Line 30 and throughout the manuscript: "like Best et al. (2015) or Krinner et al. (2018)." I would suggest that citations are included in the usual manner with the conjunctions implied. Line 30 would hence become "Such works that introduce individual evaluation schemes are *often* accompanied by studies that perform comparisons between them (Best et al., 2015; Krinner et al., 2018)."

12. Line 33: "than an ensemble of LSMs". With 'ensemble' often having a specific meaning within the LSM community, I would suggest changing this line to read "than any single LSM", or similar, to more accurately reflect the findings of Best et al.

13. Line 34: "This does not allow to judge whether the investigated method achieved a (dis-)satisfactory performance". The authors of benchmarking studies would likely disagree with this statement, with one purpose of benchmarking being to assess whether performance is satisfactory against various a-priori expectations.

14. Line 37: "… a closer look on the cause …" should be "… a closer look at the cause … ".

15. Line 44-45: "Nonetheless, … to be further explored" is a difficult sentence to parse. I recommend rewording this.

16. Line 46: "thereby" is not needed.

17. Line 47 and throughout the manuscript: FLUXNET Multi-Tree Ensembles are one type of product, considered "a precursor to FLUXCOM" as stated in the Jung et al. paper cited here. I would suggest referring simply to FLUXCOM, a well-known product in the community.

18. Line 48: The sentence containing "… is crucial to make use of LSMs …" is unusually worded and therefore difficult to parse.

19. Line 52: "… that LSMs do misrepresent …" should be "… that LSMs misrepresent …".

20. Line 52: How do LSMs misrepresent water-sensitive regions? This statement should be expounded upon.

21. Line 57: "Currently, most LSMs are not able to represent a direct vegetation control on surface exchange". Do vegetation parameters not influence transpiration within LSMs and therefore exert a control on the land-atmosphere exchange of water?

22. Line 58: "… amongst others because …" should be "amongst other reasons …".

23. Line 60: Missing comma after "files".

24. Line 61: Missing "a" between "as" and "prognostic".

25. Line 62: Missing "of" between "understanding" and "how".

26. Line 62: "… by LSMs helps to shed light on the known discrepancies". "helps" should probably be "would help". Which discrepancies are being referred to here?

27. Line 64: "Here, we investigate model performances for water and carbon fluxes especially with focus on vegetation processes." This sentence could be "Here, we investigate model performance for water and carbon fluxes with a focus on vegetation processes."

28. Line 66: "… that can only be executed for a limited set of models". This needs clarification – is this due to time constraints within the study, or is there some other characteristic of certain models that exclude them from such studies?

29. Line 69: "those LSMs" should be "the LSMs".

30. Line 70: "Does improving one variable, compromise performance in the other or improves it along with it?" could be written more clearly and without the unnecessary comma. For example, "Do improvements in model performance for one variable compromise performance for other variables?"

31. Line 71: What is meant by "different patterns" and "possible misrepresentations of the observations"? This could likely be stated more clearly.

32. Line 74: This statement is superfluous and could either be moved or removed.

33. Line 77 and throught manuscript: There is no space between FLUXNET and 2015.

34. Line 80: The data from Trabucco and Zomer (2018) should be referred to as the CGIAR-CSI Global-Aridity and Global-PET Database.

35. Line 87: "… we assumed [the sites] to be neither very predictable nor very unpredictable in total …". This statement is unclear in both its meaning and implications.

36. Figure 1: Reference the IGBP classification scheme used, as well as where this data was obtained for each site (e.g., from the FLUXNET website, or within the site netCDFs). I believe some of the sites in this study have 19 years of data available – why does the scale have an upper limit of 18? Furthermore, a continuous color scale could likely be used here to allow differentiation between adjacent numbers (and clarify which color each label belongs to).

37. Line 94: "e.g." should be "i.e."

38. Line 95: Should the soil water content have an abbreviation introduced here in the same vein as the fluxes?

39. Line 98: The Climate Data Store citation is an unusual format – stick to the normal style and create a bibliography item for this dataset.

40. Line 99: Rather than "We adopted the same procedure …", simply say that you also excluded timesteps where L <= 0.

41. Line 101: How long did the gap-filled periods need to be to be excluded from the model performance analysis? How might this affect the results from the analysis?

42. Line 105: Why were only these four quality flags allowed? Other studies indicate any value less than 64 is usable (e.g., Fang et al., 2012; Ma and Liang, 2022).

43. Line 105: "as trade-off" should be "as a trade-off".

44. Line 107: Cite papers to provide assurance that using a Savgol filter is suitable for this purpose (e.g., Cao et al., 2018; Chen et al., 2004; Huang et al., 2021).

45. Line 110: "Each following year … that specific year." is ambiguous in terms of which year is being used as the forcing.

46. Line 112: "If LAI values for more than one month were not available" – did these months need to be consecutive?

47. Line 113: Move the "also" from before "on-site" to before "available".

48. Table 1 and throughout the manuscript: Is there any potential for providing shorter labels for the "Terms" from this table? While they are descriptive which is useful, they can be unwieldy in length.

49. Line 117: "Due to the use … from smoothing. Gaps were left as they were". This needs more explanation. Why are the same data from earlier (with QC flags of 48 and 65) not used here?

50. Line 125: Typo of "represents".

51. Line 127: I would change "an under-development vegetation dynamic module" to "a vegetation dynamics module currently under development".

52. Line 130: Delete "in".

53. Line 131: Explain what "IFS cycle "CY46R1" means.

54. Line 131: How were the IGBP PFT classes from FLUXNET2015 mapped onto the 19 vegetation types within ECLand? If the two classification schemes do not exactly match (or say, the ECLand types are taken from default data based on lat/lon of the site), were any tests performed to confirm that the classes aligned in a suitable manner?

55. Line 131: "parameter" should be "parameters" or "parameter values" (or similar).

56. Line 131: "stomata resistance to water and carbon flux" should be just "stomatal resistance".

57. Line 135: Does "respective cover" refer to the fractional cover of each of the two vegetation heights?

58. Line 135 and throughout the manuscript: "… to be used for the vertical exchange with the atmosphere" is superfluous text. There are instances throughout the manuscript where slight revisions of text could improve clarity and conciseness.

59. Line 149: See the comment for line 131 regarding the PFT classes for ECLand. The same holds here for Noah-MP.

60. Line 151: Missing "the" between "between" and "canopy".

61. Line 152: Unnecessary "thereby".

62. Line 152: "Stomatal resistance is controlled by photosynthesis". Is this statement true for Noah-MP? I would think it is more of a coupled relationship where stomatal resistance can also be controlled by e.g. vapour pressure deficit which in turn would decrease the level of photosynthesis by limiting the available intercellular $CO_2$.

63. Line 160: While the height of flux tower would ideally be dependent on the vegetation height, this isn't always true – towers can be situated within the canopy or many meters above it.

64. Line 161: How deep was the uppermost soil layer?

65. Line 162: Was the ten-year spin up sufficient to reach a steady state in each model? What variables were used to check that such a steady state had been reached?

66. Line 164: What "initial data" was taken from ERA-5? Why was the FLUXNET2015 data not suitable?

67. Line 172: Why are the soil data averaged from neighbouring cells?

68. Line 175 and footnotes: Do not use footnotes. Biogeosciences journal guidelines specifically say to avoid them. Just cite this dataset as usual with a bibliography entry.

69. Line 177: What aspect of the model meant that the temperate vegetation did not regrow? Is the requirement to have green vegetation fraction set to 1 a detriment to the results or their interpretability?

70. Line 180: Delete "therefor".

71. Line 180: Is the implicit temperature time scheme for the surface temperature?

72. Table 2: Why is the IGBP class OSH in this table when it does not feature in Figure 1? How was the initial LAI changed for sites in the Southern Hemisphere, namely the Australian sites? With DBF LAI set to 0.0, it seems clear that the Noah-MP initial LAI is based on a year starting on 1 January in the Northern Hemisphere, yet the Australian sites are potentially at the peak of their growing season in January.

73. Line 183: "transferred" would be better as "aggregated".

74. Line 184: Unnecessary comma after "output".

75. Equation 1: I do not think this is needed as Pearson's correlation coefficient is widely used and available in many programming languages.

76. Line 191: Is "co-domain" the correct term? It usually refers to the domain of a dependent variable. "Domain" is likely the proper word here.

77. Line 192: Typo of "therefore".

78. Equation 2:  This is justified as avoiding division by 0 or values very close to zero. However, this doesn't strictly follow from the formulation of the divisor. If the observations have very low variance or are very biased towards 0 values, then conceivably the mean minus the minimum could still be a very small number.

79. Line 199: "To account for" should likely be "to analyse" or similar.

80. Line 200: "variable to that" should be "variable on that".

81. Line 204: "e.g." should be "i.e." as the authors list every metric.

82. Line 204: Was an abbreviation considered for the normalized standard deviation to improve the ease of referring to it, and bring it in-line with the other two metrics which are referenced with a single letter?

83. Line 205 and throughout the manuscript: An abbreviation has been introduced for latent heat, so it could be used here. This is frequently the case throughout the manuscript.

84. Line 206: Delete "as" before "independent".

85. Line 209: Again, cite the code as is standard with a bibliography entry rather than a footnote.

86. Line 212: "LAI model" should be "model LAI".

87. Line 212: Consider changing "The point of optimal model performance is indicated with a star" to "The location an optimal model would occupy is indicated …".

88. Line 220: "a bunch of" should be avoided – what was the actual number of sites?

89. Line 225: This is confusing wording as it is difficult to determine whether the authors are referring to all the sites, one specific site, or just some sites.

90. Line 228: "whether the predicted LAI fit better … was random". Was the difference in performance random with respect to the sites' classes or aridity? It might be better to say that there was no clear relationship between the difference in performance and the site characteristics explored.

91. Line 231 and throughout the manuscript: Define which classes are meant by "short or sparse vegetation types".

92. Line 231: "Especially short … performance for LAI" is a difficult sentence to parse.

93. Line 234: Comparing the static simulations across Figure 2 (and the other Taylor diagrams) is difficult as the end of the arrows are hard to locate, especially with respect to the site that the arrows represent when the arrows are clustered.

94. Line 236: "With activated vegetation dynamics … in the Taylor diagram". This statement implies that performance improves for all sites and all LAI forcings, yet clearly for the default LAI, model performance decreases for AU-Stp and US-Ton.

95. Line 238: "did not contribute to improve LAI" would be better as "did not result in improved LAI".

96. Line 240: This relates back to the restructuring of the manuscript but starting a new paragraph before "Figure 3" would improve clarity.

97. Line 242: Figures 3d-f are referenced but Figure 3 does not have sub-labels.

98. Line 249: Delete "the" from before "disaggregating".

99. Line 249: The total LAI is disaggregated into high and low, yet the model is run with either only high or low vegetation. How does this impact results, as one can imagine this results in lower LAI than truth.

100. Line 255: "Updating the LAI forcing …" is a sentence that appears misplaced.

101. Figure 2: Why does the arrow of US-Var extend outside of the plot domain in Figure 3c? How does US-GLE in Figure 3c have no change in either standard deviation or correlation yet an extremely large change in the relative bias? This would imply a simple shift in magnitude in the LAI output which would be striking if caused by the switch to dynamic vegetation.

102. Figure 2 and others: How were the aridity brackets defined for the color coding?

103. Figure 3: Since other figures are in color, I would suggest this figure also use color to differentiate between static and dynamic to help visually distinguish between the two.

104. Line 270 and throughout the manuscript: More consistency in the used definition of "model performance" would be good and can be aided by being more explicit about the metric currently being discussed.

105. Line 279: More explanation of how the opposing NEE biases indicate differences in respiration estimates is required.

106. Line 297: Delete "thereby" after "types".

107. Figure 4: Why is AU-Stp outside of the plot area for Figure 4a? The axes should be extended so that the site falls within the plot area.

108. Line 303: Delete "fluxes" before "predictive".

109. Line 304: The "is" after ECLand should be "are".

110. Line 304: "Findings from this study … modelling carbon and energy fluxes". This is a strong statement about the impact of this work and requires more discussion to support it. Which processes within ECLand has this study identified as requiring further development? How has the study provided evidence for how these processes should be improved within the model?

111. Line 309: "Statistical measures" is a broad term. I would recommend replacing with the specific metrics that were calculated and explored in this study.

112. Line 309: "Stevens … with static ECLand" is not needed as the precise results from these other studies are not critical to the discussion. Instead, these two papers could simply be cited to support the prior statement that the results are comparable to other studies. If the exact values from the prior studies are mentioned, then it would be good to also state the same metric values from this study explicitly.

113. Line 312: Without being explicit about the methodology used for the literature review, it is also not necessary to state that no other studies were found. This is semi-implicit (if even required) in only having the two above citations.

114. Line 318: "… points appeared to have the largest arrows". This statement could be supported quantitatively with a measure of length for the arrows, equivalent to the degree of performance difference between the two model runs.

115. Line 322: "… no trend regarding vegetation type or site aridity can be seen …". Were any statistical tests to check for a trend performed here? If not, then changing "trend" to "relationship" might be preferable.

116.  Line 329: It would be good to explore the low EF / high NEE performance in forests in more detail. What processes are likely to be responsible for this mismatch in model performance? It is findings such as these that, with further discussion, would support the statement from my comment 115.

117. Line 335: I would include the soil moisture plots in the appendix.

118. Line 340: Slightly more explanation for how the underestimation of GPP/LAI could cause the poor EF performance is needed. A few words on the linking mechanisms would be sufficient.

119. Line 341: Delete "and" from before "might also be the reason …".

120. Line 342: Add "and" before "sensible".

121. Line 343: Change "has the potential in improving" to "has potential for improving".

122. Line 344: Activating vegetation dynamics in Noah-MP arguably had more than "a small impact" on LE and EF for certain sites. AU-DaS noticeably has significant displacement in position between static and dynamic runs in Figures 5d and 6d. Similarly, comparing the position between static runs for AU-DaS with default and MODIS LAI, there is clearly a large difference in model performance.

123. Line 346: I would suggest more information on the possible causes of disagreement between Ma et al. and this study. Why might different results have been reached? I would also replace "already concluded" with "found", otherwise it reads as if the authors are dismissing their own results!

124. Line 350: Delete "more" from before "sufficient".

125. Line 357: To what measurements does "optimal values" refer?

126. Line 361: This statement is not clear.

127. Line 363: Add "a" before "metric" and replace "the bar plots of Fig." with "Figure".

128. Line 363: "Surprisingly, the model quality of those actually closely related variables was independent". This sentence needs work. What does model quality mean? Which variables are considered closely related, and why? How does this affect the confidence in the results?

129. Line 372: Typo of "or" as "of".

130. Line 372: Move "do" from after "LAI" to after "sites".

131. Line 377: Capitalise L in "ECland".

132. Figure 7: Keep the x axes constant across the nine plots. This ensures that comparison between the plots is easy and does not mask the differences in performance. This is also the case for the other figures – where the point of subfigures is to allow comparison between them, ensure that all scales are consistent as this provides ease of comparison. It is also necessary to describe what each element of the boxplots represents.

133. Figure 8: Which LAI is used for the models in this figure? I would suggest less transparency for the MAM and SON points, or just use different colors. Moving the range indicators outside of the plotting area would ensure they do not cover points on the plots.

134. Line 398: It is "an evergreen", not "a evergreen".

135. Line 407: I would suggest changing the units that GPP is reported in such that the values do not need to be reported at so many decimal places.

136. Line 408: Are the MODIS values of LAI varying between 1 and 7 realistic? It should be clear whether the authors believe the LAI or GPP is the most likely reason for the two variables to not align.

137. Line 412: Replace "depends next to LAI also" with "also depends".

138. Line 418: Add "of" between "values about".

139. Line 419: This sentence makes it unclear which sites were being discussed previously – the start of the paragraph indicates that all of the sites are being discussed but then here it is stated that similar behaviour is seen at a specific site.

140. Line 433: Add "a" between "shows" and "similar".

141. Line 435: Replace "govern this daily" with "govern these daily".

142. Line 436: Replace "GPP relates linear to LAI" with "GPP is linearly related to LAI".

143. Line 439: Replace "phase" with "phases" and add "the" between "biomass from" and "previous time".

144. Line 441: Add "the" in two places – between "part of" and "senescent biomass" and also between "reduced in" and "case of".

145. Line 444: Is the 11% in the model? If so, how does this compare to observations?

146. Line 446: Replace "minimize net primary production or even produce negative values" with "reduce net primary production, even producing negative values".

147. Line 454: Delete "However, ".

148. Line 458: "However, an evaluation of the representativeness of key variables like lead area index or net ecosystem exchange is rarely done". I would agree this is frequently a part of model evaluation, and therefore needs to be more specifically worded to accurately infer what the authors are saying.

149. Line 467: Replace "… higher variability in the ecosystem exchange especially of short or sparse vegetation but this was predominantly …" with "… higher variability in ecosystem exchange, especially that of short or sparse vegetation, but this was predominantly …".

150. Line 468: It is "a negligible" not "an negligible".

151. Line 473: It should be "observations".

152. Line 474: Replace "relation" with "relationship".

153. Line 475: Replace "linear" with "linearly".

154. Line 477: Replace "… pinpoints to the reasons of model behavior … " with "… pinpoints the reasons for model behavior …".

155. Line 477: In general, the conclusion is very long. I would recommend synthesising the study impacts in more detail in a Discussion section and keeping the conclusion a short summary of this.

156. Figure A1: Even though the sub-panel d would be identical to sub-panel c as stated in the caption, I would still include it. The space is free anyway so there is no cost to this, but it will emphasise the similarity of the two plots, especially if the caption still mentions that they are identical.

157. Tables A1 – A6: What are the column headings? How do they relate to the different model runs?

158. Table A6: This appears to disagree with the statement made at line 334 that model performance for soil moisture is insensitive to LAI forcing or vegetation dynamics. Assuming that each column in Table A6 is one of the different model runs, then sites such as US-SRM (relative bias of ECLand varies from 314% to 552%) appear to have quite varying performance, even if it is consistently poor.

**Bibliography**

Cao, R., Chen, Y., Shen, M., Chen, J., Zhou, J., Wang, C., & Yang, W. (2018). A simple method to improve the quality of NDVI time-series data by integrating spatiotemporal information with the Savitzky-Golay filter. *Remote Sensing of Environment*, *217*, 244–257. https://doi.org/10.1016/j.rse.2018.08.022

Chen, J., Jönsson, Per., Tamura, M., Gu, Z., Matsushita, B., & Eklundh, L. (2004). A simple method for reconstructing a high-quality NDVI time-series data set based on the Savitzky–Golay filter. *Remote Sensing of Environment*, *91*(3), 332–344. https://doi.org/10.1016/j.rse.2004.03.014

Fang, H., Wei, S., & Liang, S. (2012). Validation of MODIS and CYCLOPES LAI products using global field measurement data. *Remote Sensing of Environment*, *119*, 43–54. https://doi.org/10.1016/j.rse.2011.12.006

Huang, A., Shen, R., Di, W., & Han, H. (2021). A methodology to reconstruct LAI time series data based on generative adversarial network and improved Savitzky-Golay filter. *International Journal of Applied Earth Observation and Geoinformation*, *105*, 102633. https://doi.org/10.1016/j.jag.2021.102633

Ma, H., & Liang, S. (2022). Development of the GLASS 250-m leaf area index product (version 6) from MODIS data using the bidirectional LSTM deep learning model. *Remote Sensing of Environment*, *273*, 112985. https://doi.org/10.1016/j.rse.2022.112985

---

## Referee Comment (RC2)

Review

*General comments*
The manuscript 'Does dynamically modelled leaf area improve predictions of land surface water and carbon fluxes? - Insights into dynamic vegetation modules' by Westermann et al. aims to address the sensitivity of simulated carbon and water fluxes as well as leaf area index itself to prescribing or dynamically simulating the leaf area index in two terrestrial biosphere models as well as exploring the sensitivity to different prescribed LAI datasets. Simulations are conducted and compared against a suite of FLUXNET2015 sites. The authors find that LAI is sensitive to their simulations set-up (i.e. prescribed vs simulated LAI) but simulated carbon, water and energy fluxes are not strongly impacted. I think the research questions addressed are interesting and novel, and I appreciate that conducting the experiments and associated analysis were a lot of work. However, I have some major concerns before the manuscript can be considered for publication. I want to stress that I am aware a lengthy review can feel disheartening but my comments are meant to be helpful for revising the manuscript:) So please don't read it too negatively.

- My main concern is that the methods description is a bit confusing and I'm not sure I quite understand the experiment set-up.
    - Do you have two sets of experiments, where 1) you test the impact of LAI datasets on simulated LAI and carbon/water/energy fluxes and 2) where you 'simply' switch on/off the dynamic vegetation module? If so why is 1) not part of your research questions in the introduction? In parts of your manuscripts it reads like you prescribe LAI but it is dynamically simulated at the same time which I don't understand (see for e.g. caption Fig. 7)?
    - Where is LAI as an input driver coming from in the LUTs? How does it differ between the experiments (LUT vs your LAI? Is LUT also based on MODIS?) Not everyone is necessarily familiar with the look up tables of the specific models chosen for this study so it'd be good to clarify this.
    - The section where you compare LAI across your experiments almost seemed a bit circular to me, and I would suggest to reduce the emphasis on LAI and focus more on the simulated fluxes where you can avoid the interdependence of input and output LAI during evaluation (and this is also appropriate given the title of the manuscript). Alternative remotely sensed LAI datasets are available, although this comparison of course also would be a bit unfair.
    - Towards the end of your results/discussion section you describe what's happening in the model and how this explains some of your model results which is great! I think it could help your manuscript if in the methods the model descriptions had more detail too for the relevant processes
    - Throughout your manuscript it would help readability if you had specific experiment names that are consistently italic (or any other distinct formatting) like you attempted in L158.
    - Other small things I would like to suggest are:
        - Split the Results and Discussion section - the way it is written now, it is a bit of a back and forth and hard to follow.
        - Include a table with ALL experiments listed (expand Table 1).
        - Part of my confusion also stems from how the results are presented, and simply including headings could really help. Something like: 3

Results/ 3.1 LAI /3.1.1 Impact of different LAI initial files/ 3.1.2 Prescribed vs dynamic LAI (and so on for the remaining variables).

- ○ You need to be more careful with the metrics you used for model evaluation, see below for specific comments
- ○ I was also a bit surprised about your model selection? Why did you choose a model that couldn't provide all necessary outputs for all simulations you conducted?
- ○ Why did you initialize your model simulations differently (ECLand vs Noah-MP)?
- ○ You report that dynamically simulated vegetation leads to a lower model performance, at least in LAI. One thing I wondered is whether your model simulates the 'right' vegetation type for each site you considered (or do you define the vegetation type that is simulated)? You also point out multiple times how forests tend to show better model performance than shorter vegetation types, but you don't offer any explanations why that might be the case?

- ● The overall structure of the introduction makes sense to me but sometimes it is hard to tell what message you try to convey? I know this is quite wishy washy but I provided more specific comments below.

*Specific comments*

L7 Maybe change to 'We compare model results with **observed** fluxes from the FLUXNET [...] or similar
L8 MODIS leaf area **index** ?
L8 More detailed information? What does this mean? If the only additional output is LAI, you might as well explicitly state this here but in general I think this is weirdly specific for an abstract the way it is written now
L13-14 This is not really a reason that explains weak model performance, but just another way to phrase poor model performance! I think also the abstract is not a place for speculation but you should clearly state what the drivers for poor model performance are based on your study

L21 You could already state in the first paragraph where LSMs are used (you do give the CMIP example in L26 but LSMs are also used in meteorology models, reanalysis [...]) before diving into the more specific applications to motivate your study, and from there go to your model validation topic
L26-L36 In general this paragraph discusses schemes to evaluate model performance, which I think is a useful topic for your introduction! But I'm not sure what the key point is you're trying to make here. Are you trying to build up to presenting a new evaluation scheme in your paper?
L24-25 This doesn't offer a lot of information. For example, you could mention why more features are added to LSMs to give this sentence more value
L28 I suggest 'global, regional, and site scale'
L29-31 I'm not sure I understand this. Do you mean that evaluation schemes are compared against each other? Or model - obs comparisons?

L34-35 This is also unclear to me. Did they evaluate some LSM ensemble average against statistical methods? I also don't understand how not reporting individual model performance is linked to only having normalized metrics, and also why normalized metrics are not useful. Could you explain this a bit more?

L37 Replace 'had a closer look' with explored, investigated or something similar?
L40 This is unclear to me. Do you mean that they didn't find an error in the observations they compared the model simulations to?
L41 I get your point but to me this almost reads like model-observation comparisons can't help identify areas of uncertainty at all which is not true (see e.g. Whitley et al., 2016 for one of the many studies that identify reasons for inter-model differences and mismatches with obs)

L45 Could you name a few references to support your statement here?
L46-47 This seems a bit out of place and unnecessary to focus on a single benchmarking dataset that as far as I can tell you're not even using in your study? I think the first two sentences are a bit dis-connected from the rest of the paragraph anyway where now all of a sudden you focus on drought and water-limitation (why?)

L66-67 Why did you choose ECLand and Noah-MP? Other than them being used by others
L74-75 I think you can drop this:)
L80 Why did you invert the Aridity Index? I know you took it from a dataset but it'd be good to know how it is calculated
L83 Why +- 0.1 log AI? Is this a common threshold?
L87 I'm not sure the predictability of your sites necessarily follows from your selection procedure. But maybe I misunderstand - can you explain this a bit more?
Fig.1 Can you also explain the colorbar in the Figure caption?
L97 Why was precipitation set to zero in the gapfilling?
L97 Is using a Kalman filter a common procedure for gapfilling? How is gapfilling achieved in FLUXNET?
L97-98 ERA5 is relatively coarse if I'm not mistaken, and it'd be worthwhile mentioning the shortcomings and the reason for choosing this reanalysis? I assume you chose it because of the high (3h?) temporal resolution which would also explain the 3h cut-off between Kalman Filter and using ERA5
L100-101 What's the cut-off for 'longer gap filled' periods?
L103 You already defined LAI in L59
L104 What is the temporal and spatial resolution of MODIS LAI?
L105 I know you point to the documentation but could you please explain the choice of quality flags (and what they mean) in the manuscript?
L106-107 I'm not sure I'm following. How does selecting certain quality flags help having the 'same amount of data in a month'?
L112 You quite happily use LUT LAI as if this was a commonly known dataset but personally I have no idea where the data in your look up tables are derived from. Is this look up table derived from MODIS, which years were used for it [...] - you don't have to state every little detail here but perhaps there is a reference with a description of how this look up table was derived?
Table 1 The way you described *MODIS climatological LAI* and *MODIS single-year LAI* in the table is not super clear. The first one is just a monthly climatology (? based on which years),

the second is the first year of the simulation period on monthly timesteps? Could you rephrase this?

L127 'under-development' does this mean the model isn't 'ready' yet?

L129 Is the daily resolution really an appropriate reason to choose those quality flags? Why did you avoid smoothing your data for LAI?

L133-134 So only two vegetation types per grid cell are simulated?

L134-142 It's great you explain a bit of the model, but which photosynthesis model is used? It would also be good to know how LAI interacts with GPP/NEE, and also with heat fluxes because these relationships are key to your analysis - but of course I appreciate you can't explain every line of the model code here

L149 'taken from LUT' I would replace this with 'were prescribed' or similar

L152 'Thereby' I'm not sure this follows from the previous sentence

L150-156 Again, I think it would be interesting to know which photosynthesis scheme Noah-MP employs, and to get a rough idea how LAI interacts with GPP/NEE and heat fluxes etc in the model

L164 What does global initial data mean here? Did you conduct the model spin up using ERA5? Why didn't you choose the meteorology that comes with the flux sites (like you did with Noah-MP)?

L166 Which site information - do they all report vegetation height and you used a certain threshold?

L169 Is there a reference for the van Genuchten soil hydrologic params?

L171 Could you please update this reference - PLOS ONE is a journal and not the dataset name, and the author of the dataset surely is not 'The PLOS One Staff'!

L172 So the initial conditions for the two model runs are not identical? Why not?

L173 I would have thought each gridcell has 8 neighboring gridcells?

Table 2 Could you include the full name of the vegetation types and also the actual name of the Noah-MP vegetation classes? I'm not sure I mentioned it elsewhere but a similar table for ECLand would be great too

L183 What does 'transferred' mean here? Did you calculate daily averages/sums?

L188 You are also using the Pearson Corr. to assess non-linear relationships but this is not a suitable metric (Pearson Corr. only describes linear relationships; see further below)

L189-190 Which symbol represents the normalized standard deviation metric?

L191-193 I'm not sure I understand why you subtract the minimum from $\bar{x}$? I also wonder how you interpret the relative bias when $x_{min}$ differs strongly from $\bar{x}$?

L199-200 So you try to understand the relationship between output variables within the same model simulation?

L199-207 I don't really understand how you calculate the elasticity. What is the 'slope of their correlation'? Is there a reference that uses the same definition for elasticity you could include? Is the -0.1 - 0.1 threshold common (reference)?

L213 'The model performance of the dynamic run is shown with the symbol' - which symbol ?

L217-218 Can you name some references to support 'in line with results in the available literature'? It would also be useful to see some metric -> same for the next sentence about Noah-MP. How much more biased??

L222 Here e.g. you compare the LUT LAI runs with your generated LAI forcing but in order to fully understand the performance difference it would be useful to know where this mysterious LUT LAI comes from??

L224 'simulation results were unaffected by the type of LAI forcing with vegetation dynamics switched on' I must have completely misunderstood your experiment set-up, but I thought when vegetation dynamics are switched on, LAI is simulated (not prescribed?)

L225 'but not necessarily for all sites' can you quantify this more?

L226 Here for example (and across the entire section) it would be helpful to stick to the nomenclature for your experiment you defined in the table earlier, and highlight it with a different font type

L226 'increased variance' compared to what?

L227 'was random' - what does this mean?

L231 What are sparse vegetation types (i.e. did you define it anywhere)?

L243-244 Can you also give a value range for ECLand - it'd help the comparison between the models

L246-252 Here you kind of say a bit about where original LUT is coming from, and this belongs in the methods section already

L255 Why are you not showing the single-year LAI results - could just be in the appendix?

Fig 2 How did you set threshold to separate the AI into different aridity classes? This belongs in the methods

Fig 3. I think you could use more distinct colors here, at least on my copy it's quite hard to see the difference in the shade of gray. Also why are there only single symbols for NEE and GPP (for dynamic simulations I think?) for Noah-MP? I also don't understand why you have different values for different prescribed LAI data for the dynamic simulations - but this is because I didn't quite understand the experiment set-up, as I said above. Does LAI not emerge from the spin-up (and therefore doesn't need to be prescribed)? The values do look very similar across all experiments. One more small thing, but the red dotted line is hard to see for the Pearson correlation and Normalized STD

L265 Here you dive into the impact of LAI-set-up on ecosystem exchange variables and it would be useful to see in the methods (at least to some degree) how LAI is actually linked to those variables?

L281-282 Then why did you use Noah-MP in this study? This kind of defeats the purpose of your study, at least according to the title

L284 I'm not sure this is the right conclusion - did you also find an overestimation of GPP? If NEE is right for the wrong reasons for the historical period that doesn't mean you can have a lot of confidence in simulated NEE in future scenarios

L289 'being independent of the prescribed LAI forcing' - so this is just the dynamic simulation that does not have any prescribed LAI?

L289-290 Not sure consistent is the right word here, those are two v different experiments

L291 How does the onsite LAI actually compare with the MODIS LAI? MODIS LAI can be biased for some flux sites (so a reason for the lower performance could just be that the model was tuned to match MODIS, and MODIS and on-site LAI are actually wildly different)

L306-307 Where is this recommendation coming from? Did you not say earlier there is no performance difference depending on the LAI forcing used?

L309-316 A lot of these studies look at different temporal resolutions, and I actually wondered too how your model simulations would do if you looked at climatologies, or annually aggregated values - this could help you identify whether the models get the broad

patterns  right. But having said that, you already cover a lot of ground with your study so this is more of a curiosity rather than an actual suggestion for the revision

L329-330 Interesting indeed - do you have any idea why this is happening?

L335 Again - why not include a figure in the appendix to show these results?

L361-362 Here it is so much clearer what your elasticity metric is meant to do! In general, I really like this part of the analysis

L351 How are vegetation and soil moisture state variables coupled?

L354-359 I think this is all valuable and true, and explains the general soil moisture bias well. But it doesn't explain your actual results, i.e. not seeing an impact of static vs dynamic vegetation on soil moisture

Fig 7 Where is the footnote for elasticity?

L389-396 Why didn't you show LAI-GPP elasticity in Fig 7 when this is such a strong focus in this paragraph? Do you have an idea whether it is more realistic to have a linear or non-linear LAI-GPP relationship (i.e. what do studies say that look at observed LAI-GPP relationships)?

Fig 8 What are the arrows in the figure? Some of the relationships in this figure (as you point out earlier) are clearly non-linear, and therefore using the Pearson correlation coefficient is not suitable. I would also suggest to change the GPP units to avoid having so many decimals - in this figure and also in the text

L403 You need to define $\sigma_r$

L409-410 You already say in the methods that you are masking MODIS using quality flags (and as I said there, please be more precise about what the flags mean)

L421-431 I like that you are explaining the model relationships here and link them with the results

L434 typo 'dyanmic'

L454-455 Can you give a reference for this ('real LAI'), also maybe replace real with 'observed' or similar

L461-462 I think this is a bit harsh and not true; depending on the focus (i.e. if it's purely benchmarking) of course this can happen, but there are also many papers that actually explain reasons for mismatches between obs and model simulations, and also differences across members of the LSM ensembles (see e.g. Whitley et al. 2016)

L467 Why did you expect that dynamic vegetation would improve ecosystem exchange especially for short vegetation?

L480-481 If you don't know how representative your models are compared to other LSMs then why did you choose them? I see both are part of the PLUMBER2 experiment - how do they compare to other LSMs there (see Abramowitz et al., 2024)?

TableA1-A5 are not referenced in the manuscript

You did not include the data and code availability statement!!

References

Abramowitz, G., Ukkola, A., Hobeichi, S., Cranko Page, J., Lipson, M., De Kauwe, M., Green, S., Brenner, C., Frame, J., Nearing, G., Clark, M., Best, M., Anthoni, P., Arduini, G., Boussetta, S., Caldararu, S., Cho, K., Cuntz, M., Fairbairn, D., Ferguson, C., Kim, H., Kim, Y., Knauer, J., Lawrence, D., Luo, X., Malyshev, S., Nitta, T., Ogee, J., Oleson, K., Ottlé, C., Peylin, P., de Rosnay, P., Rumbold, H., Su, B., Vuichard, N., Walker, A., Wang-Faivre, X., Wang, Y., and Zeng, Y.: On the predictability of turbulent fluxes from land: PLUMBER2 MIP

experimental description and preliminary results, EGUsphere [preprint], https://doi.org/10.5194/egusphere-2023-3084, 2024.

Whitley, R., Beringer, J., Hutley, L. B., Abramowitz, G., De Kauwe, M. G., Duursma, R., Evans, B., Haverd, V., Li, L., Ryu, Y., Smith, B., Wang, Y.-P., Williams, M., and Yu, Q.: A model inter-comparison study to examine limiting factors in modelling Australian tropical savannas, Biogeosciences, 13, 3245–3265, https://doi.org/10.5194/bg-13-3245-2016, 2016.

---

## Referee Comment (RC3)

Dear Editor,

The manuscript entitled "Does dynamically modelled leaf area improve predictions of land surface water and carbon fluxes? – Insights into dynamic vegetation modules " by Westermann et al., focuses on the effects that enabling dynamic Leaf Area estimation can have in terms of land surface model performance. The results based on two land-surface models (ECLand, and Noah-MP) suggest that Leaf Area, carbon and turbulent fluxes, and soil moisture representation does not improve when a dynamic vegetation module is employed. This is an interesting finding to the extent that this result feels counter-intuitive, and can be quite instructive on how employing a "fancier" new model does not necessarily increase model performance. Therein lies the novelty of this work. The authors have put some considerable effort and extensively discuss their results providing both their interperation and literature findings, but the manuscript is overall difficult to follow.

Despite I understand that this work could be of interest to members of the land surface modelling community, I am not convinced that the manuscript in its current form is suitable for publication, and would expect that the authors address the following comments and revise their work, before it can be considered for publication.

**Comments:**

- I understand that the authors use some existing Plant Functional Type in the respective models and do not tune PFT parameters to fit the dominant or average plant traits of each fluxnet site. However, the authors still perform comparisons and evaluate the models against observational point-level ecosystem-specific data. Is this the case? I find this slightly inconsistent, since plant functional types conceptually are mostly meant to be used in larger-scale simulations representing the average characteristics of a vegetation category, rather than being compared with site-level data.

- If we have a model whose parameters are not set to fit the observations (see above), then why would we necessarily expect that switching on a dynamic vegetation module (which is also unparameterized) should increase model performance? One could argue that solely switching the environmental dependency of LAI on should justify this expectation, but isn't the environmental dependency of LAI on average also embedded in a prescribed climatology by definition?

- I deeply appreciate that the authors are extensively discussing past research, their interpretations, and their results in full detail, which increases transparency, but the manuscript is overall difficult to read. IT would help a lot if the authors split the results from the discussion points. The manuscript would also need further proof-reading, since one can easily still find mistakes scattered across the text.

- The results suggest that model performance "regarding latent heat flux or soil moisture is independent of how LAI is represented" (Line 380). This is very counter-intuitive, and one would wonder whether this is the case because LAI is equally badly represented in all cases.

- Given that LAI dynamics are the main focus of this study, it is important that the authors describe in more detail the LAI modules of the two models. They start doing so in Line 420 and discuss allocation, senescence etc., but they need to do this in a comprehensive manner in the method section, and not scattered across the text.

- I think it is really important to show in the appendix panels with mean annual LAI, GPP, and NEE climatologies for every site, showing prescribed and model-predicted LAI. This would help a lot the interested reader understand the dynamics at play.

- The selction criteria for the fluxnet sites used seem arbitrary. Why do the authors drop sites of roughly similar aridity index? If anything, including more sites would increase the robustness of their results.

- In my understanding, in the switched-on dynamic vegetation runs LAI is freely estimated by the model and not constrained by some prescribed climatology. Is this indeed the case? From the results (e.g., Fig. 2) it feels as if there are different types of dynamic runs – one for each of the different possible LAI forcings. How is this the case? In my understanding, somehow these prescribed climatologies are still used to

"initiate" the dynamic runs. What does that exactly mean? If this is indeed the case, still LAI is free to evolve, so why do we end up with different model performance in every run, just because initial LAI conditions have been different? If this is not the case and LAI climatologies are somehow fed also into the dynamic runs, then how does it make sense to validate these results against the forcing LAI climatology itself? The authors need to try and clarify their setup more.

- Line 155-156: Was it not technically possible to maintain GPP and NEE estimation in the static Noah-MP runs, despite prescribing LAI? Or is there some other reason?
- Line 165: It is slightly unclear in which occasion ERA5 data are used
- Appendix figures: What do the sim00, sim02 etc. mean?
- Line 255: I would suggest that the reviewers show these results in supplementary material to ensure transparency (or don't mention at all)

---

## Author Response (AR1)

Response on Review 1

First of all, we want to thank reviewer 1 for reviewing this work of me and my co-authors. Thank you for putting a great deal of effort and discovering many details. Your feedback has greatly helped improving this work and publication. In the following, I will go through and respond to your comments.

Specific comments: (reviewer comments in italic, responses in normal font, changes in blue)

*1. A key finding in this paper is that activation of vegetation dynamics in Noah-MP and ECLand does not improve model performance across several variables and metrics. Such a result appears at odds with the necessary testing that accompanies model development – new features are rarely implemented if performance is decreased. As such, I would like to see further exploration of the discrepancies between the findings from the model development team and this paper. For instance, the development of ECLand (and CHTESSEL) has papers by Miguel Nogueira, in addition to those from Souhail Boussetta, that explore model performance. It is important to synthesise such publications in the manuscript and investigate reasons for any divergence of results. As an example, if model iterations are tested against variables such as land surface temperature over water and carbon fluxes during development then perhaps the authors' results indicate a need for a broader testing process.*

Indeed, model development needs testing and novel model modules are only incorporated when there is an improvement or at least no deterioration in model performance in the variables that were chosen for evaluation. The choice of variables for model evaluation is really important. Since ECLand mostly is used in Climate Projections, the most important variable is land surface temperature which then is used for model evaluation. However, changes in vegetation representation barely affect energy balance calculations, especially not on a coarser temporal resolution that often is used for model evaluation. As a result, it is hardly surprising that model performance in our investigation diverges from published results during model testing. Thus, yes, we wanted to indicate that model testing needs a broader spectrum of target variables and different temporal resolutions. The work of Nogueira et al. (2020) is interesting but they focused more on updating land cover fractions and vegetation type clumping which had an important effect on land surface temperature. We extended the discussion on model performance and the importance of vegetation-related variables in ECLand.

*2. The methodology uses LAI taken from MODIS as both an input and the observation against which the model performance is analysed. Such analysis is circular and, although it could prove useful in determining how model outputs change based on inputs, should be discussed further within the paper. It could also be avoided by using independent datasets – were efforts extended to identify alternative sources of LAI data, beyond the on-site data used for DE-HoH? There are other remote sensing products and certainly other flux sites with on-site LAI measurements.*

The LAI from MODIS used for model input and model evaluation is not identical. Model input is a LAI climatology on monthly basis resulting from multi-year average MODIS values. Model

evaluation is done with the daily MODIS values which are 8-day means. For the static runs, this comparison provides the information whether an incorporation of more site-specific climatology results in higher representativeness of local LAI evolution. For the dynamic simulations, comparing modeled LAI with daily MODIS values is used to examine whether the models are able to capture inter- and intra-annual LAI dynamics. However, we could show that even with the same source of the data the dynamic simulations are not fitting the observations.

In the revision, we now provide more details on the MODIS LAI data and highlighted the differences between data used for input and for evaluation. (Lines 217-220)

Yes, the evaluation would really benefit from using on-site LAI data from more than one site. We were very thankful for having an additional LAI data source at all. I (first author) tried reaching out to the FLUXNET community via their contact form several times but never had any responses.

*3. The authors investigate the behaviour of two models, ECLand and Noah-MP. However, the analysis is limited in places due to the static implementation of Noah-MP not producing output for carbon fluxes. With a large array of LSMs to choose from, further justification for selecting a model that can only partially contribute to the manuscript is required.*

We chose ECLand and Noah-MP because both models can be and are widely used for coupling them as LSMs with established climate projection models. Although Noah-MP provides no GPP and NEE output for the static runs, it still is interesting to look at the LAI-GPP relationship within the model that we did for Figure 8. Nonetheless, we need to be more careful with absolute statements that we did. We adjusted the abstract and the discussion according to that.

*4. The site selection for the study was performed such that sites with the same IGBP PFT class that fell within the same aridity bracket as already selected sites were dropped. This takes place to "avoid including more than one representative site for each combination of aridity and vegetation type". This data selection criteria, in addition to the sensible choice of excluding sites with less than 5 years of data, results in 22 sites being used out of a potential of 212 FLUXNET2015 sites available (of which 120 have 5 years or more of data). More information on this is required. What were the possible consequences of having two or more sites in the same PFT and aridity classes? One would assume that including more sites could provide additional insight into the reasons for model performance – namely helping provide strength to any statements made around the role of PFT and aridity interaction. In addition, the study is then such that 5 grassland sites are included but only one savannah site and one mixed forest site. How might this unbalanced dataset affect the interpretation of results?*

Representative site selection is an important issue and we gave it detailed consideration when designing the analysis. When looking at the global distribution of FLUXNET sites, many of them are located in temperate climate on the Northern Hemisphere. Including all sites with more than 5 years would create an overrepresentation of regions with high density in sites, resulting in an imbalance of PFT-aridity combinations for model evaluation with especially (semi-)arid short vegetation being underrepresented (which is one of the limitations Martens et al. (2020) and Nogueira et al. (2021) faced in their study). Thus, we needed some sort of filter algorithm to avoid that overall model performance is either shifted towards better or worse performance due to this imbalance.

Savannah types are indeed separated within IGBP PFT, but this is not done in the models. Accordingly, I did not separate them either when selecting the sites, meaning that SAV and WSA belong to the same group within this selection process. I also merged PFT type MF with DBF since, after the selection via the aridity index, only two MF remained which is critically few (this is mentioned in the manuscript in line 91). Other possible sites had to be removed due to low-quality in soil moisture data (mentioned in line 89). Unfortunately, there are not enough sites available to create a second set of the same structure, as some aridity-PFT combinations are really rare. We are aware that such a second set would be helpful for strengthening and reproducing our findings. We now explained in more detail why and how site selection was done, and adapted Figure 1 in accordance with the model PFTs.

*5. Unfortunately, the manuscript would benefit from an additional thorough proofread. I have included many small issues in the Technical Corrections below, but there are sure to be some I have missed. Some errors are particularly important to address for a scientific publication – for instance, there is a reference to a dataset having an author of "The PLOS ONE Staff" as the bibliography reference is to a correction of the original dataset article. Aside from this, the Results and Discussion section is hard to follow in places and frequently jumps from discussing one topic to another before returning to the original topic. It is not always clear which figure or model run is being discussed. Figure 8 is introduced multiple times, and some statements are duplicated. I would suggest the authors restructure the manuscript to have separate sections for the results and the discussion as this should provide additional structure and clarity. This should be combined with the addition of more quantitative results, providing numbers to support statements, and further explanation of how results are reached. For example, on line 223, "simulation results were unaffected by the type of LAI forcing with vegetation dynamics switched on" could be explained further by clarifying that this is seen from the symbols being in similar locations in Figures 2c and 2d. Supporting this statement with the mean difference in model performance across the sites would also help satisfy the need for more quantitative results.*

We took care for mistakes in citations and linguistic deficits and separated Results and Discussion section. We thoroughly proof-read the manuscript.

Technical Corrections
Thank you for careful reading and writing down the propositions with this detail! I will only respond to those that exceed language.

*4. Line 8: "and use more … "Hohes Holz"." is overly detailed for an abstract.*

True. Made it more general.

*5. Line 9: "current implementation" – the current implementation of what exactly?*
   L9 "Current implementation" meaning the model source code as is it published currently.

Added "of dynamic vegetation" to be more clear.

*11. Line 30 and throughout the manuscript: "like Best et al. (2015) or Krinner et al. (2018)." I would suggest that citations are included in the usual manner with the conjunctions*

*implied. Line 30 would hence become "Such works that introduce individual evaluation schemes are often accompanied by studies that perform comparisons between them (Best et al., 2015; Krinner et al., 2018)."*

done

12. *Line 33: "than an ensemble of LSMs". With 'ensemble' onen having a specific meaning within the LSM community, I would suggest changing this line to read "than any single LSM", or similar, to more accurately reflect the findings of Best et al.*
done

13. *Line 34: "This does not allow to judge whether the investigated method achieved a (dis-)satisfactory performance". The authors of benchmarking studies would likely disagree with this statement, with one purpose of benchmarking being to assess whether performance is satisfactory against various a-priori expectations.*

Here we are referring to benchmarking studies that use relative metrics to create a rank order of the models, like the ones of PLUMBER. Those do not provide information on whether the best model in this ranking really achieves good fit with observations since the absolute metrics are not shown. We have stated which types of studies we mean specifically in the introduction.

17. *Line 47 and throughout the manuscript: FLUXNET Multi-Tree Ensembles are one type of product, considered "a precursor to FLUXCOM" as stated in the Jung et al. paper cited here. I would suggest referring simply to FLUXCOM, a well-known product in the community.*

Done

20. *Line 52: How do LSMs misrepresent water-sensitive regions? This statement should be expounded upon.*

This paragraph was changed completely.

21. *Line 57: "Currently, most LSMs are not able to represent a direct vegetation control on surface exchange". Do vegetation parameters not influence transpiration within LSMs and therefore exert a control on the land-atmosphere exchange of water?*

We added more explanation.

2ti. *Line 62: "... by LSMs helps to shed light on the known discrepancies". "helps" should probably be "would help". Which discrepancies are being referred to here?*

We added more explanation.

28. *Line 66i: "... that can only be executed for a limited set of models". This needs clarification – is this due to time constraints within the study, or is there some other characteristic of certain models that exclude them from such studies?*

We added further explanation.

*31. Line 71: What is meant by "different patterns" and "possible misrepresentations of the observations"? This could likely be stated more clearly.*

We changed to "What are the mechanics behind modeled temporal patterns in vegetation dynamics and occurring misfits to the observations?"

*32. Line 74: This statement is superfluous and could either be moved or removed.*

Was removed

*34. Line 80: The data from Trabucco and Zomer (2018) should be referred to as the CGIAR-CSI Global-Aridity and Global-PET Database.*

done

*35. Line 87: "… we assumed [the sites] to be neither very predictable nor very unpredictable in total …". This statement is unclear in both its meaning and implications.*

Was removed

*36. Figure 1: Reference the IGBP classification scheme used, as well as where this data was obtained for each site (e.g., from the FLUXNET website, or within the site netCDFs). I believe some of the sites in this study have 19 years of data available – why does the scale have an upper limit of 18? Furthermore, a continuous color scale could likely be used here to allow differentiation between adjacent numbers (and clarify which color each label belongs to).*

Fig. 1 reference was added. The longest time series within the selected sites was 199ti-2014 which is 18 years. We refrained from using a continuous color scale because the observational time can only be full years and, thus, the sites have distinct duration classes. We adapted the resolution of the scale.

*38. Line 95: Should the soil water content have an abbreviation introduced here in the same vein as the fluxes?*

We wanted to use as less abbreviations as possible to assure readability especially in Results and Discussions section. Additionally, the part where soil water content is referred to is limited which made it unnecessary to use an abbreviation.

*40. Line 99: Rather than "We adopted the same procedure …", simply say that you also excluded timesteps where L <= 0.*

done

*41. Line 101: How long did the gap-filled periods need to be to be excluded from the model performance analysis? How might this affect the results from the analysis?*

We excluded gap-filled periods that were longer than one month from model evaluation.

*42. Line 105: Why were only these four quality flags allowed? Other studies indicate any value less than 64 is usable (e.g., Fang et al., 2012; Ma and Liang, 2022).*

More information on quality flags of MODIS was added. Working with MODIS is challenging and you would need to select quality flags for each site separately. After revisiting MODIS flags as part of the revision, I decided to exclude flag 65 but pro 73, 81 and 97 especially in order to keep some values during winter (see Fig. R1). Thus, we also redid the model runs, affected by those adaptations.

[Figure]

Figure R1: MODIS data points for FI-Hyy when using all quality flags (top left), quality flags less than ti4 as recommended by Fang et al. (2012) (top right), specific quality flags in our selection (bottom left) and only "high quality" flags (bottom right).

*44. Line 107: Cite papers to provide assurance that using a Savgol filter is suitable for this purpose (e.g., Cao et al., 2018; Chen et al., 2004; Huang et al., 2021).*

Done

*45. Line 110: "Each following year … that specific year." is ambiguous in terms of which year is being used as the forcing.*

*46. Line 112: "If LAI values for more than one month were not available" – did these months need to be consecutive?*

Yes, this refers to consecutive months. Information added.

*48. Table 1 and throughout the manuscript: Is there any potential for providing shorter labels for the "Terms" from this table? While they are descriptive which is useful, they can be unwieldy in length.*

This is true. But descriptive simulation settings also prevent confusion, so we prefer to keep these.

*49. Line 117: "Due to the use … from smoothing. Gaps were left as they were". This needs more explanation. Why are the same data from earlier (with QC flags of 48 and 65) not used here?*

Good point. I needed to keep data with other QC for creating the climatology to ensure that each month got a value for that site, which was a challenge especially for the single-year simulations. But since the trustability of these data points is low, they were left out for the temporal higher resolved evaluation.

*53. Line 131: Explain what "IFS cycle "CY46R1" means.*

Changed "cycle" into "version" to make clear that this is the version name/number.

*54. Line 131: How were the IGBP PFT classes from FLUXNET2015 mapped onto the 19 vegetation types within ECLand? If the two classification schemes do not exactly match (or say, the ECLand types are taken from default data based on lat/lon of the site), were any tests performed to confirm that the classes aligned in a suitable manner?*

We initialized the model with the closest possible fit to the on-site conditions without changing any parameters. For ECLand, we had a global setup that we used based on ERA5. We did not adapt the parameters in the global setup. Additional tests were not conducted. We incorporated the vegetation classes for ECLand into Table 2 to enhance clarity. However, it is true that tile fractioning in ECLand into high and low vegetation in the default setup might bias the evaluation with point measurements that belong to only one of these vegetation types. So, I checked and found that the vegetation type in the initial files from the global setup did not match the FLUXNET classification for some sites. Thus, we adapted that and repeated the experiment. Substantial changes of the resulting model performance occurred for some sites (e.g. AT-Neu, BE-Lon) but the general outcome of the study was not affected.

*57. Line 135: Does "respective cover" refer to the fractional cover of each of the two vegetation heights?*

Yes, "respective cover" means the fraction of each vegetation type on the grid cell. Wording was changed to "fractional cover".

*59. Line 149: See the comment for line 131 regarding the PFT classes for ECLand. The same holds here for Noah-MP.*

Same as above in comment 54, here as well, we assured model setup to fit as closely as possible the on-site conditions.

*62. Line 152: "Stomatal resistance is controlled by photosynthesis". Is this statement true for Noah-MP? I would think it is more of a coupled relationship where stomatal resistance can also be controlled by e.g. vapour pressure deficit which in turn would decrease the level of photosynthesis by limiting the available intercellular CO2.*

Yes, as it is explained in Niu et al. (2011) section 4.2. Changed to "Among others, stomatal resistance is predominantly controlled by photosynthesis (Niu et al., 2011)…".

*63. Line 160: While the height of flux tower would ideally be dependent on the vegetation height, this isn't always true – towers can be situated within the canopy or many meters above it.*

To be honest, the information that the tower ends in the vegetation canopy, is new to me especially since the aim of the network is to capture fluxes of the respective vegetation type. I checked the given measurement heights of the sites I chose, and two of them might look suspicious but I don't know the vegetation on-site.

*64. Line 161: How deep was the uppermost soil layer?*

The uppermost soil layer for Noah-MP is 0.1 m and for ECLand 0.07m. We added this information.

*65. Line 162: Was the ten-year spin up sufficient to reach a steady state in each model? What variables were used to check that such a steady state had been reached?*

Steady state was not checked quantitatively but qualitatively.

*66. Line 164: What "initial data" was taken from ERA-5? Why was the FLUXNET2015 data not suitable?*

The initial files contain information on soil, tile fractioning, LAI climatology, state variables at the time of the start of the simulation. For the latter, I could have replaced them by measured values from Fluxnet2015 but the values adapt during the spin-up anyways.

*67. Line 172: Why are the soil data averaged from neighbouring cells?*

This was initially done to have better representation of the general conditions surrounding the tower. We checked and it would have been not necessary since soil type within the

neighboring cells was the same as for the grid cell of interest, so we removed this averaging process in the revision and only work the grid cell where the tower is located.

*69. Line 177: What aspect of the model meant that the temperate vegetation did not regrow? Is the requirement to have green vegetation fraction set to 1 a detriment to the results or their interpretability?*

This is about the minimum green vegetation fraction. The formulation was misleading in the manuscript. Setting the minimum green vegetation fraction to 1% assures that there is still a small amount of biomass after the winter, which is essential for the model to generate spring growth. Without any biomass (i.e. leaves) there would be no location for photosynthesis to take place (zero leaf area * high potential photosynthesis still is zero). Changed to "Minimum green vegetation fraction was set to 1 % to ensure that not the whole vegetation cover dies during winter which would hinder temperate short vegetation from growing in spring."

*71. Line 180: Is the implicit temperature time scheme for the surface temperature?*

Yes, vegetation canopy surface temperature is meant and added.

*72. Table 2: Why is the IGBP class OSH in this table when it does not feature in Figure 1? How was the initial LAI changed for sites in the Southern Hemisphere, namely the Australian sites? With DBF LAI set to 0.0, it seems clear that the Noah-MP initial LAI is based on a year startng on 1 January in the Northern Hemisphere, yet the Australian sites are potentially at the peak of their growing season in January.*

72. Tab. 2 removed OSH from the table.

*75. Equation 1: I do not think this is needed as Pearson's correlation coefficient is widely used and available in many programming languages.*

True, was removed

*78. Equation 2: This is justified as avoiding division by 0 or values very close to zero. However, this doesn't strictly follow from the formulation of the divisor. If the observations have very low variance or are very biased towards 0 values, then conceivably the mean minus the minimum could still be a very small number.*

Agreed. But division by 0 is successfully avoided and in case of very low variance, the numerator is also small, which results in reasonable values of the relative bias and not like 3000%.

*82. Line 204: Was an abbreviation considered for the normalized standard deviation to improve the ease of referring to it, and bring it in-line with the other two metrics which are referenced with a single letter?*

Done

*88. Line 220: "a bunch of" should be avoided – what was the actual number of sites?*

This part was reformulated

*89. Line 225: This is confusing wording as it is difficult to determine whether the authors are referring to all the sites, one specific site, or just some sites.*

This part was reformulated

*90. Line 228: "whether the predicted LAI fit better … was random". Was the difference in performance random with respect to the sites' classes or aridity? It might be better to say that there was no clear rela6onship between the difference in performance and the site characteris6cs explored.*

Agreed, we changed "random" to "ambiguous"

*91. Line 231 and throughout the manuscript: Define which classes are meant by "short or sparse vegetation types".*

Good point, thanks. We added in brackets which vegetation types we refer to.

*93. Line 234: Comparing the static simulations across Figure 2 (and the other Taylor diagrams) is difficult as the end of the arrows are hard to locate, especially with respect to the site that the arrows represent when the arrows are clustered.*

I changed the symbols to be a bit smaller so that they have less overlap, in hope that helps. But, we refrained from using thicker arrows because that could also be counter-productive by blocking the symbols.

*94. Line 236: "With activated vegetation dynamics … in the Taylor diagram". This statement implies that performance improves for all sites and all LAI forcings, yet*

This statement refers to using MODIS climatology as LAI forcing. We added this only by referring to the respective figure parts because otherwise it would be too repetitive in this paragraph where we are already talking about MODIS climatology.

*97. Line 242: Figures 3d-f are referenced but Figure 3 does not have sub-labels.*

Sublabels added.

*99. Line 249: The total LAI is disaggregated into high and low, yet the model is run with either only high or low vegetation. How does this impact results, as one can imagine this results in lower LAI than truth.*

One grid cell in ECLand is split into high and low vegetation fraction with their LAI values. Meaning, one spot cannot have both vegetation types (there is no layering). The resulting LAI is then the weighted mean according the high and low vegetation fraction. Thus, if a grid

cell in our setup is only a high vegetation type, resulting LAI is higher than for a grid cell that has also a low vegetation type fraction. This is the closest we can get to the footprint of flux tower observations.

*101. Figure 2: Why does the arrow of US-Var extend outside of the plot domain in Figure 3c? How does US-GLE in Figure 3c have no change in either standard deviation or correlation yet an extremely large change in the relative bias? This would imply a simple shin in magnitude in the LAI output which would be striking if caused by the switch to dynamic vegetation.*

Correlation coefficient for static Noah-MP LAI for US-Var was negative, so the arrow starts there. US-GLE has no change in standard deviation or correlation is because, as an evergreen forest, default LAI is constant throughout the year and, thus, correlation coefficient cannot be calculated.

*102. Figure 2 and others: How were the aridity brackets defined for the color coding?*

Fig. 2 Quantitative limits of the aridity classes based on Ashaolu & Ilorin (2018). Added that information and citation to the caption of Figure 2.

*103. Figure 3: Since other figures are in color, I would suggest this figure also use color to differentiate between static and dynamic to help visually distinguish between the two.*

done

*104. Line 270 and throughout the manuscript: More consistency in the used definition of "model performance" would be good and can be aided by being more explicit about the metric currently being discussed.*

The term model performance aims to include all the metrics that are discussed here. "Lower model performance" in general means that the majority of the metrics show deterioration. In other cases, the explicit metric is referred to.

*105. Line 279: More explanation of how the opposing NEE biases indicate differences in respiration estimates is required.*

We added more information about the model structure to the Methods section

*107. Figure 4: Why is AU-Stp outside of the plot area for Figure 4a? The axes should be extended so that the site falls within the plot area.*

done

*110. Line 304: "Findings from this study … modelling carbon and energy fluxes". This is a strong statement about the impact of this work and requires more discussion to support*

*it. Which processes within ECLand has this study iden6fied as requiring further development? How has the study provided evidence for how these processes should be improved within the model?*

More insights and evidence for this are given in section 3.3. This statement is now in section "Implications".

*112. Line 309: "Stevens … with static ECLand" is not needed as the precise results from these other studies are not critical to the discussion. Instead, these two papers could simply be cited to support the prior statement that the results are comparable to other studies. If the exact values from the prior studies are mentioned, then it would be good to also state the same metric values from this study explicitly.*

Unfortunately, using the same metrics from other studies is not possible since they basically do not have them.

*113. Line 312: Without being explicit about the methodology used for the literature review, it is also not necessary to state that no other studies were found. This is semi-implicit (if even required) in only having the two above cita6ons.*

Ok, was deleted.

*114. Line 318: "… points appeared to have the largest arrows". This statement could be supported quantitatively with a measure of length for the arrows, equivalent to the degree of performance difference between the two model runs.*

Results and Discussions are now separated in the revision (as stated before), the "new" results part includes more quantities.

*115. Line 322: "… no trend regarding vegetation type or site aridity can be seen …". Were any statistical tests to check for a trend performed here? If not, then changing "trend" to "rela6onship" might be preferable.*

Was changed to "relationship"

*116. Line 329: It would be good to explore the low EF / high NEE performance in forests in more detail. What processes are likely to be responsible for this mismatch in model performance? It is findings such as these that, with further discussion, would support the statement from my comment 115.*

We considered this and decided against intensifying the discussion on NEE-EF relationship. Clearly, more consideration of the processes would be good. However, we already have an in-depth discussion about how LAI and turbulent fluxes are related in the models, and we believe that many of those points already touch on this relation.

*117. Line 335: I would include the soil moisture plots in the appendix.*

done

*118. Line 340: Slightly more explanation for how the underes6ma6on of GPP/LAI could cause the poor EF performance is needed. A few words on the linking mechanisms would be sufficient.*

In the EF calculation, LE is in the numerator. Thus, lowering LE reduces EF. When LAI is modelled to be small, the transpiration can only be low (water balance) or, equally, LE is smaller (energy balance) because less energy is used to transpire water. With an underestimation of LAI also the EF representation deteriorates. We added "because the energy fraction that is used for transpiration is underestimated"

*122. Line 344: Activating vegetation dynamics in Noah-MP arguably had more than "a small impact" on LE and EF for certain sites. AU-DaS noticeably has significant displacement in position between static and dynamic runs in Figures 5d and 6d. Similarly, comparing the position between static runs for AU-DaS with default and MODIS LAI, there is clearly a large difference in model performance.*

Yes, true, there were some exceptions. We mentioned some.

*123. Line 346: I would suggest more information on the possible causes of disagreement between Ma et al. and this study. Why might different results have been reached? I would also replace "already concluded" with "found", otherwise it reads as if the authors are dismissing their own results!*

Disagreement between statements from Ma et al. (2017) and our study is low. Only the bias values vary. One possible reason might be the differing timescale for the evaluation (daily vs. monthly/annual). Added "…which could be due to the differing timescales for model evaluation".

*125. Line 357: To what measurements does "optimal values" refer?*

"Optimal values" refers to the values for soil characteristics in look-up tables. Rephrased to "…optimal values for soil parameters are still uncertain".

*126. Line 361: This statement is not clear.*

Was rephrased

*128. Line 363: "Surprisingly, the model quality of those actually closely related variables was independent". This sentence needs work. What does model quality mean? Which variables are considered closely related, and why? How does this affect the confidence in the results?*

Agreed, the phrasing was ambiguous and needed more explanation, which was now added. But the finding, that model performance in LAI and in LE seems to be independent of each

other although LE values depend on LAI values, does not affect the confidence in the results since it **is** one of the results.

*132. Figure 7: Keep the x axes constant across the nine plots. This ensures that comparison between the plots is easy and does not mask the differences in performance. This is also the case for the other figures – where the point of subfigures is to allow comparison between them, ensure that all scales are consistent as this provides ease of comparison. It is also necessary to describe what each element of the boxplots represents.*

Thank you! Indeed in this Figure the x axes were not consistent, and this was changed now. For other figures, I could not relate that criticism. The Figure caption is extended.

*133. Figure 8: Which LAI is used for the models in this figure? I would suggest less transparency for the MAM and SON points, or just use different colors. Moving the range indicators outside of the ploting area would ensure they do not cover points on the plots.*

LAI in Figure 8e-p is the model output. Added to the caption.

*135. Line 407: I would suggest changing the units that GPP is reported in such that the values do not need to be reported at so many decimal places.*

Good suggestion, done.

*136. Line 408: Are the MODIS values of LAI varying between 1 and 7 realistic? It should be clear whether the authors believe the LAI or GPP is the most likely reason for the two variables to not align.*

This comment likely refers to the tropical site (GF-Guy) selected for the Figure 8. We were also concerned about this large range of LAI values. However, we handled data quality as careful as possible and used only days with high standard quality flags. Some information on that can be found in the new "Limitations" part.

*139. Line 419: This sentence makes it unclear which sites were being discussed previously – the start of the paragraph indicates that all of the sites are being discussed but then here it is stated that similar behaviour is seen at a specific site.*

Was rephrased.

*145. Line 444: Is the 11% in the model? If so, how does this compare to observations?*

Yes, the 11% of assimilation in the model goes into dark respiration. Checking that ratio for the observations is tricky, and we think beyond the already very detailed analysis presented here.

*148. Line 458: "However, an evaluation of the representativeness of key variables like lead area index or net ecosystem exchange is rarely done". I would agree this is frequently a part of model evaluation, and therefore needs to be more specifically worded to accurately infer what the authors are saying.*

We added "…on high temporal resolution" to be more specific.

*157. Tables A1 – A6: What are the column headings? How do they relate to the different model runs?*

The headings were edited.

*158. Table A6: This appears to disagree with the statement made at line 334 that model performance for soil moisture is insensitive to LAI forcing or vegetation dynamics. Assuming that each column in Table A6 is one of the different model runs, then sites such as US-SRM (relative bias of ECLand varies from 314% to 552%) appear to have quite varying performance, even if it is consistently poor.*

There are some exceptions but for the majority of the sites, soil moisture did not respond to changes in LAI. We just mentioned the majority since the overall manuscript is long and provides much information anyways and we tried not to overload the Results section with small details.

References

Ashaolu, Eniola & Iroye, Kayode. (2018). Rainfall and potential evapotranspiration patterns and their effects on climatic water balance in the Western Lithoral Hydrological Zone of Nigeria. Ruhuna Journal of Science. 9. 92-11ti. 10.4038/rjs.v9i2.45.

Souhail Boussetta , Gianpaolo Balsamo , Anton Beljaars , Tomas Kral & Lionel Jarlan (2013) Impact of a satellite-derived leaf area index monthly climatology in a global numerical weather prediction model, International Journal of Remote Sensing, 34:9-10, 3520-3542, DOI: 10.1080/014311ti1.2012.71ti543

Fang, H., Wei, S., and Liang, S.: Validation of MODIS and CYCLOPES LAI products using global field measurement data, Remote Sensing of Environment, 119, 43–54, https://doi.org/10.101ti/j.rse.2011.12.00ti, 2012.

Ma, N., Niu, G.-Y., Xia, Y., Cai, X., Zhang, Y., Ma, Y., & Fang, Y. (2017). A systematic evaluation of Noah-MP in simulating land-atmosphere energy, water, and carbon exchanges over the continental United States. Journal of Geophysical Research: Atmospheres, 122, 12,245–12,2ti8. https://doi.org/10.1002/ 2017JD027597

Niu, G.-Y., et al. (2011), The community Noah land surface model with multiparameterization options (Noah-MP): 1. Model description and evaluation with local-scale measurements, J. Geophys. Res., 11ti, D12109, doi:10.1029/2010JD015139.

Nogueira, M., Albergel, C., Boussetta, S., Johannsen, F., Trigo, I. F., Ermida, S. L., Martins, J. P. A., and Dutra, E.: Role of vegetation in representing land surface temperature in the CHTESSEL (CY45R1) and SURFEX-ISBA (v8.1) land surface models: a case study over Iberia, Geosci. Model Dev., 13, 3975–3993, https://doi.org/10.5194/gmd-13-3975-2020, 2020.

Response on Review 2

First of all, I want to thank you for reviewing the work of me and my co-authors. You have spent a lot of effort and emphasized many details. Your criticism is important for improving this work and publication and your hints are useful for reaching this goal. In the following, I will go through and respond to your comments.

General comments (reviewer comment in italic, response in plain text, adaptations in blue):

- *Do you have two sets of experiments, where 1) you test the impact of LAI datasets on simulated LAI and carbon/water/energy fluxes and 2) where you 'simply' switch on/off the dynamic vegetation module? If so why is 1) not part of your research questions in the introduction? In parts of your manuscripts it reads like you prescribe LAI but it is dynamically simulated at the same time which I don't understand (see for e.g. caption Fig. 7)?*

    Our main focus was testing whether switching on dynamic vegetation in the models enhance their performance regarding the target variables. We changed the LAI source in order to find out whether this more site-related information as initial input "helps" the model in their prediction of LAI and NEE. However, we did not aim for doing data assimilation since there are many investigations published on that. Prescribing the LAI is always for initializing the models independent of dynamic vegetation. In the model simulations themselves, this prescribed LAI is only used in the model runs with static vegetation but is in any case part of the initial input of the models independently whether it is used or not. We handled the terminology and the descriptions within the manuscript more carefully.

- *Where is LAI as an input driver coming from in the LUTs? How does it differ between the experiments (LUT vs your LAI? Is LUT also based on MODIS?) Not everyone is necessarily familiar with the look up tables of the specific models chosen for this study so it'd be good to clarify this.*

    The default climatology in the initial file (what I refer as LUT LAI) of ECLand is already based on MODIS values. A time span from 2000 to 2008 and disaggregation of the gridded values for LAI was used to create that climatology (Boussetta et al., 2013). LAI values in the look-up tables of Noah-MP are defined for the plant functional types (PFTs). These values are also based on MODIS observations which were disaggregated to the different PFTs on each observational grid cell (Oleson et al., 2010). I could not find any information from which time span these values were taken or how individual LAI climatology within one PFT were merged. In the default setup, this LUT LAI was used. For the other setups, those values in the LUT were replaced by "our" LAI values from MODIS.

- *The section where you compare LAI across your experiments almost seemed a bit circular to me, and I would suggest to reduce the emphasis on LAI and focus more on the simulated fluxes where you can avoid the interdependence of input and output LAI during evaluation (and this is also appropriate given the title of the manuscript).*

*Alternative remotely sensed LAI datasets are available, although this comparison of course also would be a bit unfair.*

The LAI from MODIS used for model input and model evaluation is not identical. Model input is a LAI climatology on monthly basis resulting from multi-year average MODIS values. Model evaluation is done with the daily MODIS values which are 8-day means. For the static runs, this comparison provides the information whether an incorporation of more site-specific climatology results in higher representativeness of local LAI evolution. For the dynamic simulations, comparing modeled LAI with daily MODIS values is used to examine whether the models are able to capture inter- and intra-annual LAI dynamics. However, we could show that even with the same source of the data the dynamic simulations are not fitting the observations. We provided more details on the MODIS LAI data and highlighted the differences between data used for input and for evaluation (L190-220).

- *Towards the end of your results/discussion section you describe what's happening in the model and how this explains some of your model results which is great! I think it could help your manuscript if in the methods the model descriptions had more detail too for the relevant processes.*

   We extended explanation of model processes concerning dynamic vegetation and added important equations to the appendix.

- *Throughout your manuscript it would help readability if you had specific experiment names that are consistently italic (or any other distinct formatting) like you attempted in L158.*

   Thank you for the advice. Done.

- *Split the Results and Discussion section - the way it is written now, it is a bit of a back and forth and hard to follow.*

   Splitting Results and Discussion section is done.

- *I was also a bit surprised about your model selection? Why did you choose a model that couldn't provide all necessary outputs for all simulations you conducted?*

   We chose ECLand and Noah-MP because both models can be and are widely used for coupling them as LSMs with established climate projection models. Although Noah-MP provides no GPP and NEE output for the static runs, it still is interesting to look at the LAI-GPP relationship within the model that we did for Figure 8. Nonetheless, we tried to be more careful with absolute statements and adjusted the abstract and the discussion according to that.

- *Why did you initialize your model simulations differently (ECLand vs Noah-MP)?*

In principle, both models are initialized with the same values, fitting as close as possible to the on-site conditions. However, there are some technical differences in the model initialization which we described. We added that information (L162-163).

- *You report that dynamically simulated vegetation leads to a lower model performance, at least in LAI. One thing I wondered is whether your model simulates the 'right' vegetation type for each site you considered (or do you define the vegetation type that is simulated)? You also point out multiple times how forests tend to show better model performance than shorter vegetation types, but you don't offer any explanations why that might be the case?*

For sure for Noah-MP, since there is only one vegetation type on the grid cell. For ECLand we would have needed to adapt vegetation to be either high or low vegetation in the initial file. We did this now but it didn't change much. Regarding the model performance of short vegetation types, we could interpret a bit more. One possible reason could be that forests have less dynamics in their productivity compared to crops, grasslands or shrubs. Surely, trees have dynamics in their leaf mass and photosynthesis rate dependent on environmental impacts but, in general, have access to deeper water resources and intrinsic carbon storages to at least partly overcome water scarcity. Shorter vegetation types cannot cope for limitations in this way, resulting in higher relative temporal variations.

Specific comments (for brevity here we only give the responses to the specific comments. The line numbers refer to those in the original submission):
- L8: "More detailed information" refers to the on-site LAI. Changed to "…regarding leaf area…"
- L13-14: We didn't aim to pinpoint poor model performance of the models themselves for single or all selected sites. The question of this investigation was whether model performance can be improved by dynamic vegetation. Since this is not the case, we provide possible explanations and misrepresentation of the relationship between LAI and GPP is the major one we figured here. Reformulated.
- L21: done
- L24-25: reformulated
- L26-36: No, we don't want to come up with new evaluation schemes. Rather, we want to motivate why we did an analysis with only a few models and presenting absolute performance metrics, which seems like "a step back" in comparison with multi-model evaluations.
- L29-31: changed "them" to "models" to make it more clearly
- L34-35: added "…since all methods could have a poor individual model performance but there will still be one that performs best, resulting in the highest rank" to explain the disadvantage of only presenting normalized metrics.
- L40: Of course, there will be always uncertainty in measured data but I am sure that they accounted for that. Haughton et al. (2016) were investigating reasons for the outcomes of the PLUMBER study that simple empirical models outperformed most LSMs. They excluded systematic bias of flux tower data, time scaling effects and lack of energy conservation in the data as potential causes and stated that processes within or parameterization of the LSMs themselves need to cause poor performance. Slightly reformulated.

- L41: What we were trying to say with that sentence was that benchmarking or ranking models alone is no suitable tool to identify specific causes for a mismatch between model predictions and observations. Achieving this, needs a deeper look into single models and their individual performance. Reformulated.
- L45: This is a topic sentence and several works are cited in the following sentences.
- L46ff: Since one of the motivations to have dynamic vegetation in LSMs is to better predict impacts of water scarcity and drought events on the vegetation, we found it would be valid to argue that current implemented and used LSMs struggle in making prediction that fit observations in these conditions. However, we have shortened this paragraph a bit.
- L66-67: it's especially interesting because both models are still under development especially with respect to freshly introduced modules like that for vegetation dynamics.
- L80: Aridity describes water deficit in long-term climate conditions. Following this, it is the ratio of annual potential evapotranspiration to annual precipitation, leading to larger values of this ratio meaning larger aridity of the site. However, the ratio in this dataset was calculated the other way around which is less intuitive. Also, since we planned to filter the sites on a logarithmic scale, inverting delivered the opportunity to include more semi-arid and arid sites which differ much between each other with respect to seasonality and vegetation dynamics while humid sites are more even. We explained a bit more, referring to the aridity index that was created by Budyko & Miller.
- L83: It is not a common threshold but we needed to come up with one within our filter algorithm. The aridity indices of wetter sites are closer to each other than for drier sites. In order to not overrepresent dry sites within selection by using a threshold in absolute values of the aridity index, we transformed the aridity index to a logarithmic scale, creating almost linearity of the aridity index scale. Added an info on logarithmic scale.
- L87: Haughton et al. (2018a) found out that, within the FLUXNET sites, drier sites (higher aridity index) and wetter sites with low temperature span tend to have higher predictability, meaning that it is easier to achieve good model performance. With our selection by aridity, we assured that we do not only include sites with high or low predictability.
- L97: Filling missing precipitation data with zeros is the only option that is possible. We don't know whether it rained that hour or day. However, the model input cannot handle missing values.
- L97: I do not know how common the Kalman filter is. Gapfilling for the TERENO site "Hohes Holz" was done with it. FLUXNET usually uses Marginal Distribution Sampling which is a really complicated algorithm to implement and to run. Additionally, it cannot fill large gaps as well, which can be seen in time series data from some of the FLUXNET sites.
- L97-98: The ERA5 product I retrieved had 0.1° spatial and 1h temporal resolution and, thus, really helped with filling the gaps. The limit of 3h in using the Kalman filter evolved from the observation that the filter tends to overestimate the values when gaps are longer. This information is included now.
- L100-101: Longer periods where data is filled with Marginal Distribution Sampling (MDS) within the FLUXNET dataset can be seen visually because variability is

unnaturally low (see Fig. R1). "Longer" in this respect means at least a month. Information added.

[Figure]

Figure R1: NEE time series for FLUXNET site AU-Stp, exemplarily. Gap-filling from MDS can be identified visually, in this case from January to August 2008 and from March to May 2009. These intervals were left out for model evaluation.

- L104: Temporal resolution is 8 days. There are different MODIS datasets available. The one we used, MOD15A2H, has a spatial resolution of 500 m. Information added.
- L105: Done.
- L106-107: Creating the LAI climatology means to calculate the average annual LAI cycle. For a 10-year time series of MODIS LAI, it might happen that some months have 30 values while other months have only 3 by selecting the same quality flags (i.e. 0 and 32) (see Fig. R2 for the site FI-Hyy). For example, a tropical site is covered by ITC cloudiness nearly at the same time of each year. Thus, all the values during that time have a lower quality flag and would be excluded. It happened that we were left with some months without any LAI information, so we included a larger set of flagged data points for the climatology.

[Figure]

**Figure R2:** MODIS data points for FI-Hyy when using all quality flags (top left), quality flags less than 64 as recommended by Fang et al. (2012) (top right), specific quality flags in our selection (bottom left) and only "high quality" flags (bottom right). Using only the last category of flagged data (or even in this case all data points with QC<64) would have left us with no information on LAI during winter.

- L112: See my explanation in point 2 of the general comments.
- Table 1: Rephrased it and added information on the timespan.
- L127: "Under-development" means that these models (and here especially the modules that incorporate dynamic vegetation to the models) are constantly extended and improved. We removed that word.
- L129 (refers to 119): From MODIS documentation, every value flagged higher than 0 has some uncertainty or limitation. QC=32 might have the least uncertainty after that. So, I limited the data used to these two flags for the single-day comparisons, to lower uncertainty in the data. Smoothing was not applied to capture also potential low or high peaks in the LAI data. Additionally, due to unequal gaps within the LAI time series of QC=0 and 32, the smoothing could distort the LAI values.
- L133-134: Yes, at maximum. It could be even one or none.
- L134-142 + L150-156: We tried to leave model description as short as possible. However, more details on LAI-related processes might help and we included them. Extended process explanation and added important equations to the appendix.
- L164: Both models have two types of input: Initial files (with initial values for some variables to start with) for model setup and time series files with meteorological data for model runs. The initial files contain variables like vegetation type, deep soil temperature, soil layering, soil type, initial soil moisture, vegetation cover fraction

and initial LAI value or LAI climatology which are not all present in the FLUXNET data. But the variables included in the initial files differ for both models that is the reason why it sounds like different setups but they are not. For ECLand, these initial data files were prepared for a global setup already and we could make use of that. For Noah-MP, no such setup existed and we created the initial files by ourselves by using the information we had. After model initialization followed the spin-up phase so that these initial values were not used any longer and became overwritten by actually modelled values.

- L166: Clustering the vegetation into high or low vegetation type does not depend on vegetation height but on the vegetation type on-site. Forests in any case are high vegetation no matter how big the trees actually are.
- L169: added.
- L172: The reason for the initial conditions of the two models being different is only because these initial files look different for both models and require slightly different set of variables. Apart from that, we kept initial conditions as close to each other and as close to on-site conditions as possible.
- L173: True. we checked whether soil type would change when including 8 neighboring cells compared to just 4 or even only the grid cell with the tower on it, but this was not the case. So, we now stated and took the soil type of the grid cell of the tower location.
- Table 2: done
- L183: Yes, daily averages or sums (depend on variable). added
- L188: In principle, the relationship between observed and modelled values of a target variable is expected to be linear.
- L189-190: done
- L191-193: A "normal" relative bias was not applicable since our target variables (i.e. LE, H, NEE and GPP) have values that vary around zero. This results in relative biases that are not only really large partly but also difficult to interpret (e.g. reaching 3000% of relative bias but not because the model estimate is far away from observation but rather because the mean value is close to zero). By subtracting the minimum, the distribution is shifted to positive values only, with the minimum value being zero. As a result, the relative bias really contains an information on how much the estimates deviate from the mean since the reference system is the codomain of the variable. This works independently of the distance between $x_{min}$ and $x_{mean}$.
- L199-200: Yes, exactly.
- L199-207: I tried to explain the elasticity in more detail. Unfortunately, I found no publication from environmental sciences that use the same metric, only from economics. Tried different explanation now.
- L213: All symbols that are in the Taylor plots. Changed to "symbols".
- L217-218: shifted to and extended in discussion section
- L222: Here, we refer only to literature because Stevens et al. (2020) also replaced LUT LAI by MODIS LAI and compared model results.
- L224: For the dynamic simulations, LAI is not prescribed but still part of the initial files. It is expectable that LAI predictions for the dynamic simulations are independent of the initial input. However, dynamic ECLand still incorporates prescribed LAI to 5% (RLAIINT=0.95 was defined by the developers' team to be fully dynamic).

- L225: changed the sentence into "For ECLand, this was also the case for many sites but not necessarily for all, e.g. AT-Neu and AU-How" by adding examples.
- L226: done
- L226: Increased variance in comparison with static ECLand simulations. Added "…compared to static simulations…".
- L227: We could not find any tendencies regarding aridity or vegetation type to have positive or negative shift in relative bias. Replaced by "was ambiguous"
- L231: No, I did not. Sparse vegetation are savannas and shrublands because they have no closed canopy surface. Added "Especially short (GRA+CRO) or sparse (SAV+WSA) vegetation types…"
- L243-244: done
- L255: Model performance metrics for the MODIS single-year simulations are in the appendix tables. But they are not part of the Taylor plots.
- Fig 2: added.
- Fig 3: Static Noah-MP produces no output for NEE and GPP which is according to model structure. Thus, only the values for the dynamic runs can be presented here. Colors changed, axes extended.
- L281-282: Although Noah-MP provides no GPP and NEE output for the static runs, it still is interesting to look at the LAI-GPP relationship within the model that we did for Figure 8. Apart from that, we still can look at LAI, latent heat flux and soil moisture.
- L284: For most of the sites, GPP was overestimated with dynamic Noah-MP, but relative bias was predominantly small for forests (Tab. A3). We reformulated that conclusion and tried to be more precise.
- L289: yes
- L291: On-site LAI and MODIS LAI were linearly correlated. MODIS LAI might be biased for some sites, but so might be on-site measured LAI due to technical limitations (scatter correction, saturation effect…). During development of the dynamic vegetation modules, a tuning of the parameter sets was done but not to MODIS LAI as target variable. However, mismatch between MODIS and on-site LAI is reflected in lower performance of NEE and GPP of the static ECLand simulations. The reason is unclear: It could be that on-site LAI does not reflect actual LAI but it could also be that calculations of GPP in relation to LAI do not match reality (similar to what we have shown in Fig. 8). For the dynamic ECLand runs, differences between MODIS and on-site LAI play only a minimal role since 95% of the LAI calculations come from dynamically predicted LAI and NEE and GPP predictions are even fully dynamically predicted.
- L306-307: The performance is not different for the dynamic simulations. But for the static runs, it is. Thus, we recommended here to use static simulations with MODIS climatology forcing. However, I just recognized by reading your comment that we can only recommend this for ECLand since for static Noah-MP we don't know the actual performance regarding NEE and GPP. Shifted to "Implications".
- L329-330: It might be that carbon and water transport processes are coupled not tightly enough. With NEE estimates fitting well, the photosynthetic activity also is good captured by the model. The demand of water by the photosynthesis might be underestimated by the model and, leading to less transpiration and, thus, also to a lower fraction of energy that is used for latent heat transport. Additionally, downward $CO_2$ transport and upward water transport through turbulent fluxes

occurs in the same eddies which is not captured by the model. These are just some ideas on that so far. We refrained from intensifying the discussion on that.

- L335: done
- L351: Vegetation needs water for photosynthesis which stems from the soil. Thus, more photosynthetically active biomass extracts more water from the soil and, otherwise, less soil water restricts maximum plant productivity and biomass build-up.
- L354-359: Yes, you are right. The reason for unaffected soil moisture to vegetation dynamics still remains unclear. Referring to the point before it could be due to the implemented interaction of carbon and water processes. First, the potential photosynthetic activity in dependence of leaf area and radiative conditions is calculated. Then, the limitation factor of extractable water is estimated according to available soil water and roots. Lastly, the photosynthetic activity is adapted to that restriction and transpiration rate adapted to conductivity and atmospheric conditions. As a result, the only included path is that soil moisture impacts photosynthetic activity and biomass build-up. But there is no feedback that more biomass needs/loses more water that will be taken from the soil because photosynthetic activity relates only to the carbon fluxes but not to the water fluxes. We added this explanation to the text.
- Fig. 7: Sorry that footnote was there by accident.
- L389-396: I added LAI-GPP and LAI-NEE elasticity for ECLand in Figure 7 but excluded NEE-LE and NEE-SM instead. Other studies also found a linear relationship between LAI and GPP but with large variability. Some sites might be exceptions from the linearity (IT-Ren) where LAI-GPP relationship appears to be a non-linear saturation function.
- Fig. 8: We added description of the arrows to the figure caption. Since the most probable in the observations LAI-GPP relation is a linear one, Pearson correlation coefficient is the statistical basis of this linear regression and also the measure for the relationships from the model output. We cannot compare different kinds of correlation coefficients.
- L403: done
- L409-410: this is now part of the section "limitations".
- L454-455: We cannot replace "real" by "observed" because we are not referring to any measured values here. The reality this sentence is referring to is the fact that trees do not immediately lose their leaves when they are faced to a few days of suboptimal conditions for photosynthesis. Replaced by "realistic".
- L461-462: deleted.
- L467: Compared to forests that are more resistant and resilient for e.g. water scarcity, short vegetation more dynamically and more instantly responds to environmental limitations for its growth. Thus, firstly, assuming the same LAI cycle for each year and, secondly, assuming a constant LAI values over a whole month as in the static model simulations, do not represent reality. Our expectation was that modelling vegetation dynamically would cope for that variability and, as a result, yield in better performance of observed ecosystem fluxes.
- L480-481: Other models have processes implemented differently. So, there is no chance in directly transferring results and conclusions from these two models to others.

Literature

Best, M. J., Abramowitz, G., Johnson, H. R., Pitman, A. J., Balsamo, G., Boone, A., Cuntz, M., Decharme, B., Dirmeyer, P. A., Dong, J., Ek, M., Guo, Z., Haverd, V., van den Hurk, B. J. J., Nearing, G. S., Pak, B., Peters-Lidard, C., Santanello, J. A., Stevens, L., and Vuichard, N.: The Plumbing of Land Surface Models: Benchmarking Model Performance, Journal of Hydrometeorology, 16, 1425–1442, https://doi.org/10.1175/jhm-d-14-0158.1, 2015

Souhail Boussetta , Gianpaolo Balsamo , Anton Beljaars , Tomas Kral & Lionel Jarlan (2013) Impact of a satellite-derived leaf area index monthly climatology in a global numerical weather prediction model, International Journal of Remote Sensing, 34:9-10, 3520-3542, DOI: 10.1080/01431161.2012.716543

Haughton, N., Abramowitz, G., Pitman, A. J., Or, D., Best, M. J., Johnson, H. R., Balsamo, G., Boone, A., Cuntz, M., Decharme, B., Dirmeyer, P. A., Dong, J., Ek, M., Guo, Z., Haverd, V., van den Hurk, B. J. J., Nearing, G. S., Pak, B., Santanello, J. A., J., Stevens, L. E., and Vuichard, N.: The plumbing of land surface models: is poor performance a result of methodology or data quality?, J Hydrometeorol, 17, 1705–1723, https://doi.org/10.1175/JHM-D-15-0171.1, 2016.

Haughton, N., Abramowitz, G., De Kauwe, M. G., and Pitman, A. J.: Does predictability of fluxes vary between FLUXNET sites?, Biogeosciences, 15, 4495–4513, https://doi.org/10.5194/bg-15-4495-2018, 2018a.

Oleson, K. W., Lawrence, D. M., Bonan, G. B., Flanner, M. G., Kluzek, E., Lawrence, P. J., … Zeng, X. (2010). Technical Description of version 4.0 of the Community Land Model (CLM) (No. NCAR/TN-478+STR). University Corporation for Atmospheric Research. doi:10.5065/D6FB50WZ

Stevens, D., Miranda, P. M. A., Orth, R., Boussetta, S., Balsamo, G., and Dutra, E.: Sensitivity of Surface Fluxes in the ECMWF Land Surface Model to the Remotely Sensed Leaf Area Index and Root Distribution: Evaluation with Tower Flux Data, Atmosphere, 11, https://doi.org/10.3390/atmos11121362, 2020.

Response on Review 3

First of all, I want to thank you for reviewing our work. We tried our best to implement your suggestions and we think that the manuscript has improved substantially.

In the following, I will go through and respond to your comments. (reviewers comments in italic, answers in normal font, changes in blue)

*I understand that the authors use some existing Plant Functional Type in the respective models and do not tune PFT parameters to fit the dominant or average plant traits of each fluxnet site. However, the authors still perform comparisons and evaluate the models against observational point-level ecosystem-specific data. Is this the case? I find this slightly inconsistent, since plant functional types conceptually are mostly meant to be used in larger-scale simulations representing the average characteristics of a vegetation category, rather than being compared with site-level data.*

Indeed, the footprint of a flux tower observation has a smaller area than the grid cell of LSMs. Nonetheless, comparison of model output against point-level observations such as those from FLUXNET is a common way to perform model evaluation, especially, since most LSMs are able to be used on a wide range of spatial scales. FLUXNET delivers the basis for such a model evaluation on smaller scales.
PFTs are a concept to simplify the parameterization of vegetation that is expected to respond in a similar way to its environment. As a result, they should be transferable and representative for all subtypes of vegetation that are merged into one PFT. If not, they would have been separate groups. We set the vegetation of the considered grid cell within the model to the PFT that fit closest to the on-site conditions to minimize potential mismatches in parameterization.

*If we have a model whose parameters are not set to fit the observations (see above), then why would we necessarily expect that switching on a dynamic vegetation module (which is also unparameterized) should increase model performance? One could argue that solely switching the environmental dependency of LAI on should justify this expectation, but isn't the environmental dependency of LAI on average also embedded in a prescribed climatology by definition?*

Just to clarify, the dynamic vegetation modules are not unparameterized, only not additionally calibrated for that specific site. We agree that switching on dynamic vegetation introduces environmental dependency of LAI. If the model is allowed to adapt the vegetation (and its productivity and LAI) to environmental conditions, it can be expected that model predictions are closer to the observations compared to simulations with static vegetation. The climatology contains long-term seasonality of LAI. It represents the average temporal pattern of LAI that is adapted to the long-term mean environmental conditions. Intra- and interannual variability as a result of environmental conditions cannot be included into the LAI climatology. To cope for this, dynamic vegetation modules were implemented.

*I deeply appreciate that the authors are extensively discussing past research, their interpretations, and their results in full detail, which increases transparency, but the manuscript is overall difficult to read. It would help a lot if the authors split the results from*

*the discussion points. The manuscript would also need further proof-reading, since one can easily still find mistakes scattered across the text.*

Thank you for this positive feedback. In the revision Results and Discussion have been separated. We also carefully proofread the manuscript.

*The results suggest that model performance "regarding latent heat flux or soil moisture is independent of how LAI is represented" (Line 380). This is very counter-intuitive, and one would wonder whether this is the case because LAI is equally badly represented in all cases.*

True, we were also surprised by this result. The answer is within section 3.3. This independency does not mean that the predicted values of latent heat flux do not change when LAI is changing. Rather, the model performance in latent heat flux does not change. Together, this means that the mismatch between modeled and observed latent heat flux might be small or large (depending on the site) but is in almost the same extent small or large with a different LAI representation, resulting in the same model performance. We tried to phrase this even more explicitly in the revision.

*Given that LAI dynamics are the main focus of this study, it is important that the authors describe in more detail the LAI modules of the two models. They start doing so in Line 420 and discuss allocation, senescence etc., but they need to do this in a comprehensive manner in the method section, and not scattered across the text.*

The Model description was extended and now explains all processes involved in modeling vegetation dynamically and important equations can be found in the appendix now.

*I think it is really important to show in the appendix panels with mean annual LAI, GPP, and NEE climatologies for every site, showing prescribed and model-predicted LAI. This would help a lot the interested reader understand the dynamics at play.*

Good point. It would be definitely interesting to show some time series plots. However, this is not possible for all sites and all model simulations without stretching the appendix a lot or risking that figures are too small to detect graphs. Also, selecting only single sites would add no additional information compared to the graphs of Fig. 8. We therefore decided not to follow this suggestion.

*The selction criteria for the fluxnet sites used seem arbitrary. Why do the authors drop sites of roughly similar aridity index? If anything, including more sites would increase the robustness of their results.*

Representative site selection is an important issue and we gave it detailed consideration when designing the analysis. When looking at the global distribution of FLUXNET sites, many of them are located in temperate climate on the Northern Hemisphere. Including all sites with more than 5 years would create an overrepresentation of regions with high density in sites, resulting in an imbalance of PFT-aridity combinations for model evaluation with especially (semi-)arid short vegetation being underrepresented. Thus, we needed some sort of filter algorithm to avoid that overall model performance is either shifted towards better or worse performance due to this imbalance. Also, some sites had to be removed due to

low-quality in soil moisture data. Unfortunately, there are not enough sites available to create a second set of the same structure (e.g. a representative coverage of climate zones and PFTs), as some aridity-PFT combinations are really rare. We are aware that such a second set would be helpful for strengthening and reproducing our findings. We now explained in more detail how the systematic site selection was done, and adapted Figure 1 in accordance with the model PFTs.

*In my understanding, in the switched-on dynamic vegetation runs LAI is freely estimated by the model and not constrained by some prescribed climatology. Is this indeed the case? From the results (e.g., Fig. 2) it feels as if there are different types of dynamic runs – one for each of the different possible LAI forcings. How is this the case? In my understanding, somehow these prescribed climatologies are still used to "initiate" the dynamic runs. What does that exactly mean? If this is indeed the case, still LAI is free to evolve, so why do we end up with different model performance in every run, just because initial LAI conditions have been different? If this is not the case and LAI climatologies are somehow fed also into the dynamic runs, then how does it make sense to validate these results against the forcing LAI climatology itself? The authors need to try and clarify their setup more.*

Even when the models run with vegetation dynamics, the LAI climatology is still part of the initial files, independently whether it is used or not. For Noah-MP, these climatological values are not used for the dynamic setup. This is why we end up with the same model performance for all dynamic Noah-MP runs. In ECLand, the vegetation is not totally dynamic. For instance, with a fraction of 5% the prescribed LAI still merges into the LAI estimate for that simulation day (defined by Souhail Boussetta, used as a dynamic ECLand setup). Thus, also model performance of ECLand differs a bit depending on LAI climatology source. The two columns in the Figure should show in which direction and how much model performance shifts for the dynamic simulations compared to only relying on the prescribed LAI climatology of the simulations with static vegetation. We have now added more information on how LAI was used to the Methods section (Section 2.3)

*Line 155-156: Was it not technically possible to maintain GPP and NEE estimation in the static Noah-MP runs, despite prescribing LAI? Or is there some other reason?*

Static Noah-MP does not calculate GPP and NEE (missing values in the output file). This relates to model structure. LAI for the next time step is already known, so there is no need to estimate assimilation by photosynthesis or allocation to plant tissues. So yes, technically, this is not a model output.

*Line 165: It is slightly unclear in which occasion ERA5 data are used*

The default initial files were based on ERA5 dataset. We added more specific information.

*Line 255: I would suggest that the reviewers show these results in supplementary material to ensure transparency (or don't mention at all)*

Ok. The Taylor plot for soil moisture now is in the appendix.

---

## Referee Report (RR1)

**Second round review of "Does dynamically modeled leaf area improve predictions of land surface water and carbon fluxes? – Insights into dynamic vegetation modules" by Westermann et al.**

In their major revisions, Westermann et al. have substantially improved this manuscript which investigates the model performance of two land surface models (LSMs) when utilising static vs dynamic vegetation representations. The initial round of reviews resulted in many reviewer comments to be addressed and, in general, these have been resolved. The authors must be commended for their effort in the responses. The manuscript is cleaner and the message clearer.

Despite the improvements realised in the first round of author revisions, I still have some reservations regarding the implementation of this study. These are detailed further below along with some additional minor technical comments. As such, I regretfully recommend another round of revision before the manuscript can be published.

**General Comments**

1. I still find the choice of Noah-MP in the study perplexing. One of the main thrusts of the manuscript, as evidenced by the title, is gaining insights into the carbon fluxes of eddy-covariance sites when using LSMs with different vegetation initial conditions and representations. However, since Noah-MP does not produce GPP/NEE output when run statically, a sizeable amount of potential data/analysis is lacking. For example, Figures 3, 4, 7, 8, and A1 and Tables A2 and A3 highlight this conspicuous unavailability of information, where effectively only a single LSM is being used to assess the static/dynamic vegetation influence on carbon fluxes.
   In their response to this issue in the first round of reviews, the authors justify their choice of Noah-MP and ECLand as "both models can be and are widely used for coupling them as LSMs with established climate projection models." This is true for many LSMs and as such is not particularly convincing. However, I understand that access to resources and expertise for particular models is a challenge and the authors likely used the two models they could reliably run. In light of this, I believe there are two potential avenues for addressing my concerns, and I would be very interested if either is acceptable to the authors. Firstly, the manuscript could be amended to focus more on the water fluxes since all model runs output the relevant data for these. The carbon fluxes would then be additional/supplementary information and the missing outputs would not be as detrimental to the message. This option would result in a change of title

and reordering of the manuscript to address latent heat, evaporative fraction and soil moisture first.

Alternatively, the manuscript could instead focus on specifically assessing ECLand, and use Noah-MP as a benchmark. Again, this would reduce the impact of the missing carbon fluxes from the static Noah-MP. As well as changes to the text to focus more on ECLand, this would require the figures to be rearranged to emphasise the ECLand results.

Neither of these options would necessitate additional model runs, and I believe would minimise the amount of work required from the authors to mitigate the apparent issues of missing NEE/GPP data. Of course, it is possible that this aspect of the manuscript has been considered appropriately addressed in the author response by the other two reviewers, in which case I am happy to defer to the majority.

2. In a similar vein to the above, I am also unconvinced by the authors' response to the comments on their site selection criteria. I agree with the authors that the FLUXNET dataset is biased both geographically and relative to vegetation types. However, I would argue that individual sites can exhibit unique behaviour and may not be representative of their "aridity - PFT" class, and that this is far more likely to influence results negatively than by (further) introducing the well-known and understood biases from the heavily skewed location of FLUXNET sites. There are ways to reduce the impact of the PFT-aridity class imbalance that would exist if more sites were selected. For instance, contributions to the aggregate model performance metrics could be weighted by PFT-aridity class size. In fact, most figures and results in the paper are discussed on a site-by-site basis (e.g., the Taylor diagrams) and so the class imbalance would not present issues here.

3. The comparison of LAI output from the static model runs to MODIS LAI is another aspect that continues to trouble me regarding the applicability of the results from this study. For the dynamic runs, it is understandable as the vegetation evolves away from the initial inputs and so the use of MODIS data as an initial condition avoids any circular comparisons. However, under static runs, it would appear to me that the study is simply comparing the same MODIS data at different levels of time aggregation with zero influence from the models. Hence I struggle to derive any messages from this analysis for future model development.

**Technical Comments**

4. Line 1: "the surface" is not clear. It would be better to use "the Earth's surface" or "the land surface", for example.

5. Line 3: "some of these models". Some of which models? "Some land surface models" or similar would clarify this.

6. Line 7: add an "and" between "the FLUXNET2015 dataset" and "the MODIS leaf area".

7. Line 11: I would argue that latent heat flux is both a vegetation- and hydrology- related variable and therefore this sentence is not quite correct.

8. Line 93: "we assumed them to be neither very predictable nor very unpredictable in total" - I think this needs clarification.

9. Line 103: Include the citation to the FLUXNET website.

10. Line 108: spelling of "tends".

11. Line 110: Cite the Climate Data Store properly.

12. Line 180: It should be "Noah-MP" not "the Noah-MP" and "a global soil grid" not "the global soil grid".

13. Line 182: Initialising all LAI values based on Table 1 for model runs starting on January 1$^{st}$ would misrepresent the four Australian sites and may cause model performance issues.

14. Line 189: spelling of "therefore"

15. Line 189: "Other options were used as their defaults" is not clear. I recommend "All other settings used default configurations" or similar.

16. Line 257 - 259: I would suggest explicitly explaining how this follows from the figures e.g., that the symbols are in the same location / there are no arrows.

17. Figure 2: This was raised in the first-round reviews, but the arrows should not extend beyond the plot area. I understand that this is because the normalised standard deviation of the static run falls outside the plot limits, but this is not acceptable. The axes must be extended such that the arrows are fully located within the plot area.

18. Line 269: It is unconventional to refer to the performance in the static runs as having "increased" when these runs are the 'baseline', and this data is plotted as the beginning of arrows.

19. Line 285: Is it possible to quantify the increase in arrow length in Figure 4? This would be preferable to the qualitative use of "longer arrows" in this instance.

20. Line 294: Similar to comment 19, can the "scattered more closely" be quantified?

21. Line 296: Is it not the case that the sites with the "best" performance depends on how one prioritises the metrics, or are the 12 sites mentioned the best performing across all three metrics used?

22. Figure 3: Caption uses "die" rather than "the".

23. Line 314: Why does CH-Oe2 exhibit such improved performance compared to all other sites? Does this have any lessons for model development?

24. Line 330: "Despite being low" - to what is this referring?
25. Line 351: "less uncertainty" is not the terminology to be used here. Maybe "weaker"?
26. Line 355: This reads as though it is introducing Figure 8 but Figure 8 has already been discussed in the previous paragraph.
27. Line 366: I would check the literature for examples of MODIS LAI being inaccurate for tropical sites.
28. Figure 8: I suggest rearranging the panels so that the facets are, in descending order, "Observation", "Static ECLand", "Dynamic ECLand", "Dynamic Noah-MP". This keeps the ECLand runs next to each other, but also places the dynamic runs adjacent to each other as well.
29. Figure 8: The caption refers to the fitted linear regression models as "applied as additional information" which does not read correctly. I would delete "as additional information" in this instance.
30. Line 380: I would argue that comparison of modelled and observed fluxes on a daily basis is performed more frequently than "rarely".
31. Line 410: What is the Noah-MP Crop module?
32. Line 420: In which scenario was a frequent reset of LAI applied to ECLand as compared to the other studies? I do not follow where this was applied and had no effect?
33. Line 425: "low predictive efficiencies" is unusual terminology. Maybe "low predictability" or "low predictive power"?
34. Line 441: "inclusively LAI" should be "inclusive of LAI".
35. Line 565: Why do the authors suggest "alternative remote sensing LAI products"? No other products were tested in this study and such products may not perform well.
36. Line 568: Haughton et al. (2016) explicitly checked the sites used in PLUMBER for observational errors. This study shares only three sites with the PLUMBER study and therefore this citation likely shouldn't be used in support here.
37. Line 590: "Using alternative input ... but this needs to be evaluated in more detail". Is this not what was investigated in this manuscript? What additional detail should be checked in any future studies? What are the authors' suggestions to model developers?
38. Code and Data Availability: I suggest including the datasets in the bibliography and citing them here properly rather than the current use of weblinks. Proper citations would ensure reproducibility by containing additional information such as dataset versions, date accessed, etc.

---

## Referee Report (RR2)

**"Does dynamically modeled lead area improve predictions of land surface water and carbon fluxes? - Insights into dynamic vegetation modules" Review – 3rd Round**

Following the 2nd round of revisions, the authors have amended the submitted manuscript further, with the most substantial change being the implementation of carbon cycle simulation for static Noah-MP. The static runs now produce GPP and NEE output which significantly fleshes out the analysis, allowing comparison between ECLand and Noah-MP. This addition supports the inclusion of Noah-MP in the manuscript, which previously felt extraneous as mentioned in my 2nd review.

My major concerns regarding the manuscript have been addressed. There are still some minor technical comments – once addressed, I believe the manuscript is ready for publication.

I appreciate and commend the authors' efforts in addressing the concerns of myself and the other reviewers. I hope they agree with me that the manuscript in its current state is much improved through their work and the review process was worthwhile if long!

**Comments**

Line 165: "but resetting all variables that would be dynamically predicted within the same function". I do not think this is clear about the steps taken. I assume this is meant to clarify that, in the static runs, it is only GPP and NEE that are modified and that the rest of the model remains in its static configuration?

Line 195: "The Noah-MP simulations were done with soil parametrization from look-up tables, Ball-Berry stomatal resistance approach with using matric potential". Please correct the grammar in this sentence to make the implementation clear.

Table 2: This detail is good but should likely be in Supplementary Information. If it is included for Noah-MP, consistency would suggest the same information be supplied in the same table for ECLand.

Line 317: "lowered from –32 % - +69 % to –28 % - +42 %" is confusing to read with the hyphens and minus signs. I would suggest something like "lowered from between –32 % and +69 % to between –28 % and +42 %".

---

## Author Response (AR2)

**Response on Referee #1's second round review**

First of all, we want to thank reviewer 1 for accomplishing a second round of reviewing this work of me and my co-authors. In the following, I will go through and respond to your comments. Please note that reviewer comments are in italic, our responses in normal font and an explanation of changes/adaptations made by us in blue font. In case, your suggested terms/phrases were incorporated without further adaptations, I refrained from copying them again to this document.

**General comments**

1. *I still find the choice of Noah-MP in the study perplexing. One of the main thrusts of the manuscript, as evidenced by the title, is gaining insights into the carbon fluxes of eddy-covariance sites when using LSMs with different vegetation initial conditions and representations. However, since Noah-MP does not produce GPP/NEE output when run statically, a sizeable amount of potential data/analysis is lacking. For example, Figures 3, 4, 7, 8, and A1 and Tables A2 and A3 highlight this conspicuous unavailability of information, where effectively only a single LSM is being used to assess the static/dynamic vegetation influence on carbon fluxes.*
*In their response to this issue in the first round of reviews, the authors justify their choice of Noah-MP and ECLand as "both models can be and are widely used for coupling them as LSMs with established climate projection models." This is true for many LSMs and as such is not particularly convincing. However, I understand that access to resources and expertise for particular models is a challenge and the authors likely used the two models they could reliably run. In light of this, I believe there are two potential avenues for addressing my concerns, and I would be very interested if either is acceptable to the authors. Firstly, the manuscript could be amended to focus more on the water fluxes since all model runs output the relevant data for these. The carbon fluxes would then be additional/supplementary information and the missing outputs would not be as detrimental to the message. This option would result in a change of title and reordering of the manuscript to address latent heat, evaporative fraction and soil moisture first. Alternatively, the manuscript could instead focus on specifically assessing ECLand, and use Noah-MP as a benchmark. Again, this would reduce the impact of the missing carbon fluxes from the static Noah-MP. As well as changes to the text to focus more on ECLand, this would require the figures to be rearranged to emphasise the ECLand results. Neither of these options would necessitate additional model runs, and I believe would minimise the amount of work required from the authors to mitigate the apparent issues of missing NEE/GPP data. Of course, it is possible that this aspect of the manuscript has been considered appropriately addressed in the author response by the other two reviewers, in which case I am happy to defer to the majority.*

Thank you for providing further ideas for dealing with the limitation that Noah-MP is only partly suitable for answering our research questions. We now decided to keep the intention and the research questions of the manuscript as they are but, instead, changed the source code of Noah-MP a bit so that GPP and NEE are calculated also for the static simulations. As a result, GPP is calculated from the current conditions for photosynthesis and NEE results from that GPP and an estimate of respiration of the current available biomass. All other variables that usually would have been dynamically computed within the same function were reset to their values before calling the

function. We checked whether our adaptation had an impact on the predictions of latent heat flux and could not find any.

We explained that adaptation in the method section (L160-161), added some results for model performance of NEE and GPP (L295-308), included the LAI-NEE and LAI-GPP relationships for Noah-MP in Figure 7 showing the elasticity and changed the paragraph L434-448 in the discussion section a bit according to the new results.

2. *In a similar vein to the above, I am also unconvinced by the authors' response to the comments on their site selection criteria. I agree with the authors that the FLUXNET dataset is biased both geographically and relative to vegetation types. However, I would argue that individual sites can exhibit unique behaviour and may not be representative of their "aridity - PFT" class, and that this is far more likely to influence results negatively than by (further) introducing the well-known and understood biases from the heavily skewed location of FLUXNET sites. There are ways to reduce the impact of the PFT-aridity class imbalance that would exist if more sites were selected. For instance, contributions to the aggregate model performance metrics could be weighted by PFT-aridity class size. In fact, most figures and results in the paper are discussed on a site-by-site basis (e.g., the Taylor diagrams) and so the class imbalance would not present issues here.*

Yes, this issue is specifically addressed in Haughton et al. (2018a), a publication we are aware of and taken into consideration. Those authors found no relationship between the uniqueness of a site and its PFT or data record length and, thus, both characteristics do not legitimize site selection. However, at least for dryness (aridity) there is some evidence that it determines the uniqueness of a site. Furthermore, they also said that it is more important to communicate how and why a certain site selection was done, which we did. To address the comment, we now explicitly checked the predictability measures from Haughton et al. (2018a) for our selected sites. They had a RMSE uniqueness between 0.03 and 0.41 for NEE, 0.04 and 0.43 for sensible heat flux, and 0.04 and 0.30 for latent heat flux. Note that this is in a comparable range for all three variables, avoiding that one of our target variables could be more predictable on average. Second, this confirms that our dataset includes sites with medium to high predictability. Indeed, our selection ended up not including sites with low predictability which is against the suggestion by the authors to include the whole spectrum of predictability in a model evaluation study. At the same time, our selection strategy yielded sites that are less unique and, thus, more representative. In other words, site uniqueness did not affect our results. Indeed, we argue that less predictable sites would not allow enhanced insights: since dynamic vegetation modules of the two LSMs we tested here are already challenged to predict vegetation processes even for (very) predictable sites, they are unexpected to improve for sites with less predictability which yields no additional information. To check whether site predictability affects the sensitivity of the model performance to vegetation dynamics, we tested whether greater changes in model performance for e.g. NEE coincides with less predictability and whether more unique sites show greater sensitivity to activating vegetation dynamics, and found no evidence for both. We argue that our site selection captures "adequate diversity of site characteristics" for the purpose our investigation as proposed in Haughton et al., (2018a).

3. *The comparison of LAI output from the static model runs to MODIS LAI is another aspect that continues to trouble me regarding the applicability of the results from this study. For the dynamic runs, it is understandable as the vegetation evolves away from the initial inputs and so the use of MODIS data as an initial condition avoids any circular comparisons. However, under static runs, it would appear to me that the study is simply comparing the same MODIS data at different levels of time aggregation with zero influence from the models. Hence I struggle to derive any messages from this analysis for future model development.*

Agreed, it was expected for the static runs with MODIS climatology to have good performance (since it's the same data but averaged and with coarser temporal resolution for input)
However, we also had the expectation that dynamic runs should better capture single values from MODIS resulting in higher performance, even higher than with MODIS climatology which is not the case. In order to show that, we needed the assessment of the static runs with MODIS climatology as a basis for comparison. We cannot state that dynamic vegetation modules perform worse than a more site-specific climatology if we wouldn't have done this test. We have explicitly stated in the paper why we use those runs (L219-222). Thus, the message is: best performance can be achieved by using a more site-specific climatology, since dynamic vegetation modules still need improvements before they will do better than that.

**Technical comments**
4. *Line 1: "the surface" is not clear. It would be better to use "the Earth's surface" or "the land surface", for example.*

Changed to "the land surface" (L1).

5. *Line 3: "some of these models". Some of which models? "Some land surface models" or similar would clarify this.*

Changed to "Some land surface models" (L3).

6. *Line 7: add an "and" between "the FLUXNET2015 dataset" and "the MODIS leaf area".*

Done (L8).

7. *Line 11: I would argue that latent heat flux is both a vegetation- and hydrology-related variable and therefore this sentence is not quite correct.*

Agreed, latent heat flux is largely determined by the vegetation but, in the end, it is a flux of water and, thus, part of the water cycle.
Changed to "the performance regarding variables of the carbon and water cycle was unrelated for both models" to make it more clearly (L11).

8. *Line 93: "we assumed them to be neither very predictable nor very unpredictable in total" - I think this needs clarification.*

This relates to the predictability of FLUXNET sites found by Haughton et al. (2018a). Please see the answer to comment 2.

The sentence is now reformulated to "We were left with 24 sites, covering a wide range of site characteristics as recommended by Haughton et al. (2018a) including aridity, vegetation types and observation periods (Fig. 1)" (L92-94) and left out the reference to site predictability since this is not the focus of our site selection.

9. *Line 103: Include the citation to the FLUXNET website.*

   Citation added as "(fluxnet.org, 2020)" inclusively bibliography entry (L102).

10. *Line 108: spelling of "tends".*

    Done (L107).

11. *Line 110: Cite the Climate Data Store properly.*

    Citation added as "(Copernicus, 2018)" inclusively bibliography entry (L109).

12. *Line 180: It should be "Noah-MP" not "the Noah-MP" and "a global soil grid" not "the global soil grid".*

    Done (L181).

13. *Line 182: Initialising all LAI values based on Table 1 for model runs starting on January 1st would misrepresent the four Australian sites and may cause model performance issues.*

    Absolutely right.
    Added those values in brackets and explained in the table caption as "The values in brackets for Noah-MP initial LAI refer to sites on the Southern Hemisphere due to shifted seasons" (Tab. 1).

14. *Line 189: spelling of "therefore"*

    Part of the sentence was changed to "…with using matric potential limitation" (L190).

15. *Line 189: "Other options were used as their defaults" is not clear. I recommend "All other settings used default configurations" or similar.*

    All other options for the Noah-MP simulations are now explicitly listed in Table 2.

16. *Line 257 - 259: I would suggest explicitly explaining how this follows from the figures e.g., that the symbols are in the same location / there are no arrows.*

    Added "…since the symbols in Figure 2 c+d have the same positions" to that sentence (L258-259).

17. *Figure 2: This was raised in the first-round reviews, but the arrows should not extend beyond the plot area. I understand that this is because the normalised standard deviation of the static run falls outside the plot limits, but this is not acceptable. The axes must be extended such that the arrows are fully located within the plot area.*

    Axis for correlation coefficient extended in the new Figure 2. The same appeared for Figure 4, Figure A1 and Figure A2.

18. *Line 269: It is unconventional to refer to the performance in the static runs as having "increased" when these runs are the 'baseline', and this data is plotted as the beginning of arrows.*

    The "increased" here refers to the change in model performance by using MODIS climatology compared to using default climatology.
    Changed to "…using *MODIS climatology* instead of *default climatology* in static simulations…" (L268).

19. *Line 285: Is it possible to quantify the increase in arrow length in Figure 4? This would be preferable to the qualitative use of "longer arrows" in this instance.*

    In principle, it would be possible to quantify the arrow length since it is approximately the hypotenuse of the change in correlation coefficient and the change in normalized standard deviation. However, this quantity has no interpretable meaning.

20. *Line 294: Similar to comment 19, can the "scattered more closely" be quantified?*

    Quantified as the average deviance from nstd=1. Added "…Noah-MP seemed to capture NEE representations better as the mean deviance from a normalized standard deviation of 1 was 0.33 (ECLand: 0.39) …" (L309-310).

21. *Line 296: Is it not the case that the sites with the "best" performance depends on how one prioritises the metrics, or are the 12 sites mentioned the best performing across all three metrics used?*

    The "best" performing sites here referred to their position in the Taylor diagram and, thus, to the combination of correlation coefficient and normalized standard deviation. Including the relative bias into that ranking would give a different picture.
    Adapted the sentence to "…best sites regarding NEE correlation and variance were forests…" (L311).

22. *Figure 3: Caption uses "die" rather than "the".*

    Done (Fig. 3).

23. *Line 314: Why does CH-Oe2 exhibit such improved performance compared to all other sites? Does this have any lessons for model development?*

The site CH-Oe2 shows improved performance not for all variables but consistently large changes in model performance. After checking the initial file for the ECLand setup, the reason can be found in low LAI values of 0.16 throughout the year for the default climatology. The grid cell with the flux tower in the global setup, where the initial file was taken from, contained no short vegetation. However, during the setup of the default climatology simulations, the vegetation on that grid cell was assigned to be low crops (coverage of the low vegetation = 1) to fit the sites descriptions. This means that the low LAI values were used as climatology since they remained unchanged (which is one of the conditions of the default setup). Consequently, switching on vegetation dynamics or replacing the LAI climatology had a large impact on modelled fluxes and their performance. There is not really a lesson to learn for model development rather for preparing land use datasets for model setup since they might miss some patches of different vegetation types.

Added the sentence "The big exception appeared for CH-Oe2 which was caused by its default LAI climatology that did not fit the vegetation type" (L328-329).

24. *Line 330: "Despite being low" - to what is this referring?*

To the improvement in model performance for soil moisture when activating vegetation dynamics.

Changed to "Some sites showed improvement of soil moisture prediction by activating vegetation dynamics for both models although the improvement was very weak" (L345-346).

25. *Line 351: "less uncertainty" is not the terminology to be used here. Maybe "weaker"?*

What is meant here is the variability of the values and, thus, the scattering around the regression line.

Changed to "…considerable less variability compared to the observations" (L383).

26. *Line 355: This reads as though it is introducing Figure 8 but Figure 8 has already been discussed in the previous paragraph.*

The statements of the first paragraph in section 3.3 were distributed in different locations now. The third sentence (former Line 344-346) became the last sentence of that paragraph (Line 365-367). Instead, explanation of Figure 8 (former Line 351-353) was placed after introducing the Figure (Line 362-365). The sentences "In general, ECLand shows a linear relationship with considerable less uncertainty compared to the observations" and "The slope and intercept of the linear regression is dependent on the choice of static or dynamic vegetation" were moved to Line 382-383. The sentence "In contrast, Noah-MP shows a non-linear relationship with a pronounced hysteresis" (former Line 358) was merged with the sentence "Noah-MP shows a marked hysteresis effect at all sites except the tropical one (Fig. 8 e-h)" from former Line 364 to "Noah-MP shows a non-linear relationship with a pronounced hysteresis effect at all sites except the tropical one (Fig. 8 e-h)" in Line 377. The sentences "This hysteresis is related to the partitioning of GPP to the carbon pools in the plants. Noah-MP uses a non-linear function for allocation of GPP to the leaves that limits the maximum LAI the model can

grow" from former Line 349-350 were moved to Discussion section 4.3 and merged with the sentences in former Line 516-519 (now Line 537-540).

27. *Line 366: I would check the literature for examples of MODIS LAI being inaccurate for tropical sites.*

    Now included in Discussion section 4.4 where limitation of MODIS data is discussed by "Noisy and uncertain LAI data from MODIS for tropical forests was already reported among the literature (Weiss et al., 2007; Garrigues et al., 2008; Xiao et al., 2016; Zhang et al., 2024)" (L595-596).

28. *Figure 8: I suggest rearranging the panels so that the facets are, in descending order, "Observation", "Static ECLand", "Dynamic ECLand", "Dynamic Noah-MP". This keeps the ECLand runs next to each other, but also places the dynamic runs adjacent to each other as well.*

    Done (Fig. 8).

29. *Figure 8: The caption refers to the fitted linear regression models as "applied as additional information" which does not read correctly. I would delete "as additional information" in this instance.*

    Done (Fig. 8).

30. *Line 380: I would argue that comparison of modelled and observed fluxes on a daily basis is performed more frequently than "rarely".*

    Agreed regarding heat fluxes. However, for specifically LAI, NEE and GPP, it really was challenging to find comparable model evaluation studies to discuss with, especially for ECLand. Thus, in my opinion, "rarely" is the right phrase for that.

31. *Line 410: What is the Noah-MP Crop module?*

    Noah-MP-Crop is an extension of Noah-MP developed by Liu et al. (2016) that explicitly considers field management practices for certain crops. The reference is given in the text (L493).

32. *Line 420: In which scenario was a frequent reset of LAI applied to ECLand as compared to the other studies? I do not follow where this was applied and had no effect?*

    We were trying to check our results for plausibility by including studies in our discussion that did data assimilation of LAI within their model runs. In our study, we have not done data assimilation per se, but rather updated the LAI input on an annual basis in the MODIS single-year setup where we could not find any effect of this additional adaptation. I guess the formulation of the statement was misleading here.
    Changed to "…but did not have an effect for the annual resolution applied here" (L432).

33. *Line 425: "low predictive efficiencies" is unusual terminology. Maybe "low predictability" or "low predictive power"?*

    Changed to "low predictive power" (L443).

34. *Line 441: "inclusively LAI" should be "inclusive of LAI".*

    Done (L457).

35. *Line 565: Why do the authors suggest "alternative remote sensing LAI products"? No other products were tested in this study and such products may not perform well.*

    True.
    Changed to "Overall, we recommend using MODIS climatology forcing for static simulations which yielded the best model performances for carbon and water fluxes. This might be valid for other remote sensing LAI products as well but would need to be tested beforehand" (L582-584).

36. *Line 568: Haughton et al. (2016) explicitly checked the sites used in PLUMBER for observational errors. This study shares only three sites with the PLUMBER study and therefore this citation likely shouldn't be used in support here.*

    The methodology of measurements and quality control is standardized within FLUXNET. Thus, uncertainty from measurements should be comparable/transferable from the sites Haughton et al. (2016) considered to others. Additionally, by citing their work we did not want to claim that our sites have the same level of uncertainty but rather giving an evidence that it was already shown that poor model performance likely has other sources than the uncertainty of measured data.
    Changed to "…but Haughton et al. (2016) demonstrated that observational errors, in general, are unlikely to cause poor model performance" (L585-586).

37. *Line 590: "Using alternative input ... but this needs to be evaluated in more detail". Is this not what was investigated in this manuscript? What additional detail should be checked in any future studies? What are the authors' suggestions to model developers?*

    Surely, this was one of the aims in this study but, of course, our investigation cannot absolutely answer this question, since we tested only one remote sensing product and also had on-site LAI measurements from only one site.
    Added "…since we were limited in data sources" (L611). For suggesting future directions, we changed the last statement to "Additionally, they might be a good starting point for a similar intensive investigation with other land surface models or other alternative LAI climatology" (L625-626).

38. *Code and Data Availability: I suggest including the datasets in the bibliography and citing them here properly rather than the current use of weblinks. Proper citations would ensure reproducibility by containing additional information such as dataset versions, date accessed, etc.*

Done (L627-633).

**Response on Referee #2's review**

First of all, we thank Reviewer 2 for giving detailed feedback to our work. The critical remarks have helped us to improve this publication. We are sorry that we omitted to submit a complete list of the response and we have now added this below.

*Thanks a lot for the effort in revising the manuscript. Theoretically of course I'm happy to review the revised manuscript, but unfortunately the response document is set up in a way that makes it very hard to track down what changes have been made. The requested format of the author's response is online where it says*
*'The author's response in case of "minor" or "major" revisions must be submitted as one separate \*.pdf file (indicating page and line numbers) structured in a clear and easy-to-follow sequence: (1) comments from referees/public, (2) author's response, and (3) author's changes in manuscript. [...]'*

*I would greatly appreciate it if this is how the author's response was structured too (see also here [https://www.biogeosciences.net/for_authors/](https://www.biogeosciences.net/for_authors/)).*

*At least for my comments (referee 2) the specific comments I made were left out and 'only' the authors' responses were included so it's not immediately obvious which comments the authors are referring to, and doing this to shorten the overall document is not an argument to make a response letter incomprehensible. I would prefer having a long document and being able to follow the responses and how they are addressed in the revised manuscript.*

*I further noticed that the actual changes in the revised manuscript were not included in the response letter for any of the three reviews. I don't think every new word added needs to be defended, but at the same time responses like 'information added' or similar are not sufficient to indicate which changes were made specifically in the manuscript. Lastly, I'm not clear what the color coding in the response letter means? I assume this is to indicate what changes have been made, but if this could be made more clear that would be very helpful.*

*All reviewers are volunteering their time to review this manuscript and having to go back and forth between three documents (the original review, your responses, and the tracked changes manuscript) to get a sense of how concerns are addressed and which changes were made is quite time consuming and unnecessary given this information could just be combined into one document.*

We acknowledge and understand this comment. We apologize for omitting to send a proper response together with the last revision. We now attached below the point-to-point response to the first review of reviewer 2 below. We also give a short explanation of the formatting.

*I did briefly go through the responses and wanted to point out that I am still not satisfied by the justification for the model set up. This needs to be phrased more carefully. Including a model because 'it is still interesting to look at' isn't exactly a powerful argument - but maybe this is elaborated on in the revised manuscript which I have not read. Another thing I would like to point out is that there seems to be some contradiction: In the introduction the authors pose three research questions that read like they are aiming to generalize their results*

*derived based on a single model (given Noah-MP couldn't be run with static vegetation), but then they argue in the response to reviews that there is 'no chance in directly transferring results and conclusions' from their model to others models. This makes me wonder what the point of the study is if the study set up and results are so model specific that it is not possible to derive any conclusions for land surface models in general?*

To address this comment, we have now included additional runs with Noah-MP that mimic the static vegetation and yield output on carbon fluxes. This was done by using the dynamic version (which produces output for the carbon fluxes), but resetting the LAI every year to the original. We double checked, whether the output corresponds to the original static runs based on the water fluxes, which are output in either version. In this way we are able to show comparable results for both models, even if static Noah-MP is specifically not made to look at carbon fluxes. With regard to model selection: We really made an effort to look into the code in those two widely used models in order to understand whether and how the surprising results came to place. We agree that it would be great to do the same for many more models. But at the same time, this almost forensic investigation cost a great deal of time as not all of the reasons are obvious from the model description. We hope that our insight from those two exemplary models inspires more research in this direction. Notwithstanding that, we have rephrased the research question to specifically address ECLand and Noah-MP in the revised manuscript.

**Response on Referee #2's 1st review**

Thank you for giving this detailed feedback. We think addressing them has improved the manuscript in both revisions. Below we give the missing point-to-point response of the 1st review by reviewer 2. They also include changes made during the second most recent revision.

The reviewer comments are in italic, our responses in normal font and an explanation of changes/adaptations made by us in blue font.

**General comments**

- *Do you have two sets of experiments, where 1) you test the impact of LAI datasets on simulated LAI and carbon/water/energy fluxes and 2) where you 'simply' switch on/off the dynamic vegetation module? If so why is 1) not part of your research questions in the introduction? In parts of your manuscripts it reads like you prescribe LAI but it is dynamically simulated at the same time which I don't understand (see for e.g. caption Fig. 7)?*

  Our main focus was testing whether switching on dynamic vegetation in the models enhance their performance regarding the target variables. We changed the LAI source in order to find out whether this more site-related information as initial input "helps" the model in their prediction of LAI and NEE. However, we did not aim for doing data assimilation since there are many investigations published on that. Information on LAI is always required for initializing the models independently of whether the runs are with dynamic or static vegetation.

We handled the terminology and the descriptions throughout the manuscript more carefully.

- *Where is LAI as an input driver coming from in the LUTs? How does it differ between the experiments (LUT vs your LAI? Is LUT also based on MODIS?) Not everyone is necessarily familiar with the look up tables of the specific models chosen for this study so it'd be good to clarify this.*

The default climatology in the initial file (what I refer as LUT LAI) of ECLand is already based on MODIS values (as mentioned in the manuscript L195-196). A time span from 2000 to 2008 and disaggregation of the gridded values for LAI was used to create that climatology (Boussetta et al., 2013). LAI values in the look-up tables of Noah-MP are defined for the plant functional types (PFTs). I could not find any information from where these values were generated from or which time span these values were taken from or how individual LAI climatology within one PFT was merged. In the default setup, this LUT LAI was used (default climatology). For the other setups, those values in the LUT were replaced by "our" LAI values from MODIS (L197).

- *The section where you compare LAI across your experiments almost seemed a bit circular to me, and I would suggest to reduce the emphasis on LAI and focus more on the simulated fluxes where you can avoid the interdependence of input and output LAI during evaluation (and this is also appropriate given the title of the manuscript). Alternative remotely sensed LAI datasets are available, although this comparison of course also would be a bit unfair.*

The LAI from MODIS used for model input and model evaluation is not identical. Model input is a LAI climatology on monthly basis resulting from multi-year average MODIS values. Model evaluation is done with the daily MODIS values which are 8-day means. For the static runs, this comparison provides the information whether an incorporation of more site-specific climatology results in higher representativeness of local LAI development. For the dynamic simulations, comparing modeled LAI with daily MODIS values is used to examine whether the models are able to capture inter- and intra-annual LAI dynamics. Surprisingly, we found that even with the same source of the data the dynamic simulations are not fitting the observations.
We provided more details on the MODIS LAI data and highlighted the differences between data used for input and for evaluation (L198-222).

- *Towards the end of your results/discussion section you describe what's happening in the model and how this explains some of your model results which is great! I think it could help your manuscript if in the methods the model descriptions had more detail too for the relevant processes.*

Addressing this comment, we extended explanation of model processes concerning dynamic vegetation and added important equations to the appendix for the last revision.

- *Throughout your manuscript it would help readability if you had specific experiment names that are consistently italic (or any other distinct formatting) like you attempted in L158.*

  Thank you for the advice.
  Done.

- *Split the Results and Discussion section - the way it is written now, it is a bit of a back and forth and hard to follow.*

  Addressing this comment, splitting Results and Discussion section was done already in the last revision.

- *I was also a bit surprised about your model selection? Why did you choose a model that couldn't provide all necessary outputs for all simulations you conducted?*

  We chose ECLand and Noah-MP because both models can be and are widely used for coupling them as LSMs with established climate projection models. The fact that Noah-MP is only partly suitable to answer the research questions of this study was brought up more often by now. Thus, we decided to adapt the source code of Noah-MP a bit in a way that NEE and GPP are calculated also for the simulations with static vegetation. This is done by calling the function that usually processes the carbon dynamics but resetting all dynamically calculated variables afterwards to their values before calling that function. This gives estimated for GPP and NEE for the current atmospheric (photosynthetic) conditions of the current available biomass, without changing those. We checked whether our adaptation had an impact on the predictions of latent heat flux and could not find any.
  We explained that adaptation in the method section (L160-161), added some results for model performance of NEE and GPP (L295-308), included the LAI-NEE and LAI-GPP relationships for Noah-MP in Figure 7 showing the elasticity and changed the paragraph L434-448 in the discussion section a bit according to the new results.

- *Why did you initialize your model simulations differently (ECLand vs Noah-MP)?*

  In principle, both models are initialized with the same values, fitting as close as possible to the on-site conditions. However, there are some technical differences in the model initialization which we described.
  We added "The models were set up as closely as possible to the available site information but there are some technical differences in the structure of the model input, i.e. in the initial files" (L165-166).

- *You report that dynamically simulated vegetation leads to a lower model performance, at least in LAI. One thing I wondered is whether your model simulates the 'right' vegetation type for each site you considered (or do you define the vegetation type that is simulated)? You also point out multiple times how forests tend to show better model performance than shorter vegetation types, but you don't offer any explanations why that might be the case?*

For Noah-MP, I agree with you since there is only one vegetation type on the grid cell. For ECLand this requires adapting vegetation to be either high or low vegetation in the initial file. We did not do it originally, but changed it now. It did not affect the results much. Regarding the model performance of short vegetation types, we can interpret a bit more. One possible reason could be that forests have less dynamics in their productivity compared to crops, grasslands or shrubs. Surely, trees have dynamics in their leaf mass and photosynthesis rate dependent on environmental impacts but, commonly, have access to deeper water resources and intrinsic carbon storages to at least partly overcome water scarcity. Shorter vegetation types cannot cope for limitations in this way, resulting in higher relative temporal variations.
This explanation can be found in the discussion section (L443-448).

**Specific comments**

- *L7 Maybe change to 'We compare model results with observed fluxes from the FLUXNET [...] or similar*
*L8 MODIS leaf area index?*
*L8 More detailed information? What does this mean? If the only additional output is LAI, you might as well explicitly state this here but in general I think this is weirdly specific for an abstract the way it is written now*

"More detailed information" refers to the on-site LAI.
The sentence changed to "We compared model results with observations across a range of climate and vegetation types from the FLUXNET2015 dataset and the MODIS leaf area product, and used on-site measured leaf area from an additional site" (L7-8).

- *L13-14 This is not really a reason that explains weak model performance, but just another way to phrase poor model performance! I think also the abstract is not a place for speculation but you should clearly state what the drivers for poor model performance are based on your study*

We did not aim to pinpoint poor model performance of the models themselves for single or all selected sites. The question of this investigation was whether model performance can be improved by dynamic vegetation. Since this is not the case, we provide possible explanations and misrepresentation of the relationship between LAI and GPP is the major one we figured here.
Reformulated into "We show that one potential reason for this could be that the implemented ecosystem processes diverge from the observations in their seasonal patterns and variability" (L13-14).

- *L21 You could already state in the first paragraph where LSMs are used (you do give the CMIP example in L26 but LSMs are also used in meteorology models, reanalysis [...]) before diving into the more specific applications to motivate your study, and from there go to your model validation topic*

Merged into "Traditionally, their main purpose has been to provide a surface component in coupled atmosphere-land models. LSMs are applied in meteorological models, reanalysis products or in the Coupled Model Intercomparison Project (CMIP)" (L21-23).

- *L24-25 This doesn't offer a lot of information. For example, you could mention why more features are added to LSMs to give this sentence more value*

  Changed to "There is active development within the land surface modeling community, with more and more features being added to existing models to make them more realistic (Blyth et al., 2021)" (L25-26).

- *L26-L36 In general this paragraph discusses schemes to evaluate model performance, which I think is a useful topic for your introduction! But I'm not sure what the key point is you're trying to make here. Are you trying to build up to presenting a new evaluation scheme in your paper?*

  No, we don't want to come up with new evaluation schemes. Rather, we want to motivate why we did an analysis with only a few models and presenting absolute performance metrics, which seems like "a step back" in comparison with multi-model evaluations.

- *L28 I suggest 'global, regional, and site scale'*

  Done (L28).

- *L29-31 I'm not sure I understand this. Do you mean that evaluation schemes are compared against each other? Or model - obs comparisons?*

  This was referring to inter-model comparisons.
  Changed "them" to "models" (L30).

- *L34-35 This is also unclear to me. Did they evaluate some LSM ensemble average against statistical methods? I also don't understand how not reporting individual model performance is linked to only having normalized metrics, and also why normalized metrics are not useful. Could you explain this a bit more?*

  Best et al. (2015) compared the model performance of several LSMs and simple statistical models and ranked them based on normalized (relative) statistical metrics. The disadvantage of only presenting normalized metrics is that in any case there will be one model with the highest rank although it could be that this model misrepresents the target variable but the others are doing even worse.
  Changed to "Using this method, Best et al. (2015) reported that simple statistical methods achieve a higher performance in energy partitioning at eddy-covariance sites than any single LSM tested. One limitation of that study is that they did not report metrics of individual model performance, but only normalized ones. This procedure does not allow to judge whether the investigated methods have achieved a (dis-)satisfactory performance, since all methods might have a poor individual model performance" (L32-36).

- *L37 Replace 'had a closer look' with explored, investigated or something similar?*

Replaced by "…more closely explored…" (L38).

- *L40 This is unclear to me. Do you mean that they didn't find an error in the observations they compared the model simulations to?*

Of course, there will be always uncertainty in measured data but I am sure that they accounted for that. Haughton et al. (2016) were investigating reasons for the outcomes of the PLUMBER study that simple empirical models outperformed most LSMs. They excluded systematic bias of flux tower data, time scaling effects and lack of energy conservation in the data as potential causes and stated that processes within or parameterization of the LSMs themselves need to cause poor performance.
Reformulated to "…and not related to errors in the observations" (L41).

- *L41 I get your point but to me this almost reads like model-observation comparisons can't help identify areas of uncertainty at all which is not true (see e.g. Whitley et al., 2016 for one of the many studies that identify reasons for inter-model differences and mismatches with obs)*

What we were trying to say with that sentence was that benchmarking or ranking models alone is no suitable tool to identify specific causes for a mismatch between model predictions and observations. Achieving this, needs a deeper look into single models and their individual performance.
Changed to "Yet, specific reasons for this mismatch, for example over-parameterization, missing processes, calibration issues etc., cannot be identified by benchmarking studies or model rankings alone, but requires further investigation of individual model performance" (L41-43).

- *L45 Could you name a few references to support your statement here?*

This is an introducing topic sentence and several works are cited in the following sentences (L47-48).

- *L46-47 This seems a bit out of place and unnecessary to focus on a single benchmarking dataset that as far as I can tell you're not even using in your study? I think the first two sentences are a bit dis-connected from the rest of the paragraph anyway where now all of a sudden you focus on drought and water-limitation (why?)*

Since one of the motivations to have dynamic vegetation in LSMs is to better predict impacts of water scarcity and drought events on the vegetation, we found it would be valid to argue that current implemented and used LSMs struggle in making prediction that fit observations in these conditions.
We have shortened this paragraph a bit (L47-61).

- *L66-67 Why did you choose ECLand and Noah-MP? Other than them being used by others*

Both models are still under development especially with respect to freshly introduced modules like that for vegetation dynamics (L65).

- *L74-75 I think you can drop this:)*

Deleted.

- *L80 Why did you invert the Aridity Index? I know you took it from a dataset but it'd be good to know how it is calculated*

Aridity describes water deficit in long-term climate conditions. Following this, it is the ratio of annual potential evapotranspiration to annual precipitation, leading to larger values of this ratio meaning larger aridity of the site. However, the ratio in this dataset was calculated the other way around which is less intuitive. Also, since we planned to filter the sites on a logarithmic scale, inverting delivered the opportunity to include more semi-arid and arid sites which differ much between each other with respect to seasonality and vegetation dynamics while humid sites are more even.
We explained a bit more by adding "…and inverted afterwards, bringing it back to the initial definition as the ratio of the long-term mean annual potential evapotranspiration to the long-term mean annual precipitation by Budyko (1974)" (L78-80).

- *L83 Why +- 0.1 log AI? Is this a common threshold?*

It is not a common threshold but we needed to come up with one within our filter algorithm. The aridity indices of wetter sites are closer to each other than for drier sites. In order to not overrepresent dry sites within selection by using a threshold in absolute values of the aridity index, we transformed the aridity index to a logarithmic scale, creating almost linearity of the aridity index scale.
We explained a bit more: "Next, other sites with similar aridity (±0.1 logarithmic aridity index) were dropped to avoid an overrepresentation of some vegetation type-aridity combinations due to heterogeneous site distribution within FLUXNET. We used logarithmic values to create a linear scale of the aridity index, avoiding an overrepresentation of drier sites within the selection process" (L84-87).

- *L87 I'm not sure the predictability of your sites necessarily follows from your selection procedure. But maybe I misunderstand - can you explain this a bit more?*

Haughton et al. (2018a) found out that, within the FLUXNET sites, drier sites (higher aridity index) and wetter sites with low temperature span tend to have higher predictability, meaning that it is easier to achieve good model performance. With our selection by aridity, we assured that we do not only include sites with high or low predictability.
Since we don't want to focus too much on the predictability of the sites, but keeping it in mind, we changed the sentence to "We were left with 24 sites, covering a wide range of site characteristics as recommended by Haughton et al. (2018a) including aridity, vegetation types and observation periods (Fig. 1)" (L92-94).

- *Fig.1 Can you also explain the colorbar in the Figure caption?*

Added "The color scale represents the duration of the available time series in years" to the figure caption (Fig. 1).

- *L97 Why was precipitation set to zero in the gapfilling?*

Filling missing precipitation data with zeros is the only option that is possible. We don't know whether it rained that hour or day. However, the model input cannot handle missing values.

- *L97 Is using a Kalman filter a common procedure for gapfilling? How is gapfilling achieved in FLUXNET?*

I do not know how common the Kalman filter is. Gapfilling for the TERENO site "Hohes Holz" was done with it. FLUXNET usually uses Marginal Distribution Sampling which is a really complicated algorithm to implement and to run. Additionally, it cannot fill large gaps as well, which can be seen in time series data from some of the FLUXNET sites (e.g. Fig. R1).

- *L97-98 ERA5 is relatively coarse if I'm not mistaken, and it'd be worthwhile mentioning the shortcomings and the reason for choosing this reanalysis? I assume you chose it because of the high (3h?) temporal resolution which would also explain the 3h cut-off between Kalman Filter and using ERA5*

The ERA5 product we retrieved had 0.1° spatial and 1h temporal resolution and, thus, really helped with filling the gaps. The limit of 3h in using the Kalman filter evolved from the observation that the filter tends to overestimate the values when gaps are longer. Changed to "For longer gaps, the Kalman procedure tends to overestimate the observations which resulted in offsets at the end of the filling periods. Thus, filling data for these gaps was retrieved from the ERA5 (Hersbach et al., 2020) data product (Copernicus, 2018) with 0.1° spatial and 1 h temporal resolution" (L107-109).

- *L100-101 What's the cut-off for 'longer gap filled' periods?*

Longer periods where data is filled with Marginal Distribution Sampling (MDS) within the FLUXNET dataset can be seen visually because variability is unnaturally low (see Fig. R1). "Longer" in this respect means at least a month.
Added "…we excluded gap filled periods that were longer than one month" (L112).

[Figure]

**Figure R1**: NEE time series for FLUXNET site AU-Stp, exemplarily. Gap-filling from MDS can be identified visually, in this case from January to August 2008 and from March to May 2009. These intervals were left out for model evaluation.

- *L103 You already defined LAI in L59*

  Deleted.

- *L104 What is the temporal and spatial resolution of MODIS LAI?*

  Temporal resolution is 8 days. There are different MODIS datasets available. The one we used, MOD15A2H, has a spatial resolution of 500 m.
  Added "One grid cell of 500 m x 500 m was selected per eddy covariance tower according to the site coordinates and LAI values with temporal resolution of eight days were extracted for the years 2000 to 2014" (L197-199).

- *L105 I know you point to the documentation but could you please explain the choice of quality flags (and what they mean) in the manuscript?*

  We explained usage and quality control of the MODIS LAI in more details now in L197-205.

- *L106-107 I'm not sure I'm following. How does selecting certain quality flags help having the 'same amount of data in a month'?*

  Creating the LAI climatology means to calculate the average annual LAI cycle. For a 10-year time series of MODIS LAI, it might happen that some months have 30 values while other months have only 3 by selecting the same quality flags (i.e. 0 and 32) (see Fig. R2 for the site FI-Hyy). For example, a tropical site is covered by ITC cloudiness nearly at the same time of each year. Thus, all the values during that time have a lower quality flag

and would be excluded. It happened that we were left with some months without any LAI information, so we included a larger set of flagged data points for the climatology. As before, more explanation now can be found in L197-205.

[Figure]

**Figure R2**: MODIS data points for FI-Hyy when using all quality flags (top left), quality flags less than 64 as recommended by Fang et al. (2012) (top right), specific quality flags in our selection (bottom left) and only "high quality" flags (bottom right). Using only the last category of flagged data (or even in this case all data points with QC<64) would have left us with no information on LAI during winter.

- *L112 You quite happily use LUT LAI as if this was a commonly known dataset but personally I have no idea where the data in your look up tables are derived from. Is this look up table derived from MODIS, which years were used for it [...] - you don't have to state every little detail here but perhaps there is a reference with a description of how this look up table was derived?*

See my explanation in point 2 of the general comments.

- *Table 1 The way you described MODIS climatological LAI and MODIS single-year LAI in the table is not super clear. The first one is just a monthly climatology (? based on which years), the second is the first year of the simulation period on monthly timesteps? Could you rephrase this?*

Explanation extended and information on timespan added (now Table 3).

- *L127 'under-development' does this mean the model isn't 'ready' yet?*

"Under-development" means that these models (and here especially the modules that incorporate dynamic vegetation to the models) are constantly extended and improved. We removed that word.

- *L129 (refers to L119) Is the daily resolution really an appropriate reason to choose those quality flags? Why did you avoid smoothing your data for LAI?*

From MODIS documentation, every value flagged higher than 0 has some uncertainty or limitation. QC=32 might have the least uncertainty after that. So, we limited the data used to these two flags for the single-day comparisons, to lower uncertainty in the data. Smoothing was not applied to capture also potential low or high peaks in the LAI data. Additionally, due to unequal gaps within the LAI time series of QC=0 and 32, the smoothing could distort the LAI values.
Added in "Due to the usage of single day values, we solely used data with quality flags 0 (no issues) and 32 (saturated) to lower the uncertainty. Additionally, we refrained from smoothing to avoid an offset of the LAI values and left gaps as they were" (L216-218).

- *L133-134 So only two vegetation types per grid cell are simulated?*

Yes, at maximum. It could be even one or none.

- *L134-142 It's great you explain a bit of the model, but which photosynthesis model is used? It would also be good to know how LAI interacts with GPP/NEE, and also with heat fluxes because these relationships are key to your analysis - but of course I appreciate you can't explain every line of the model code here*
*L150-156 Again, I think it would be interesting to know which photosynthesis scheme Noah-MP employs, and to get a rough idea how LAI interacts with GPP/NEE and heat fluxes etc in the model*

We tried to leave model description as short as possible. However, more details on LAI-related processes might help and we included them.
We extended the process explanations in L125-141 for ECLand and in L147-160 for Noah-MP and added important equations to the appendix.

- *L149 'taken from LUT' I would replace this with 'were prescribed' or similar*

Changed to "…parameter values … of the 27 vegetation types are taken from look-up tables" (L147-148).

- *L152 'Thereby' I'm not sure this follows from the previous sentence*

Changed to "Among others, …" (L151).

- *L164 What does global initial data mean here? Did you conduct the model spin up using ERA5? Why didn't you choose the meteorology that comes with the flux sites (like you did with Noah-MP)?*

Both models have two types of input: Initial files (with initial values for some variables to start with) for model setup and time series files with meteorological data for model runs. The initial files contain variables like vegetation type, deep soil temperature, soil layering, soil type, initial soil moisture, vegetation cover fraction and initial LAI value or LAI climatology which are not all present in the FLUXNET data. But the variables included in the initial files differ for both models that is the reason why it sounds like different setups but they are not. For ECLand, these initial data files were prepared for a global setup already and we could make use of that. For Noah-MP, no such setup existed and we created the initial files by ourselves by using the information we had. After model initialization followed the spin-up phase so that these initial values were not used any longer and became overwritten by actually modelled values.
Added the sentence "These initial files contain information on albedo, orography, soil type, surface roughness and monthly LAI which is not available in the FLUXNET metadata" to give more background on that (L171-172).

- *L166 Which site information - do they all report vegetation height and you used a certain threshold?*

Clustering the vegetation into high or low vegetation type does not depend on vegetation height but on the vegetation type on-site. Forests in any case are high vegetation no matter how big the trees actually are.
Added "Forests and savannas were treated as high vegetation types while grasslands and croplands were allocated to low vegetation types. The vegetation type that fits most to the FLUXNET characterization was selected (see Tab. 1)" (L174-176).

- *L169 Is there a reference for the van Genuchten soil hydrologic params?*

Done (L178).

- *L171 Could you please update this reference - PLOS ONE is a journal and not the dataset name, and the author of the dataset surely is not 'The PLOS One Staff'!*

Done (L181).

- *L172 So the initial conditions for the two model runs are not identical? Why not?*

The reason for the initial conditions of the two models being different is only because these initial files look different for both models and require slightly different set of variables. Apart from that, we kept initial conditions as close to each other and as close to on-site conditions as possible.
Added "The models were set up as closely as possible to the available site information but there are some technical differences in the structure of the model input, i.e. in the initial files" (L164-165).

- *L173 I would have thought each gridcell has 8 neighboring gridcells?*

True. we checked whether soil type would change when including 8 neighboring cells compared to just 4 or even only the grid cell with the tower on it, but this was not the case.
Now, we took the soil type of the grid cell of the tower location and stated as follows: "… by selecting the grid cell including the flux tower location" (L181-182).

- *Table 2 Could you include the full name of the vegetation types and also the actual name of the Noah-MP vegetation classes? I'm not sure I mentioned it elsewhere but a similar table for ECLand would be great too*

Done (now Table 1).

- *L183 What does 'transferred' mean here? Did you calculate daily averages/sums?*

Yes, daily averages or sums (depend on variable).
Changed to "…were averaged/summed to daily values for direct comparison" (L223).

- *L188 You are also using the Pearson Corr. to assess non-linear relationships but this is not a suitable metric (Pearson Corr. only describes linear relationships; see further below)*

In principle, the relationship between observed and modelled values of a target variable is expected to be linear.

- *L189-190 Which symbol represents the normalized standard deviation metric?*

Added $s_n$ (L229).

- *L191-193 I'm not sure I understand why you subtract the minimum from $\bar{x}$ ? I also wonder how you interpret the relative bias when xmin differs strongly from $\bar{x}$ ?*

A "normal" relative bias was not applicable since our target variables (i.e. LE, H, NEE and GPP) have values that vary around zero. This results in relative biases that are not only really large partly but also difficult to interpret (e.g. reaching 3000% of relative bias but not because the model estimate is far away from observation but rather because the mean value is close to zero). By subtracting the minimum, the distribution is shifted to positive values only, with the minimum value being zero. As a result, the relative bias really contains an information on how much the estimates deviate from the mean since the reference system is the codomain of the variable. This works independently of the distance between $x_{min}$ and $x_{mean}$.
Changed to "For this purpose, the distribution of the observed values was shifted by their minimum, resulting in only positive values with a minimum of zero" (L232-233).

- *L199-200 So you try to understand the relationship between output variables within the same model simulation?*

Yes, exactly.

Changed to "To investigate the sensitivity of dynamically modelled vegetation on the model performance, we checked how strongly the quality of the model simulation of one target variable (e.g. LE) depends on the model quality of another one (e.g LAI). For this, we used the elasticity as a metric" (L241-243).

- *L199-207 I don't really understand how you calculate the elasticity. What is the 'slope of their correlation'? Is there a reference that uses the same definition for elasticity you could include? Is the -0.1 - 0.1 threshold common (reference)?*

Unfortunately, I found no publication from environmental sciences that use the same metric, only from economics.
Changed to "Elasticity is calculated as ratio of the change in one statistical measure (analogous to equation 2) for two different target variables" (L243-244).

- *L213 'The model performance of the dynamic run is shown with the symbol' - which symbol?*

All symbols that are in the Taylor plots.
Changed to "symbols" (L253).

- *L217-218 Can you name some references to support 'in line with results in the available literature'? It would also be useful to see some metric -> same for the next sentence about Noah-MP. How much more biased??*

Shifted to discussion section 4.1 and extended to "For Noah-MP, model quality metrics were in the range of other studies (Brunsell et al., 2020; Li et al., 2022; Xu et al., 2021; Liang et al., 2020). However, dynamic LAI modelled by Noah-MP in our assessment with a mean of +70 % was more biased compared to a mean of +20 % for annual LAI values reported by Li et al. (2022)" (L392-395).

- *L222 Here e.g. you compare the LUT LAI runs with your generated LAI forcing but in order to fully understand the performance difference it would be useful to know where this mysterious LUT LAI comes from??*

Here, we refer only to literature because Stevens et al. (2020) also replaced LUT LAI by MODIS LAI and compared model results. However, as stated in L194-196 ECLand LUT LAI is already MODIS-based.

- *L224 'simulation results were unaffected by the type of LAI forcing with vegetation dynamics switched on' I must have completely misunderstood your experiment set-up, but I thought when vegetation dynamics are switched on, LAI is simulated (not prescribed?)*

For the dynamic simulations, LAI is not prescribed but still part of the initial files. It is expectable that LAI predictions for the dynamic simulations are independent of the initial input. However, dynamic ECLand still incorporates prescribed LAI to 5% (RLAIINT=0.95 was defined by the developers' team to be fully dynamic).

- *L225 'but not necessarily for all sites' can you quantify this more?*

  Changed the sentence into "For ECLand, this was also the case for many sites but not necessarily for all, e.g. AT-Neu and AU-How" by adding examples (L259-260).

- *L226 Here for example (and across the entire section) it would be helpful to stick to the nomenclature for your experiment you defined in the table earlier, and highlight it with a different font type*

  Done.

- *L226 'increased variance' compared to what?*

  Increased variance in comparison with static ECLand simulations.
  Added "…compared to static simulations…" (L260-261).

- *L227 'was random' - what does this mean?*

  We could not find any tendencies regarding aridity or vegetation type to have positive or negative shift in relative bias.
  Replaced by "was ambiguous" (L263).

- *L231 What are sparse vegetation types (i.e. did you define it anywhere)?*

  No, we did not. Sparse vegetation are savannas and shrublands because they have no closed canopy surface.
  Added "Especially short (GRA+CRO) or sparse (SAV+WSA) vegetation types…" (L265).

- *L243-244 Can you also give a value range for ECLand - it'd help the comparison between the models*

  Added "…−30% on average…" (L280-281).

- *L246-252 Here you kind of say a bit about where original LUT is coming from, and this belongs in the methods section already*

  This is now discussed in L407-412.

- *L255 Why are you not showing the single-year LAI results - could just be in the appendix?*

  Relative bias for the MODIS single-year simulations are in the appendix tables. But they are not part of the Taylor plots.

- *Fig 2 How did you set threshold to separate the AI into different aridity classes? This belongs in the methods*

  We included the thresholds in the figure caption now as "The symbol colors indicate the site aridity (top right legend) as following: very humid - aridity index (AI) < 0.6, humid -

AI < 1.25, sub-humid - AI < 1.54, dry sub-humid - AI < 2, semi-arid - AI < 5, arid - AI ≥ 5 (Ashaolu and Iroye, 2018)" (Fig. 2).

- *Fig 3. I think you could use more distinct colors here, at least on my copy it's quite hard to see the difference in the shade of gray. Also why are there only single symbols for NEE and GPP (for dynamic simulations I think?) for Noah-MP? I also don't understand why you have different values for different prescribed LAI data for the dynamic simulations - but this is because I didn't quite understand the experiment set-up, as I said above. Does LAI not emerge from the spin-up (and therefore doesn't need to be prescribed)? The values do look very similar across all experiments. One more small thing, but the red dotted line is hard to see for the Pearson correlation and Normalized STD*

  Static Noah-MP produces no output for NEE and GPP which is according to model structure. Thus, only the values for the dynamic runs can be presented here. Colors changed, axes extended (Fig. 3).

- *L265 Here you dive into the impact of LAI-set-up on ecosystem exchange variables and it would be useful to see in the methods (at least to some degree) how LAI is actually linked to those variables?*

  This should work now since model description was extended, also by including process equations.

- *L281-282 Then why did you use Noah-MP in this study? This kind of defeats the purpose of your study, at least according to the title*

  Please see my answer to the seventh point of the general comments.

- *L284 I'm not sure this is the right conclusion - did you also find an overestimation of GPP? If NEE is right for the wrong reasons for the historical period that doesn't mean you can have a lot of confidence in simulated NEE in future scenarios*

  For most of the sites, GPP was overestimated with dynamic Noah-MP, but relative bias was predominantly small for forests (Tab. A3).
  We reformulated that conclusion and tried to be more precise: "Thus, although some previous studies found substantial uncertainties in modeled GPP for different vegetation types (Ma et al., 2017; Liang et al., 2020; Li et al., 2022), predicting ecosystem variables using dynamic Noah-MP could be useful at least for forests in studies when LAI climatology cannot be used such as climate change impact studies" (L449-452)

- *L289 'being independent of the prescribed LAI forcing' - so this is just the dynamic simulation that does not have any prescribed LAI?*

  In principle, the dynamic simulation has the prescribed LAI (since it is part of the setup file) but it is not using it and, thus, runs fully dynamically.

- *L289-290 Not sure consistent is the right word here, those are two v different experiments*

Changed to "In line with the finding that model performances of dynamic Noah-MP were independent…" (L315).

- *L291 How does the onsite LAI actually compare with the MODIS LAI? MODIS LAI can be biased for some flux sites (so a reason for the lower performance could just be that the model was tuned to match MODIS, and MODIS and on-site LAI are actually wildly different)*

On-site LAI and MODIS LAI were linearly correlated. MODIS LAI might be biased for some sites, but so might be on-site measured LAI due to technical limitations (scatter correction, saturation effect…). During development of the dynamic vegetation modules, a tuning of the parameter sets was done but not to MODIS LAI as target variable. However, mismatch between MODIS and on-site LAI is reflected in lower performance of NEE and GPP of the static ECLand simulations. The reason is unclear: It could be that on-site LAI does not reflect actual LAI but it could also be that calculations of GPP in relation to LAI do not match reality (similar to what we have shown in Fig. 8). For the dynamic ECLand runs, differences between MODIS and on-site LAI play only a minimal role since 95% of the LAI calculations come from dynamically predicted LAI and NEE and GPP predictions are even fully dynamically predicted.

- *L306-307 Where is this recommendation coming from? Did you not say earlier there is no performance difference depending on the LAI forcing used?*

The performance is not different for the dynamic simulations. But for the static runs, it is. Thus, we recommended here to use static simulations with MODIS climatology forcing.
Shifted to section "4.4 Implications" (L580-581).

- *L309-316 A lot of these studies look at different temporal resolutions, and I actually wondered too how your model simulations would do if you looked at climatologies, or annually aggregated values - this could help you identify whether the models get the broad patterns right. But having said that, you already cover a lot of ground with your study so this is more of a curiosity rather than an actual suggestion for the revision*

Yes, I agree, definitively would be interesting to look at but, as you said, was beyond the scope of this investigation.

- *L329-330 Interesting indeed - do you have any idea why this is happening?*

It might be that carbon and water transport processes are coupled not tightly enough. With NEE estimates fitting well, the photosynthetic activity also is good captured by the model. The demand of water by the photosynthesis might be underestimated by the model and, leading to less transpiration and, thus, also to a lower fraction of energy that is used for latent heat transport. Additionally, downward CO2 transport and upward water transport through turbulent fluxes occurs in the same eddies which is not captured by the model. These are just some ideas on that so far. We refrained from intensifying the discussion on that.

- *L335 Again - why not include a figure in the appendix to show these results?*

Done (Fig. A2).

- *L351 How are vegetation and soil moisture state variables coupled?*

Vegetation needs water for photosynthesis which stems from the soil. Thus, more photosynthetically active biomass extracts more water from the soil and, otherwise, less soil water restricts maximum plant productivity and biomass build-up.
This is further discussed in L495-503.

- *L354-359 I think this is all valuable and true, and explains the general soil moisture bias well. But it doesn't explain your actual results, i.e. not seeing an impact of static vs dynamic vegetation on soil moisture*

Yes, you are right. The reason for unaffected soil moisture to vegetation dynamics still remains unclear. Referring to the point before, it could be due to the implemented interaction of carbon and water processes. First, the potential photosynthetic activity in dependence of leaf area and radiative conditions is calculated. Then, the limitation factor of extractable water is estimated according to available soil water and roots. Lastly, the photosynthetic activity is adapted to that restriction and transpiration rate adapted to conductivity and atmospheric conditions. As a result, the only included path is that soil moisture impacts photosynthetic activity and biomass build-up. But there is no feedback that more biomass needs/loses more water that will be taken from the soil because photosynthetic activity relates only to the carbon fluxes but not to the water fluxes.
We added this explanation to the text (L495-503).

- *Fig 7 Where is the footnote for elasticity?*

Sorry, that footnote was there by accident.
Removed.

- *L389-396 Why didn't you show LAI-GPP elasticity in Fig 7 when this is such a strong focus in this paragraph? Do you have an idea whether it is more realistic to have a linear or non-linear LAI-GPP relationship (i.e. what do studies say that look at observed LAI-GPP relationships)?*

Other studies also found a linear relationship between LAI and GPP but with large variability. Some sites might be exceptions from the linearity (IT-Ren) where LAI-GPP relationship appears to be a non-linear saturation function.
We now added LAI-GPP and LAI-NEE elasticity for ECLand in Figure 7 but excluded NEE-LE and NEE-SM instead (Fig. 7).

- *Fig 8 What are the arrows in the figure? Some of the relationships in this figure (as you point out earlier) are clearly non-linear, and therefore using the Pearson correlation*

*coefficient is not suitable. I would also suggest to change the GPP units to avoid having so many decimals - in this figure and also in the text*

Since the most probable in the observations LAI-GPP relation is a linear one, Pearson correlation coefficient is the statistical basis of this linear regression and also the measure for the relationships from the model output. We cannot compare different kinds of correlation coefficients.
We added description of the arrows to the figure caption with "The arrows represent the range of GPP and LAI values for the individual seasons" (Fig. 8).

- *L403 You need to define σr*

  Done (L371).

- L409-410 You already say in the methods that you are masking MODIS using quality flags (and as I said there, please be more precise about what the flags mean)

  This is now part of the section 4.4 "Limitations" (L592-597).

- *L454-455 Can you give a reference for this ('real LAI'), also maybe replace real with 'observed' or similar*

  We cannot replace "real" by "observed" because we are not referring to any measured values here. The reality this sentence is referring to is the fact that trees do not immediately lose their leaves when they are faced to a few days of suboptimal conditions for photosynthesis.
  Replaced by "realistic" (L571).

- *L461-462 I think this is a bit harsh and not true; depending on the focus (i.e. if it's purely benchmarking) of course this can happen, but there are also many papers that actually explain reasons for mismatches between obs and model simulations, and also differences across members of the LSM ensembles (see e.g. Whitley et al. 2016)*

  Not in the manuscript anymore.

- *L467 Why did you expect that dynamic vegetation would improve ecosystem exchange especially for short vegetation?*

  Compared to forests that are more resistant and resilient for e.g. water scarcity, short vegetation more dynamically and more instantly responds to environmental limitations for its growth. Thus, firstly, assuming the same LAI cycle for each year and, secondly, assuming a constant LAI values over a whole month as in the static model simulations, do not represent reality. Our expectation was that modelling vegetation dynamically would cope for that variability and, as a result, yield in better performance of observed ecosystem fluxes.

- *L480-481 If you don't know how representative your models are compared to other LSMs then why did you choose them? I see both are part of the PLUMBER2 experiment - how do they compare to other LSMs there (see Abramowitz et al., 2024)?*

Other models have processes implemented differently. So, there is no chance in directly transferring results and conclusions from these two models to others.
Added "…since they have processes implemented differently" (L624).

Literature

Best, M. J., Abramowitz, G., Johnson, H. R., Pitman, A. J., Balsamo, G., Boone, A., Cuntz, M., Decharme, B., Dirmeyer, P. A., Dong, J., Ek, M., Guo, Z., Haverd, V., van den Hurk, B. J. J., Nearing, G. S., Pak, B., Peters-Lidard, C., Santanello, J. A., Stevens, L., and Vuichard, N.: The Plumbing of Land Surface Models: Benchmarking Model Performance, Journal of Hydrometeorology, 16, 1425–1442, https://doi.org/10.1175/jhm-d-14-0158.1, 2015

Boussetta, S., Balsamo, G., Beljaars, A., Kral, T. & Jarlan, L.: Impact of a satellite-derived leaf area index monthly climatology in a global numerical weather prediction model, International Journal of Remote Sensing, 34:9-10, 3520-3542, DOI: 10.1080/01431161.2012.716543, 2013.

Haughton, N., Abramowitz, G., Pitman, A. J., Or, D., Best, M. J., Johnson, H. R., Balsamo, G., Boone, A., Cuntz, M., Decharme, B., Dirmeyer, P. A., Dong, J., Ek, M., Guo, Z., Haverd, V., van den Hurk, B. J. J., Nearing, G. S., Pak, B., Santanello, J. A., J., Stevens, L. E., and Vuichard, N.: The plumbing of land surface models: is poor performance a result of methodology or data quality?, J Hydrometeorol, 17, 1705–1723, https://doi.org/10.1175/JHM-D-15-0171.1, 2016.

Haughton, N., Abramowitz, G., De Kauwe, M. G., and Pitman, A. J.: Does predictability of fluxes vary between FLUXNET sites?, Biogeosciences, 15, 4495–4513, https://doi.org/10.5194/bg-15-4495-2018, 2018a.

Stevens, D., Miranda, P. M. A., Orth, R., Boussetta, S., Balsamo, G., and Dutra, E.: Sensitivity of Surface Fluxes in the ECMWF Land Surface Model to the Remotely Sensed Leaf Area Index and Root Distribution: Evaluation with Tower Flux Data, Atmosphere, 11, https://doi.org/10.3390/atmos11121362, 2020.

---

## Author Response (AR3)

**Response on Referee's third round review**

First of all, we want to thank you for accomplishing another round of reviewing this work of me and my co-authors and your hard work during the review process. In the following, I will go through and respond to your final comments. Please note that reviewer comments are in italic, our responses in normal font and an explanation of changes/adaptations made by us in blue font.

**Comments**

- *Line 165: "but resetting all variables that would be dynamically predicted within the same function". I do not think this is clear about the steps taken. I assume this is meant to clarify that, in the static runs, it is only GPP and NEE that are modified and that the rest of the model remains in its static configuration?*

  Yes, you are right. The intention of that sentence was to confirm that we assured to keep the static setup as it was.
  Changed to "... but resetting all variables that would be dynamically predicted within the same function to their prior values. This assured that the model still ran in static configuration" (Line 159-161).

- *Line 195: "The Noah-MP simulations were done with soil parametrization from look-up tables, Ball-Berry stomatal resistance approach with using matric potential". Please correct the grammar in this sentence to make the implementation clear.*

  Adapted to "The Noah-MP simulations were done with soil parameterization from look-up tables and Ball-Berry stomatal resistance approach (Ball et al., 1987; Bonan, 1996) using a matric potential limitation" (Line 190-191).

- *Table 2: This detail is good but should likely be in Supplementary Information. If it is included for Noah-MP, consistency would suggest the same information be supplied in the same table for ECLand.*

  Creating the same sort of table for ECLand is difficult since there is no option to choose from different approaches for processes, rather just switching them on or off.
  We listed the used options for ECLand processes now as well in a table. Both tables are now in the appendix (Tab. A1 and A2).

- *Line 317: "lowered from −32 % - +69 % to −28 % - +42 %" is confusing to read with the hyphens and minus signs. I would suggest something like "lowered from between −32 % and +69 % to between −28 % and +42 %".*

  Done.